# Complexity Analysis of Normalizing Constant Estimation: from Jarzynski Equality to Annealed Importance Sampling and beyond

**Wei Guo, Molei Tao, Yongxin Chen**
Georgia Institute of Technology
{wei.guo, mtao, yongchen}@gatech.edu

## Abstract

Given an unnormalized probability density $\pi \propto \mathrm{e}^{-V}$, estimating its normalizing constant $Z = \int_{\mathbb{R}^d} \mathrm{e}^{-V(x)}\mathrm{d}x$ or free energy $F = -\log Z$ is a crucial problem in Bayesian statistics, statistical mechanics, and machine learning. It is challenging especially in high dimensions or when $\pi$ is multimodal. To mitigate the high variance of conventional importance sampling estimators, annealing-based methods such as Jarzynski equality and annealed importance sampling are commonly adopted, yet their quantitative complexity guarantees remain largely unexplored. We take a first step toward a non-asymptotic analysis of annealed importance sampling. In particular, we derive an oracle complexity of $\widetilde{O}\left(\frac{d\beta^2 \mathcal{A}^2}{\varepsilon^4}\right)$ for estimating $Z$ within $\varepsilon$ relative error with high probability, where $\beta$ is the smoothness of $V$ and $\mathcal{A}$ denotes the action of a curve of probability measures interpolating $\pi$ and a tractable reference distribution. Our analysis, leveraging Girsanov's theorem and optimal transport, does not explicitly require isoperimetric assumptions on the target distribution. Finally, to tackle the large action of the widely used geometric interpolation, we propose a new algorithm based on reverse diffusion samplers, establish a framework for analyzing its complexity, and empirically demonstrate its efficiency in tackling multimodality.

## 1 Introduction

We study the problem of estimating the normalizing constant $Z = \int_{\mathbb{R}^d} \widehat{\pi}(x)\mathrm{d}x$ of an unnormalized probability density function (p.d.f.) $\pi \propto \widehat{\pi} := \mathrm{e}^{-V}$ on $\mathbb{R}^d$, so that $\pi(x) = \frac{\widehat{\pi}(x)}{Z}$. The normalizing constant appears in various fields: in Bayesian statistics, when $\widehat{\pi}$ is the product of likelihood and prior, $Z$ is also referred to as the marginal likelihood or evidence (Gelman et al., 2013); in statistical mechanics, when $V$ is the Hamiltonian,[1] $Z$ is known as the partition function, and $F := -\log Z$ is called the free energy (Chipot & Pohorille, 2007; Lelièvre et al., 2010; Pohorille et al., 2010). The task of normalizing constant estimation has numerous applications, including computing log-likelihoods in probabilistic models (Sohl-Dickstein & Culpepper, 2012), estimating free energy differences (Lelièvre et al., 2010), and training energy-based models in generative modeling (Song & Kingma, 2021; Carbone et al., 2023; Sander et al., 2025).

Estimating normalizing constants is challenging in high dimensions or when $\pi$ is multimodal (i.e., $V$ has a complex landscape). Conventional approaches based on importance sampling (Meng & Wong, 1996) are widely adopted to tackle this problem, but they suffer from high variance due to the mismatch between the proposal and the target when $\pi$ is complicated (Chatterjee & Diaconis, 2018). To alleviate this issue, the technique of *annealing* tries constructing a sequence of intermediate distributions that bridge these two distributions, which motivates several popular methods including path sampling (Chen & Shao, 1997; Gelman & Meng, 1998), annealed importance sampling (AIS, Neal (2001)), and sequential Monte Carlo (SMC, Doucet et al. (2000); Del Moral et al. (2006); Syed et al. (2024)) in statistics literature, as well as thermodynamic integration (TI, Kirkwood (1935)) and

---

[1]Up to a multiplicative constant $\beta = \frac{1}{k_{\mathrm{B}}T}$ known as the thermodynamic beta, where $k_{\mathrm{B}}$ is the Boltzmann constant and $T$ is the temperature. When borrowing physical terminologies, we ignore this for simplicity.

Jarzynski equality (JE, Jarzynski (1997); Ge & Jiang (2008); Hartmann et al. (2019)) in statistical mechanics literature. In particular, JE points out the connection between the free energy difference between two states and the work done over a series of trajectories linking these two states, while AIS constructs a sequence of intermediate distributions and estimates the normalizing constant by importance sampling over these distributions. These two methods are our primary focus in this paper.

Despite the empirical success of annealing-based methods (Ma et al., 2013; Krause et al., 2020; Mazzanti & Romero, 2020; Yasuda & Takahashi, 2022; Chen & Ying, 2024; Schönle et al., 2025), the theoretical understanding of their performance is still limited. Existing works for importance sampling mainly focus on the **asymptotic** bias and variance of the estimator (Meng & Wong, 1996; Gelman & Meng, 1998), while works on JE usually simplify the problem by assuming the work follows simple distributions (e.g., Gaussian or gamma) (Echeverria & Amzel, 2012; Arrar et al., 2019). Moreover, only analyses asymptotic in the number of particles derived from central limit theorem exist (Lelièvre et al., 2010, Sec. 4.1). This paper aims to establish a rigorous **non-asymptotic** analysis of estimators based on JE and AIS, while introducing minimal assumptions on the target distribution. We also propose a new algorithm based on reverse diffusion samplers to tackle a shortcoming of AIS.

**Contributions.** Our key technical contributions are summarized as follows.

**1.** We discover a novel strategy for analyzing the complexity of normalizing constant estimation, applicable to a wide range of target distributions (Assumps. 1 and 2) that may not satisfy isoperimetric conditions such as log-concavity.

**2.** In Sec. 3, we study JE and prove an upper bound on the time required for running the annealed Langevin dynamics to estimate the normalizing constant within $\varepsilon$ relative error with high probability. The final bound depends on the action (the integral of the squared metric derivative in Wasserstein-2 distance) of the curve.

**3.** Building on the insights from this analysis of the continuous dynamics, in Sec. 4 we establish the first non-asymptotic oracle complexity bound for AIS, representing the first analysis of normalizing constant estimation algorithms without assuming a log-concave target distribution.

**4.** Finally, in Sec. 5, we first point out a potential limitation of the commonly used geometric interpolation, which provides a quantitative explanation of the mass teleportation phenomenon. We then propose a series of new algorithms based on reverse diffusion samplers and formalize a framework for analyzing its oracle complexity. Our experimental results demonstrate the superiority of the proposed algorithm over AIS in overcoming multimodality.

**Related Works.** Below, we summarize the related works in four aspects.

**I. Methods for normalizing constant estimation.** We mainly discuss two classes of methods here. First, the *equilibrium* methods, such as TI (Kirkwood, 1935) and its variants (Brosse et al., 2018; Ge et al., 2020; Chehab et al., 2023; Kook & Vempala, 2025), which involve sampling sequentially from a series of equilibrium Markov transition kernels. Second, the *non-equilibrium* methods, such as AIS (Neal, 2001), which samples from a non-equilibrium stochastic process that gradually evolves from a prior distribution to the target distributions. In App. H.1, we show that TI is a special case of AIS using the "perfect" transition kernels. Recent years have also witnessed the emergence of *learning-based* non-equilibrium methods, which are typically byproducts of neural samplers (Nüsken & Richter, 2021; Zhang & Chen, 2022; Máté & Fleuret, 2023; Richter & Berner, 2024; Sun et al., 2024; Vargas et al., 2024; Máté et al., 2024; Albergo & Vanden-Eijnden, 2025; Blessing et al., 2025; Chen et al., 2025; Havens et al., 2025; Du et al., 2025). Finally, there are also methods based on particle filtering (Kostov & Whiteley, 2017; Jasra et al., 2018; Ruzayqat et al., 2022).

**II. Variance reduction in JE and AIS.** Our proof methodology focuses on the discrepancy between the sampling path measure and the reference path measure, which is related to the variance reduction technique in applying JE and AIS. For example, Vaikuntanathan & Jarzynski (2008) introduced the idea of escorted simulation, Hartmann et al. (2017) proposed a method for learning the optimal control protocol in JE through the variational characterization of free energy, and Doucet et al. (2022) leveraged score-based generative model to learn the optimal backward kernel. Quantifying the discrepancy between path measures is the core of our analysis.

**III. Complexity analysis for normalizing constant estimation.** Chehab et al. (2023) studied the asymptotic statistical efficiency of the curve for TI measured by the asymptotic mean-squared error,

and highlighted the advantage of the geometric interpolation. In terms of non-asymptotic analysis, existing works mainly rely on the isoperimetry of the target distribution. For instance, Andrieu et al. (2016) derived bounds of bias and variance for TI under Poincaré inequality (PI), Brosse et al. (2018) provided complexity guarantees for TI under both strong and weak log-concavity conditions, while Ge et al. (2020) improved the complexity under strong log-concavity using multilevel Monte Carlo.

**IV. Complexity analysis of sampling beyond isoperimetry.** Our analysis of estimating normalizing constants of non-log-concave distributions is also closely related to the study of sampling beyond log-concavity. In general, such problems are NP hard (Ge et al., 2018; He & Zhang, 2025). Existing works providing convergence guarantees have leveraged more general isoperimetric inequalities such as weak PI (Mousavi-Hosseini et al., 2023), tried to establish convergence in weaker notions (Balasubramanian et al., 2022; Cheng et al., 2023), or utilized denoising diffusion models (Huang et al., 2024a; He et al., 2024). We highlight Guo et al. (2025) that this paper mainly draws inspiration from, which introduced the action of a curve in quantifying the convergence of annealed sampling. While they focused on sampling and presented cases where annealing works, we extend the analysis to a conceptually different task, and further establish lower bounds on the action of the commonly used geometric interpolation, motivating a new algorithm based on reverse diffusion samplers.

**Notations and Definitions.** For $a, b \in \mathbb{R}$, let $[\![a, b]\!] := [a, b] \cap \mathbb{Z}$, $a \wedge b := \min(a, b)$, and $a \vee b := \max(a, b)$. For $a, b > 0$, the notations $a \lesssim b$, $b \gtrsim a$, $a = O(b)$, $b = \Omega(a)$ indicate that $a \leq Cb$ for some universal absolute constant $C > 0$, and the notations $a \asymp b$, $a = \Theta(b)$ stand for $a \lesssim b \lesssim a$. $\widetilde{O}(\cdot), \widetilde{\Theta}(\cdot)$ hide logarithmic dependence in $O(\cdot), \Theta(\cdot)$. A function $U \in C^2(\mathbb{R}^d)$ is $\alpha(> 0)$-strongly-convex if $\nabla^2 U \succeq \alpha I$, and is $\beta(> 0)$-smooth if $-\beta I \preceq \nabla^2 U \preceq \beta I$. We do not distinguish probability measures on $\mathbb{R}^d$ from their Lebesgue densities. For two probability measures $\mu, \nu$, the total-variation (TV) distance is $\mathrm{TV}(\mu, \nu) = \sup_{\text{measurable } A} |\mu(A) - \nu(A)|$, and the Kullback-Leibler (KL) divergence is $\mathrm{KL}(\mu \| \nu) = \int \log \frac{\mathrm{d}\mu}{\mathrm{d}\nu} \mathrm{d}\mu$. Finally, a function $T : \mathbb{R}^d \times \mathbb{R}^d \to [0, +\infty)$ is a transition kernel if for any $x$, $T(x, \cdot)$ is a p.d.f. Throughout this paper, $(B_t)$ and $(W_t)$ represent standard Brownian motions (BM) on $\mathbb{R}^d$.

**Preliminaries.** For brevity, we integrate the required background information into the main text, with a detailed exposition available in App. A.

## 2 PRELIMINARIES AND PROBLEM SETTING

To motivate the study of normalizing constant estimation, we first present several examples.

**Example 1.** [*Free energy difference.*] In many statistical physics problems (Lelièvre et al., 2010), given two energy functions $U_0, U_1$ (possibly linked through some thermodynamic process), one is often interested in estimating the free energy difference $\Delta F := -\frac{1}{\beta} \log(\int e^{-\beta U_1} \mathrm{d}x / \int e^{-\beta U_0} \mathrm{d}x)$, which is related with the normalizing constant of the distributions $\pi_i \propto e^{-\beta U_i}$.

**Example 2.** [*Likelihood in latent variable models.*] In latent variable models such as variational autoencoders (Kingma & Welling, 2013), a common evaluation metric is the marginal likelihood of a data point $x$, $p_\theta(x) = \int p_\theta(x|z) p(z) \mathrm{d}z$. This is nothing but the normalizing constant of the posterior distribution of the latent variable $z$ given data $x$, $p_\theta(z|x) \propto_z p_\theta(x|z) p(z)$.

**Example 3.** [*Volume of convex bodies.*] In theoretical computer science, a classical problem is to estimate the volume of a convex body $\mathcal{K}$ (Dyer et al., 1991; Cousins & Vempala, 2018; Kook et al., 2024), which is equivalent to the normalizing constant of the uniform distribution on $\mathcal{K}$, $\pi \propto 1_\mathcal{K}$.

Building on prior theoretical results (Brosse et al., 2018; Ge et al., 2020), we study the oracle complexity of estimating the normalizing constant of a density under the following criterion:

> **Aim:** Given a density $\pi \propto \widehat{\pi} := e^{-V}$ on $\mathbb{R}^d$, bound the complexity of obtaining an estimator $\widehat{Z}$ of $Z = \int_{\mathbb{R}^d} \widehat{\pi}(x) \mathrm{d}x$ such that with constant probability, the relative error is within $\varepsilon(\ll 1)$:
>
> $$\Pr\left(\left|\frac{\widehat{Z}}{Z} - 1\right| \leq \varepsilon\right) \geq \frac{3}{4}. \tag{1}$$

**Remark 1.** *We make two remarks regarding* (1). *First, similar to how taking the mean of i.i.d. estimates reduces variance, we show in Lem. 10 that the probability above can be boosted to $1 - \zeta$,*

$\forall \zeta \in \left(0, \frac{1}{4}\right)$ using the median trick: obtaining $O\left(\log \frac{1}{\zeta}\right)$ i.i.d. estimates satisfying (1) and taking their median. Therefore, we focus on the task of obtaining a single estimate satisfying (1) hereafter. Second, (1) also allows us to quantify the complexity of estimating the free energy $F = -\log Z$, which is often of greater interest in statistical mechanics than the partition function $Z$. We show in App. G that estimating $Z$ with $O(\varepsilon)$ relative error and estimating $F$ with $O(\varepsilon)$ absolute error share the same complexity up to constants. Further discussion of this guarantee, including a literature review and the comparison with bias and variance, is deferred to App. G.

A straightforward method for estimating $Z$ is through *importance sampling*, i.e., $Z = \mathbb{E}_{\pi_0} \frac{\hat{\pi}}{\pi_0}$ for some tractable proposal distribution $\pi_0$, yet its variance can be large due to the mismatch between $\pi_0$ and $\pi$. The rationale behind **annealing** involves a gradual transition from $\pi_0$ to $\pi_1 = \pi$. Throughout this paper, we consider a curve of probability measures denoted as

$$\left(\pi_\theta = \frac{1}{Z_\theta} e^{-V_\theta}\right)_{\theta \in [0,1]},$$

where $V_1 = V$ is the potential of $\pi$, and $Z_1 = Z$ is what we need to estimate. We do not specify the exact form of this curve now, but only introduce the following mild regularity assumption on the curve, as assumed in classical textbooks such as Ambrosio et al. (2008; 2021); Santambrogio (2015):

**Assumption 1.** *The potential* $[0,1] \times \mathbb{R}^d \ni (\theta, x) \mapsto V_\theta(x) \in \mathbb{R}$ *is jointly* $C^1$, *and the curve* $(\pi_\theta)_{\theta \in [0,1]}$ *is **absolute continuous** with finite **action*** $\mathcal{A} := \int_0^1 |\dot{\pi}|_\theta^2 \, d\theta$.

Here, $|\dot{\pi}|_\theta := \lim_{\delta \to 0} \frac{W_2(\pi_{\theta+\delta}, \pi_\theta)}{|\delta|}$ is the Wasserstein-2 (W$_2$) **metric derivative** of the curve $(\pi_\theta)_{\theta \in [0,1]}$ at $\theta$, which measures the "speed" of the curve in the space of probability distributions, and **absolute continuity** means the above limit exists and is finite for all $\theta \in [0,1]$. A curve having a finite action is a weaker condition than requiring each $\pi_\theta$ to satisfy isoperimetric inequalities (e.g., Poincaré or log-Sobolev). We refer readers to App. A.2 for details of optimal transport (OT), and in particular, we highlight the connection between the metric derivative and the continuity equation (Lem. 4), which will serve as a key tool in our analysis.

For the purpose of non-asymptotic analysis, we further introduce the following mild assumption:

**Assumption 2.** *$V$ is $\beta$-smooth, $\nabla V(0) = 0$, and $m := \sqrt{\mathbb{E}_\pi \|\cdot\|^2} < +\infty$.*

**Remark 2.** *One can always find a stationary point $x_*$ of (possibly non-convex) $V$ using optimization methods within negligible cost compared with the complexity for estimating $Z$. By considering the translated distribution $\pi(\cdot - x_*)$, we assume $0$ is a stationary point without loss of generality.*

Equipped with this fundamental setup, we now proceed to introduce the JE and AIS, and establish an analysis for their complexity.

## 3    ANALYSIS OF THE JARZYNSKI EQUALITY

To elucidate how annealing works in the task of normalizing constant estimation, we first consider **annealed Langevin diffusion (ALD)**, which runs **Langevin diffusion (LD)** with a dynamically changing target distribution. Recall that the LD with target distribution is the stochastic differential equation (SDE) $dX_t = \nabla \log \pi(X_t) dt + \sqrt{2} dB_t$, which converges to $\pi$ as $t \to \infty$. To define ALD, we introduce a reparameterized curve $(\widetilde{\pi}_t = \pi_{t/T})_{t \in [0,T]}$ for some large time duration $T$ to be determined later, and consider the following SDE:

$$dX_t = \nabla \log \widetilde{\pi}_t(X_t) dt + \sqrt{2} dB_t, \; t \in [0,T]; \; X_0 \sim \widetilde{\pi}_0. \tag{2}$$

The following Jarzynski equality provides a connection between the work functional and the free energy difference, which naturally yields an estimator of normalizing constant.

**Theorem 1** (Jarzynski equality (Jarzynski, 1997)). *Let $\mathbb{P}^\rightarrow$ be the path measure of (2). Then the work functional $W$ and the free energy difference $\Delta F$ have the following relation:*

$$\mathbb{E}_{\mathbb{P}^\rightarrow} e^{-W} = e^{-\Delta F}, \quad \text{where } W(X) := \frac{1}{T} \int_0^T \partial_\theta V_\theta|_{\theta = \frac{t}{T}}(X_t) dt \text{ and } \Delta F := -\log \frac{Z_1}{Z_0}.$$

Below, we sketch the proof from Vargas et al. (2024, Prop. 3.3), which offers a crucial aspect for our analysis: the forward and backward SDEs. See App. A.1 for a detailed introduction.

**Sketch of Proof** Let $\mathbb{P}^{\leftarrow}$ be the path measure of the following backward SDE with time-reversed Brownian motion (BM) $(B_t^{\leftarrow})_{t \in [0,T]}$ (i.e., $(t \mapsto B_{T-t}^{\leftarrow})_{t \in [0,T]}$ is a standard BM, see Def. 1):

$$\mathrm{d}X_t = -\nabla \log \widetilde{\pi}_t(X_t)\mathrm{d}t + \sqrt{2}\mathrm{d}B_t^{\leftarrow}, \ t \in [0,T]; \ X_T \sim \widetilde{\pi}_T. \tag{3}$$

Intuitively, this is running the ALD backward in time from $T$ to $0$, targeting distribution $\widetilde{\pi}_t$ at time $t$. Leveraging the Girsanov's theorem (Lem. 1) and Itô's formula, one can establish the following identity of the Radon-Nikodým (RN) derivative between the forward and backward path measures, known as the *Crooks fluctuation theorem* (Crooks, 1998; 1999):

$$\log \frac{\mathrm{d}\mathbb{P}^{\rightarrow}}{\mathrm{d}\mathbb{P}^{\leftarrow}}(X) = -\int_0^T (\partial_t \log \widetilde{\pi}_t)(X_t)\mathrm{d}t = W(X) - \Delta F, \quad \text{a.s. } X \sim \mathbb{P}^{\rightarrow}, \tag{4}$$

which implies JE by the identity $\mathbb{E}_{\mathbb{P}^{\rightarrow}} \frac{\mathrm{d}\mathbb{P}^{\leftarrow}}{\mathrm{d}\mathbb{P}^{\rightarrow}} = 1$. Complete proof can be found in App. C.1. $\qquad\square$

Under the *ideal* setting where (i) $Z_0$ is known, (ii) the ALD in (2) can be simulated exactly, and (iii) the work functional $W(X)$ can be computed precisely, Thm. 1 provides an unbiased estimator $\widehat{Z} := Z_0 \mathrm{e}^{-W(X)}$ for $Z = Z_0 \mathrm{e}^{-\Delta F}$. Despite its dominant use (Chipot & Pohorille, 2007; Lelièvre et al., 2010), the statistical efficiency of this estimator is not well understood. While it is known that the variance of $\widehat{Z}$ can be large, *non-asymptotic* analyses quantifying its efficiency is lacking. We address this gap by establishing an upper bound on the time $T$ required for the ALD to satisfy the accuracy criterion (1) in the following theorem, whose proof is detailed in App. C.2.

**Theorem 2.** *Under Assump. 1, it suffices to choose* $T = \frac{32\mathcal{A}}{\varepsilon^2}$ *to obtain* $\Pr\left(\left|\frac{\widehat{Z}}{Z} - 1\right| \le \varepsilon\right) \ge \frac{3}{4}$.

We first observe that our bound aligns with the decay rate of the variance of the work in Mazonka & Jarzynski (1999) (see also Lelièvre et al. (2010, Chap. 4.1.4)), which considered a special case $\pi_\theta = \mathcal{N}\left(\theta L, \frac{1}{K}\right)$. They showed that $W \sim \mathcal{N}(B_T, 2B_T)$ with $B_T = \frac{L^2}{T}\left(1 - \frac{(1-\mathrm{e}^{-KT})}{KT}\right)$, and hence the *normalized variance* $\mathrm{Var}_{\mathbb{P}^{\rightarrow}} \frac{\widehat{Z}}{Z} = \mathrm{e}^{2B_T} - 1$ is **asymptotically** $O\left(\frac{1}{T}\right)$ as $T \to \infty$. Our bound, under a different criterion (1), is $O\left(\frac{1}{T}\right)$ for **all** $T > 0$.

To illustrate the proof idea of Thm. 2, note that while the ALD (2) targets the distribution $\widetilde{\pi}_t$ at time $t$, there is always a lag between $\widetilde{\pi}_t$ and the actual law of $X_t$. Similarly, the same lag exists in the backward ALD (3). This lag turns out to be the source of the error in the estimator $\widehat{Z}$.

In practice, to alleviate the issue of high variance in estimating free energy differences, Vaikuntanathan & Jarzynski (2008) proposed adding a compensatory drift term $v_t(X_t)$ to the ALD (2). Ideally, the optimal choice would eliminate the lag entirely, ensuring $X_t \sim \widetilde{\pi}_t$ for all $t \in [0, T]$. Inspired by this, we compare the path measure of ALD $\mathbb{P}^{\rightarrow}$ to the SDE having the perfect compensatory drift term, whose path measure $\mathbb{P}$ has marginal distribution $\widetilde{\pi}_t$ at time $t$. To make possible the perfect match, it turns out that $v_t$ must satisfy the Fokker-Planck equation with $\widetilde{\pi}_t$. The Girsanov's theorem (Lem. 1) enables the computation of $\mathrm{KL}(\mathbb{P}\|\mathbb{P}^{\rightarrow})$ and $\mathrm{KL}(\mathbb{P}\|\mathbb{P}^{\leftarrow})$, which are related to $\|v_t\|_{L^2(\widetilde{\pi}_t)}^2$. Finally, among all admissible drift terms $v_t$, Lem. 4 suggests an optimal choice of $v_t^*$ to minimize this norm, thereby leading to the metric derivative $|\dot{\widetilde{\pi}}|_t$ and the action $\mathcal{A}$. This way avoids the explicit dependence of isoperimetric assumptions in our bound.

A similar connection between free energy and action integral was discovered in stochastic thermodynamics (Sekimoto, 2010; Seifert, 2012), one paradigm for non-equilibrium thermodynamics. By the second law of thermodynamics, the averaged dissipated work, defined as the averaged work minus the free energy difference, i.e., $\mathcal{W}_{\mathrm{diss}} := \mathcal{W} - \Delta F := \mathbb{E}_{\mathbb{P}^{\rightarrow}} W - \Delta F$, is non-negative. When the underlying process is modeled by an overdamped LD, $\mathcal{W}_{\mathrm{diss}}$ can be quantified by an action integral divided by the time duration (Aurell et al., 2011; Chen et al., 2020). This follows from the observation that $\mathcal{W}_{\mathrm{diss}} = \mathrm{KL}(\mathbb{P}^{\rightarrow}\|\mathbb{P}^{\leftarrow})$ and then a similar argument to that above. This connection provides a finer description of the second law of thermodynamics (Aurell et al., 2012) over a finite time horizon.

Finally, we place Thm. 2 within the broader theme of **sampling v.s. normalizing constant estimation** by comparing Thm. 2 with the complexity of non-log-concave sampling. Guo et al. (2025) proved

that under the same assumptions, the ALD (2) can draw a sample within $\varepsilon^2$-error in $\mathrm{KL}(\pi\|\cdot)$ with the same order of time $T \asymp \frac{A}{\varepsilon^2}$. While the classical work Jerrum et al. (1986) proved the existence of a polynomial-time algorithm for sampling and a polynomial-time algorithm for estimating normalizing constant imply each other in the *discrete* settings, we establish a similar quantitative connection between the complexities of these two tasks in the *continuous* settings *without* log-concavity, opening a new avenue of research on understanding their relationship. Though reaching similar results, the proof strategies are different: Guo et al. (2025) is a direct application of Girsanov's theorem between $\mathbb{P}^{\rightarrow}$ and $\mathbb{P}$, while Thm. 2 involves more complicated backward SDE arguments.

## 4 ANALYSIS OF THE ANNEALED IMPORTANCE SAMPLING

In practice, it is not feasible to simulate the ALD precisely, nor is it possible to evaluate the exact value of the work $W(X)$. Therefore, discretization and approximation are required. To address this, we first outline the following annealed importance sampling (AIS) equality akin to JE.

**Theorem 3** (Annealed importance sampling equality (Neal, 2001))**.** *Suppose we have probability distributions* $\pi_\ell = \frac{f_\ell}{Z_\ell}$, $\ell \in [\![0, M]\!]$ *and transition kernels* $F_\ell(x, \cdot)$, $\ell \in [\![1, M]\!]$, *and assume that each* $\pi_\ell$ *is an invariant distribution of* $F_\ell$, $\ell \in [\![1, M]\!]$. *Define the path measure*

$$\mathbb{P}^{\rightarrow}(x_{0:M}) = \pi_0(x_0) \prod_{\ell=1}^{M} F_\ell(x_{\ell-1}, x_\ell). \tag{5}$$

*Then the same relation between the work function* $W$ *and free energy difference* $\Delta F$ *holds:*

$$\mathbb{E}_{\mathbb{P}^{\rightarrow}} \, \mathrm{e}^{-W} = \mathrm{e}^{-\Delta F}, \quad \text{where} \ W(x_{0:M}) := \log \prod_{\ell=0}^{M-1} \frac{f_\ell(x_\ell)}{f_{\ell+1}(x_\ell)} \ \text{and} \ \Delta F := -\log \frac{Z_M}{Z_0}.$$

*Proof.* Since $\pi_\ell$ is invariant for $F_\ell$, the following backward transition kernels are well-defined:

$$B_\ell(x, x') = \frac{\pi_\ell(x')}{\pi_\ell(x)} F_\ell(x', x), \ \ell \in [\![1, M]\!] \, .$$

By applying these backward transition kernels sequentially, we define the backward path measure

$$\mathbb{P}^{\leftarrow}(x_{0:M}) = \pi_M(x_M) \prod_{\ell=1}^{M} B_\ell(x_\ell, x_{\ell-1}). \tag{6}$$

It can be easily demonstrated, as in (4), that $\log \frac{\mathrm{d}\mathbb{P}^{\rightarrow}}{\mathrm{d}\mathbb{P}^{\leftarrow}}(x_{0:M}) = W(x_{0:M}) - \Delta F$. Consequently, the identity $\mathbb{E}_{\mathbb{P}^{\rightarrow}} \frac{\mathrm{d}\mathbb{P}^{\leftarrow}}{\mathrm{d}\mathbb{P}^{\rightarrow}} = 1$ implies the desired equality. $\qquad\square$

While the frameworks of JE and AIS hold for *general* curves of interpolation, for the study of non-asymptotic complexity guarantees, we focus on a widely used curve in theoretical analysis (Brosse et al., 2018; Ge et al., 2020), which we refer to as the **geometric interpolation**:[2]

$$\pi_\theta = \frac{1}{Z_\theta} f_\theta = \frac{1}{Z_\theta} \exp\left(-V - \frac{\lambda(\theta)}{2}\|\cdot\|^2\right), \ \theta \in [0, 1], \tag{7}$$

where $\lambda(\cdot)$ is a decreasing function with $\lambda(0) = 2\beta$ and $\lambda(1) = 0$, referred to as the *annealing schedule*. With this choice of $\lambda(0)$, by Assump. 2, the potential of $\pi_0$ is $\beta$-strongly-convex and $3\beta$-smooth, making sampling and normalizing constant estimation relatively easy. To estimate $Z_0$, we use the thermodynamic integration (TI) algorithm from Ge et al. (2020), which requires $\widetilde{O}\left(\frac{d^{4/3}}{\varepsilon^2}\right)$ gradient oracle calls. In a nutshell, TI is an equilibrium method that constructs a series of intermediate distributions and estimates adjacent normalizing constant ratios via expectation under these intermediate distributions, realized through MCMC sampling from each intermediate distribution. As TI is peripheral to our primary focus, we defer its full description, including the choice of hyperparameters and complexity bound, to App. H.1.

---

[2](7) differs slightly from a widely used curve in applications (Gelman & Meng, 1998; Neal, 2001): $\pi_\theta \propto \pi^{1-\lambda(\theta)} \phi^{\lambda(\theta)}$, where $\phi$ is a prior distribution (typically Gaussian). We refer to both as *geometric interpolation*.

Given (7), we introduce time points $0 = \theta_0 < \theta_1 < ... < \theta_M = 1$ to be specified later, and adopt the framework outlined in Thm. 3 by setting $\pi_\ell = \frac{f_\ell}{Z_\ell}$ to correspond to $\pi_{\theta_\ell} = \frac{f_{\theta_\ell}}{Z_{\theta_\ell}}$, albeit with a slight abuse of notation. To estimate the normalizing constant, we need to sample from the forward path measure $\mathbb{P}^{\rightarrow}$ and compute the work function along the trajectory. Since $\pi_{\theta_\ell}$ must be an invariant distribution of the transition kernel $F_\ell$ in $\mathbb{P}^{\rightarrow}$, we define $F_\ell$ via running LD targeting $\pi_{\theta_\ell}$ for a short time $T_\ell$, i.e., $F_\ell(x, \cdot)$ is given by the law of $X_{T_\ell}$ in the following SDE initialized at $X_0 = x$:

$$\mathrm{d}X_t = \nabla \log \pi_{\theta_\ell}(X_t)\mathrm{d}t + \sqrt{2}\mathrm{d}B_t, \ t \in [0, T_\ell]. \tag{8}$$

In this setting, AIS can be interpreted as a discretization of JE (Lelièvre et al., 2010, Rmk. 4.5). However, in practice, exact samples from $\pi_0$ are often unavailable, and the simulation of LD cannot be performed perfectly.[3] To capture these considerations, we define the following path measure:

$$\widehat{\mathbb{P}}^{\rightarrow}(x_{0:M}) = \widehat{\pi}_0(x_0) \prod_{\ell=1}^{M} \widehat{F}_\ell(x_{\ell-1}, x_\ell), \tag{9}$$

where $\widehat{\pi}_0$ is the law of an approximate sample from $\pi_0$, and the transition kernel $\widehat{F}_\ell$ is a discretization of the LD in $F_\ell$, defined as running *one step* of **annealed Langevin Monte Carlo (ALMC)** using the exponential integrator discretization scheme (Zhang & Chen, 2023; Zhang et al., 2023b;a) with step size $T_\ell$. Formally, $\widehat{F}_\ell(x, \cdot)$ is the law of $X_{T_\ell}$ in the following SDE initialized at $X_0 = x$:

$$\mathrm{d}X_t = -\left(\nabla V(X_0) + \lambda\left(\theta_{\ell-1} + \frac{t}{T_\ell}(\theta_\ell - \theta_{\ell-1})\right)X_t\right)\mathrm{d}t + \sqrt{2}\mathrm{d}B_t, \ t \in [0, T_\ell]. \tag{10}$$

Here, instead of simply setting $\widehat{F}_\ell$ as one step of LMC targeting $\pi_{\theta_\ell}$, the dynamically changing $\lambda(\cdot)$ helps reduce the discretization error, as will be shown in our proof. Furthermore, with a sufficiently small step size, the overall discretization error can also be minimized, motivating us to apply just one update step in each transition kernel.

We refer readers to Alg. 1 in App. B for a summary of the detailed implementation of our proposed AIS algorithm, including the TI procedure and the update rules in (10). The following theorem delineates the oracle complexity of the algorithm required to obtain an estimate $\widehat{Z}$ meeting the desired accuracy criterion (1), whose detailed proof can be located in App. D. The required values of hyperparameters $M$, and $T_\ell$ can be found at the end of the proof.

**Theorem 4.** *Let $\widehat{Z}$ be the AIS estimator described as in Alg. 1, i.e., $\widehat{Z} := \widehat{Z}_0 \mathrm{e}^{-W(x_{0:M})}$ where $\widehat{Z}_0$ is estimated by TI and $x_{0:M} \sim \widehat{\mathbb{P}}^{\rightarrow}$. Under Assumps. 1 and 2, consider the annealing schedule $\lambda(\theta) = 2\beta(1-\theta)^r$ for some $1 \leq r \lesssim 1$. Use $\mathcal{A}_r$ to denote the action of $(\pi_\theta)_{\theta \in [0,1]}$ to emphasize the dependence on $r$. Then, the oracle complexity for obtaining an estimate $\widehat{Z}$ that satisfies (1) is*

$$\widetilde{O}\left(\frac{d^{\frac{4}{3}}}{\varepsilon^2} \vee \frac{m\beta\mathcal{A}_r^{\frac{1}{2}}}{\varepsilon^2} \vee \frac{d\beta^2\mathcal{A}_r^2}{\varepsilon^4}\right). \tag{11}$$

We present a high-level proof sketch using Fig. 1. The continuous dynamics, comprising the forward path $\mathbb{P}^{\rightarrow}$, the backward path $\mathbb{P}^{\leftarrow}$, and the reference path $\mathbb{P}$, are depicted as three black curves. To address discretization error, the $\ell$-th red (purple) arrow proceeding from left to right represents the transition kernel $\widehat{F}_\ell$ ($B_\ell$), whose composition forms $\widehat{\mathbb{P}}^{\rightarrow}$ ($\mathbb{P}^{\leftarrow}$).

**(I)** Analogously to the analysis of JE (Thm. 2), define the reference path measure $\mathbb{P}$ with transition kernels $F_\ell^*$ such that $x_\ell \sim \pi_{\theta_\ell}$. Given the sampling path measure $\widehat{\mathbb{P}}^{\rightarrow}$, define $\overline{\mathbb{P}}^{\rightarrow}$ as the version of $\widehat{\mathbb{P}}^{\rightarrow}$ without the initialization error, i.e., by replacing $\widehat{\pi}_0$ with $\pi_0$ in (9).

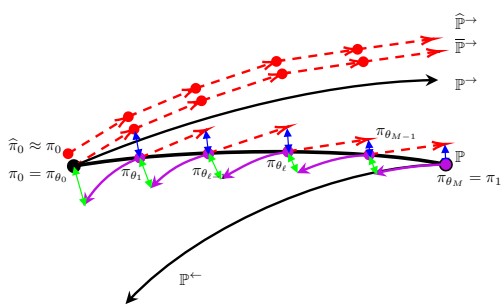

Figure 1: Illustration of the proof idea for Thm. 4.

---

[3]Recall that we need $F_\ell$ to have invariant distribution $\pi_\ell$ in Thm. 3.

**(II)** Show that it suffices to obtain an accurate estimate $\widehat{Z}_0$ and initialization distribution $\widehat{\pi}_0$, together with sufficiently small KL divergences $\text{KL}(\mathbb{P}\|\mathbb{P}^{\leftarrow})$ and $\text{KL}(\mathbb{P}\|\overline{\mathbb{P}}^{\rightarrow})$, which quantify the closeness between the continuous dynamics and the discretization error in implementation, respectively.

**(III)** Using the chain rule, decompose $\text{KL}(\mathbb{P}\|\mathbb{P}^{\leftarrow})$ into the sum of KL divergences between each pair of transition kernels $F_\ell$ and $F_\ell^*$ (i.e., green "distances"). As in Thm. 2, $F_\ell^*$, a transition kernel from $\pi_{\theta_{\ell-1}}$ to $\pi_{\theta_\ell}$, is realized by ALD with a compensatory vector field, ensuring the SDE exactly follows the trajectory $(\pi_\theta)_{\theta \in [\theta_{\ell-1}, \theta_\ell]}$. Similarly, by applying the chain rule and Girsanov's theorem, we can express $\text{KL}(\mathbb{P}\|\overline{\mathbb{P}}^{\rightarrow})$ as the sum of the blue "distances", allowing for a similar analysis.

**(IV)** Finally, derive three necessary conditions on the time steps $\theta_\ell$ to control both $\text{KL}(\mathbb{P}\|\mathbb{P}^{\leftarrow})$ and $\text{KL}(\mathbb{P}\|\overline{\mathbb{P}}^{\rightarrow})$. Choosing a proper schedule yields the desired complexity bound.

Our proposed algorithm consists of two phases: first, estimating $Z_0$ by TI, which is provably efficient for well-conditioned distributions, and second, estimating $Z$ by AIS, which is better suited for handling non-log-concave distributions. The three terms in (11) arise from (i) ensuring the accuracy of $\widehat{Z}_0$, (ii) controlling $\text{KL}(\mathbb{P}\|\mathbb{P}^{\leftarrow})$, and (iii) controlling $\text{KL}(\mathbb{P}\|\overline{\mathbb{P}}^{\rightarrow})$, respectively, as discussed in **(II)** above. Due to the non-log-concavity of $\pi$, the action $\mathcal{A}$ is typically large, making (iii), the cost for controlling the discretization error, the dominant complexity. The $\varepsilon$-dependence can be interpreted as the total duration $T = \Theta\left(\frac{1}{\varepsilon^2}\right)$ required for the continuous dynamics to converge (as in Thm. 2) divided by the step size $\widetilde{\Theta}(\varepsilon^2)$ to control the discretization error. Finally, we remark that although Thm. 4 is only proved for geometric interpolation, the proof strategy can be generalized to *any* curve of distributions $(\pi_\theta)_{\theta \in [0,1]}$ satisfying weak regularity conditions, and possibly with an additional compensatory drift term, as long as we know the expression of the score $\nabla \log \pi_\theta$, and also the Lipschitz constants of the score and the drift term for controlling the discretization error.

## 5 NORMALIZING CONSTANT ESTIMATION VIA DENOISING DIFFUSION

**Disadvantage of Geometric Interpolation.** From the analysis of JE and AIS (Thms. 2 and 4), the choice of the interpolation curve $(\pi_\theta)_{\theta \in [0,1]}$ is crucial for the complexity. The geometric interpolation (7) is widely adopted due to the availability of closed-form scores of the intermediate distributions $\pi_\theta$, and for certain structured non-log-concave distributions, the associated action is polynomially large, enabling efficient AIS. For instance, Guo et al. (2025, Ex. 2) analyzed a Gaussian mixture target distribution with identical covariance, means having the *same* norm, and arbitrary weights. However, for general target distributions, the action of the related curve can grow prohibitively large. We now establish an exponential lower bound on the action of a curve starting from a Gaussian mixture, highlighting the potential inefficiency of AIS under geometric interpolation.

**Proposition 1.** *Consider the Gaussian mixture target distribution $\pi = \frac{1}{2}\mathcal{N}(0,1) + \frac{1}{2}\mathcal{N}(m,1)$ on $\mathbb{R}$ for some sufficiently large $m \gtrsim 1$, whose potential is $\frac{m^2}{2}$-smooth. Under the setting in AIS (Thm. 4), define $\pi_\theta(x) \propto \pi(x)\mathrm{e}^{-\frac{\lambda(\theta)x^2}{2}}$, $\theta \in [0,1]$, where $\lambda(\theta) = m^2(1-\theta)^r$ for some $r \geq 1$. Then, the action of the curve $(\pi_\theta)_{\theta \in [0,1]}$, $\mathcal{A}_r$, is lower bounded by $\mathcal{A}_r \gtrsim m^4 \mathrm{e}^{\frac{m^2}{40}}$.*

The full proof is in App. E.1. The key technical tool is a closed-form expression of the $\text{W}_2$ distance in $\mathbb{R}$ expressed by the inverse cumulative distribution functions (c.d.f.s) of the involved distributions, and we lower bound the metric derivative near the target distribution, where the curve changes the most drastically. This observation provides a novel perspective on the quantitative description of the **mass teleportation** or **mode switching** phenomenon (Woodard et al., 2009; Tawn et al., 2020; Syed et al., 2021; Chemseddine et al., 2025), motivating us to explore alternative curves that can potentially yield smaller action, thereby enhancing the efficiency of normalizing constant estimation. Intuitively, during the annealing process, the probability mass needs to be transported from one mode of the distribution to another well-separated mode in a short period of time (e.g., through the change of weights in both modes), which leads to torpid mixing for many samplers.

**Reverse Diffusion Samplers.** Inspired by score-based generative models (Song et al., 2021), recent advancements have led to the development of multimodal samplers based on reversing the Ornstein-Uhlenbeck (OU) process, such as reverse diffusion Monte Carlo (RDMC, Huang et al. (2024a)), recursive score diffusion-based Monte Carlo (RSDMC, Huang et al. (2024b)), zeroth-order diffusion

Monte Carlo (ZODMC, He et al. (2024)), and self-normalized diffusion Monte Carlo[4] (SNDMC, Vacher et al. (2025)). We collectively refer to these methods as the **reverse diffusion samplers (RDS)**. The key idea is to simulate the time reversal of the following OU process, which transforms any target distribution $\pi$ into $\phi := \mathcal{N}(0, I)$ as $T \to \infty$:

$$\mathrm{d}Y_t = -Y_t\mathrm{d}t + \sqrt{2}\mathrm{d}B_t, \ t \in [0, T]; \ Y_0 \sim \pi. \tag{12}$$

Let $Y_t \sim \overline{\pi}_t$. The time-reversal $(Y_t^{\leftarrow} := Y_{T-t} \sim \overline{\pi}_{T-t})_{t \in [0,T]}$ satisfies the SDE

$$\mathrm{d}Y_t^{\leftarrow} = (Y_t^{\leftarrow} + 2\nabla \log \overline{\pi}_{T-t}(Y_t^{\leftarrow}))\mathrm{d}t + \sqrt{2}\mathrm{d}W_t, \ t \in [0, T]; \ Y_0^{\leftarrow} \sim \overline{\pi}_T(\approx \phi). \tag{13}$$

Hence, to draw samples $Y_0^{\leftarrow} \sim \pi$, it suffices to approximate the scores $\nabla \log \overline{\pi}_t$ and discretize (13), which can be implemented in various *learning-free (non-parametric)* ways in the literature of RDS mentioned above. See App. E.5 for a detailed review.

**RDS-based Normalizing Constant Estimation.** We now propose to leverage $(\overline{\pi}_{T-t})_{t \in [0,T]}$ in AIS. To support this idea, we first present the following proposition. We remark that the result can be generalized to a bound on the Wasserstein gradient flow for the KL divergence to *any* target distribution with weak regularity condition, not necessary the standard normal distribution. See the proof in App. E.2 for details.

**Proposition 2.** *Define $\overline{\pi}_t$ as the law of $Y_t$ in the OU process* (12) *initialized from $Y_0 \sim \pi \propto \mathrm{e}^{-V}$, where $V$ is $\beta$-smooth. Let $m^2 := \mathbb{E}_\pi \|\cdot\|^2 < \infty$. Then, $\int_0^\infty |\dot{\overline{\pi}}|_t^2 \mathrm{d}t \leq d\beta + m^2$.*

Prop. 2 shows that under fairly weak conditions on the target distribution, the action of the curve along the OU process, $(\overline{\pi}_{T-t})_{t \in [0,T]}$, behaves much better than (7). Hence, our analysis of JE (Thm. 2) suggests that this curve is likely to yield more efficient normalizing constant estimation. Furthermore, recall that in our earlier proof, we introduced a compensatory drift term $v_t$ to eliminate the lag in JE. The same principle applies here: ensuring $X_t$ precisely following the reference trajectory is advantageous, which results in the time-reversal of OU process (13). Building on this insight, we propose an RDS-based algorithm for normalizing constant estimation, and establish a framework for analyzing its oracle complexity. See App. E.3 for the proof.

**Theorem 5.** *Assume a total time duration $T$, an early stopping time $\delta \geq 0$, and discrete time points $0 = t_0 < t_1 < ... < t_N = T - \delta \leq T$. For $t \in [0, T - \delta)$, let $t_-$ denote $t_k$ if $t \in [t_k, t_{k+1})$. Let $s. \approx \nabla \log \overline{\pi}.$ be a score estimator, and $\phi = \mathcal{N}(0, I)$. Consider the following two SDEs on $[0, T - \delta]$ representing the sampling trajectory and the time-reversed OU process, respectively:*

$$\mathbb{Q}^{\dagger}: \quad \mathrm{d}X_t = (X_t + 2s_{T-t_-}(X_{t_-}))\mathrm{d}t + \sqrt{2}\mathrm{d}B_t, \qquad\qquad X_0 \sim \phi; \tag{14}$$
$$\mathbb{Q}: \quad \mathrm{d}X_t = (X_t + 2\nabla \log \overline{\pi}_{T-t}(X_t))\mathrm{d}t + \sqrt{2}\mathrm{d}B_t, \qquad X_0 \sim \overline{\pi}_T.$$

*Let $\widehat{Z} := \mathrm{e}^{-W(X)}$, $X \sim \mathbb{Q}^{\dagger}$ be the estimator of $Z$, where the functional $X \mapsto W(X)$ is defined as*

$$\log \phi(X_0) + V(X_{T-\delta}) - (T - \delta)d + \int_0^{T-\delta} \left( \|s_{T-t_-}(X_{t_-})\|^2 \mathrm{d}t + \sqrt{2} \langle s_{T-t_-}(X_{t_-}), \mathrm{d}B_t \rangle \right).$$

*Then, to ensure $\widehat{Z}$ satisfies* (1), *it suffices that $\mathrm{KL}(\mathbb{Q}\|\mathbb{Q}^{\dagger}) \lesssim \varepsilon^2$ and $\mathrm{TV}(\pi, \overline{\pi}_\delta) \lesssim \varepsilon$.*

For detailed implementation of the update rule in (14) and the computation of $W(X)$, see Alg. 2. To determine the overall complexity, we can leverage existing results for RDS to derive the oracle complexity to achieve $\mathrm{KL}(\mathbb{Q}\|\mathbb{Q}^{\dagger}) \lesssim \varepsilon^2$. When early stopping is needed (i.e., $\delta > 0$), we prove in Lem. 7 that choosing $\delta \asymp \frac{\varepsilon^2}{\beta^2 d^2}$ suffices to ensure $\varepsilon$-closeness in TV distance between $\overline{\pi}_\delta$ and $\pi$, under weak assumptions similar to Assump. 2. For RDMC, RSDMC, ZODMC, and SNDMC, the total complexities are, respectively, $O\left(\mathrm{poly}\left(d, \frac{1}{\zeta}\right) \exp\left(\frac{1}{\varepsilon}\right)^{O(n)}\right)$, $\exp\left(\beta^3 \log^3 \mathrm{poly}\left(\beta, d, m^2, \frac{1}{\zeta}\right)\right)$, $\exp\left(\widetilde{O}(d) \log \beta \log \frac{1}{\varepsilon}\right)$, and $O\left(\left(\frac{\beta(m^2 \vee d)}{\varepsilon}\right)^{O(d)}\right)$.[5] The full analysis can be found in App. E.5.

As discussed, RDS can be viewed as an *optimally compensated* ALD using the OU process as the trajectory. We conclude this section by contrasting these two approaches. On the one hand,

---

[4]This name is introduced by us as the original paper did not provide a name for the proposed algorithm.

[5]In RDMC, one assumes there exists $n, c > 0$ such that $\forall r > 0$, $V + r\|\cdot\|^2$ is convex for all $\|x\| \geq \frac{c}{r^n}$. In RDMC and RSDMC, $\zeta$ is the probability threshold that the estimator may fail to achieve the desired accuracy $\varepsilon$.

analytically-tractable curves such as the geometric interpolation offer closed-form drift terms at all time points, but may exhibit poor action properties (Prop. 1) or bad isoperimetric constants (Chehab et al., 2025), making annealed sampling challenging. On the other hand, alternative curves like the OU process may have better properties in action and isoperimetric constants, but their drift terms, often related to the scores of the intermediate distributions, lack closed-form expressions, and estimating these terms is also non-trivial. This highlights a fundamental trade-off between the complexity of the drift term estimation and the property of the interpolation curve.

**Experiments.** We now compare the performance of the methods of normalizing constant estimation for non-log-concave distributions that have been discussed in the paper, including TI, AIS, and the four RDS-based methods. We consider two multimodal target distributions: a modified Müller Brown (MMB) distribution and Gaussian mixture (GM) with $4$ components, both in $\mathbb{R}^2$. The quantitative results are summarized in Tab. 1, where we report the relative error of $\widehat{Z}$ and, for GM, the maximum mean discrepancy (MMD) and $\mathrm{W}_2$ distance between the generated samples $\widehat{\pi}_{\mathrm{samp}}$ and ground truth samples from $\pi$. All RDS-based methods provide accurate estimates of the normalizing constant and high quality samples, while TI and AIS (based on geometric annealing) produce seriously biased estimates due to lack of mode coverage. Further details are presented in App. I, and the codes for the experiments are available at `https://github.com/AlexandreGUO2001/NormConstEst`.

Table 1: Quantitative results of normalizing constant estimation (mean $\pm$ std), best in **bold**.

| Target | Metric | TI | AIS | RDMC | RSDMC | ZODMC | SNDMC |
|---|---|---|---|---|---|---|---|
| MMB | $\widehat{Z}/Z$ | $0.7527 \pm 0.0086$ | $2.9740 \pm 7.6705$ | $0.9829 \pm 0.2116$ | $1.2885 \pm 12.7843$ | $0.9878 \pm 0.1154$ | $\mathbf{1.0053 \pm 0.1192}$ |
| GM | $\widehat{Z}/Z$ | $0.2427 \pm 0.0016$ | $0.2042 \pm 0.0008$ | $\mathbf{1.0001 \pm 0.0850}$ | $0.9202 \pm 1.0276$ | $0.9766 \pm 0.2835$ | $0.9973 \pm 0.0834$ |
| | $\mathrm{MMD}(\widehat{\pi}_{\mathrm{samp}}, \pi)$ | $2.5407 \pm 0.0281$ | $2.4618 \pm 0.0270$ | $0.3581 \pm 0.0366$ | $0.3124 \pm 0.0395$ | $0.2591 \pm 0.0381$ | $\mathbf{0.1576 \pm 0.0279}$ |
| | $\mathrm{W}_2(\widehat{\pi}_{\mathrm{samp}}, \pi)$ | $10.5602 \pm 0.0794$ | $10.4842 \pm 0.0851$ | $7.0242 \pm 0.9104$ | $2.6012 \pm 0.2482$ | $2.4506 \pm 0.2963$ | $\mathbf{1.5494 \pm 0.6820}$ |

## 6 CONCLUSION, LIMITATIONS, AND FUTURE DIRECTIONS

This paper investigates the complexity of normalizing constant estimation using JE, AIS, and RDS, and takes a first step in establishing non-asymptotic convergence guarantees based on insights from continuous-time analysis. Our analysis of JE (Thm. 2) applies to general interpolations without explicit dependence of isoperimetry, thereby substantially extending prior work limited to log-concave distributions. Several limitations remain: the tightness of our upper bounds (Thms. 2 and 4) are unknown; the lower bound on the action in Prop. 1 does not directly imply that JE needs exponentially long time to converge; though the action provides a clean analysis of the statistical efficiency of annealing–which isoperimetric inequalities cannot deal with–its practical interpretability is not well understood. Finally, we conjecture that our proof techniques can extend to samplers beyond overdamped LD (e.g., Hamiltonian or underdamped LD (Sohl-Dickstein & Culpepper, 2012), parallel samplers (Anari et al., 2024; Zhou et al., 2025; Zhou & Sugiyama, 2025a;b)), and may also apply to estimating normalizing constants of compactly supported distributions on $\mathbb{R}^d$ (e.g., convex bodies volume estimation (Cousins & Vempala, 2018)) or discrete distributions (e.g., Ising model and restricted Boltzmann machines (Huber, 2015; Krause et al., 2020)) via the Poisson stochastic integral framework (Ren et al., 2025a;b), which we leave as a direction for future research.

### ACKNOWLEDGMENTS

WG thanks Omar Chehab and Huanjian Zhou for insightful discussions. We also thank Jannis Chemseddine for pointing out the paper Chemseddine et al. (2025) after we initially submitted our work to arXiv. WG and YC acknowledge supports from NSF Grants ECCS-1942523, DMS-2206576, and AFOSR Grant FA9550-25-1-0169. MT is thankful for partial supports by NSF Grants DMS-1847802, DMS-2513699, DOE Grants NA0004261, SC0026274, and Richard Duke Fellowship.

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

CONTENTS

# A PRELIMINARIES

## A.1 STOCHASTIC ANALYSIS: FORWARD-BACKWARD SDEs AND GIRSANOV'S THEOREM

For a stochastic differential equation (SDE) $X = (X_t)_{t \in [0,T]}$ defined on $\Omega = C([0,T]; \mathbb{R}^d)$, the distribution of $X$ over $\Omega$ is called the **path measure** of $X$, defined by $\mathbb{P}^X$: measurable $A \subset \Omega \mapsto \Pr(X \in A)$. The following lemma, as a corollary of the **Girsanov's theorem** (Üstünel & Zakai, 2013, Prop. 2.3.1 & Cor. 2.3.1), provides a method for computing the Radon-Nikodým (RN) derivative and KL divergence between two path measures, which serves as a key technical tool in our proof.

**Lemma 1.** *Assume we have the following two SDEs with $t \in [0,T]$:*

$$\mathrm{d}X_t = a_t(X_t)\mathrm{d}t + \sigma \mathrm{d}B_t, \ X_0 \sim \mu; \quad \mathrm{d}Y_t = b_t(Y_t)\mathrm{d}t + \sigma \mathrm{d}B_t, \ Y_0 \sim \nu.$$

*Denote the path measures of $X$ and $Y$ as $\mathbb{P}^X$ and $\mathbb{P}^Y$, respectively. Then for any trajectory $\xi \in \Omega$,*

$$\log \frac{\mathrm{d}\mathbb{P}^X}{\mathrm{d}\mathbb{P}^Y}(\xi) = \log \frac{\mathrm{d}\mu}{\mathrm{d}\nu}(\xi_0) + \frac{1}{\sigma^2} \int_0^T \langle a_t(\xi_t) - b_t(\xi_t), \mathrm{d}\xi_t \rangle - \frac{1}{2\sigma^2} \int_0^T (\|a_t(\xi_t)\|^2 - \|b_t(\xi_t)\|^2)\mathrm{d}t.$$

*In particular, plugging in $\xi \leftarrow X \sim \mathbb{P}^X$, we can compute the KL divergence:*

$$\mathrm{KL}(\mathbb{P}^X \| \mathbb{P}^Y) = \mathrm{KL}(\mu \| \nu) + \frac{1}{2\sigma^2} \int_0^T \mathbb{E}_{\mathbb{P}^X} \|a_t(X_t) - b_t(X_t)\|^2 \mathrm{d}t.$$

**Remark 3.** *The Girsanov's theorem requires a technical condition ensuring that a local martingale is a true martingale, typically verified via the Novikov condition (Karatzas & Shreve, 1991, Chap. 3, Cor. 5.13), which can be challenging to establish. However, when only an upper bound of the KL divergence is needed, the approximation argument from Chen et al. (2023, App. B.2) circumvents the verification of the Novikov condition. For additional context, see Chewi (2022, Sec. 3.2). In this paper, we omit these technical details and always assume that the Novikov condition holds.*

We now present the theory of backward stochastic integral and the Girsanov's theorem, which are adapted from Vargas et al. (2024). Here, we include relevant results and proofs to ensure a self-contained presentation.

The backward SDE can be perceived as the time-reversal of a forward SDE:

**Definition 1** (Backward SDE). *Given a BM $(B_t)_{t \in [0,T]}$, let its time-reversal be $(B_t^{\leftarrow} := B_{T-t})_{t \in [0,T]}$. We say that a process $(X_t^{\leftarrow})_{t \in [0,T]}$ satisfies the **backward SDE***

$$\mathrm{d}X_t^{\leftarrow} = a_t(X_t^{\leftarrow})\mathrm{d}t + \sigma \mathrm{d}B_t^{\leftarrow}, \ t \in [0,T]; \ X_T^{\leftarrow} \sim \nu$$

*if its time-reversal $(X_t = X_{T-t}^{\leftarrow})_{t \in [0,T]}$ satisfies the following forward SDE:*

$$\mathrm{d}X_t = -a_{T-t}(X_t)\mathrm{d}t + \sigma \mathrm{d}B_t, \ t \in [0,T]; \ X_0 \sim \nu.$$

**Remark 4.** *Intuitively, one can understand the backward SDE through the following Euler-Maruyama discretization: with $\Delta t > 0$:*

$$X_{t-\Delta t}^{\leftarrow} \approx X_t^{\leftarrow} + a_t(X_t^{\leftarrow})(-\Delta t) + \sigma(B_{t-\Delta t}^{\leftarrow} - B_t^{\leftarrow})$$
$$\Longleftrightarrow X_{T-t+\Delta t} \approx X_{T-t} - a_t(X_{T-t})\Delta t + \sigma(B_{T-t+\Delta t} - B_{T-t}).$$

*where $B_{t-\Delta t}^{\leftarrow} - B_t^{\leftarrow} \sim \mathcal{N}(0, \Delta t I)$ and is independent of $(X_s^{\leftarrow})_{s \in [t,T]}$.*

The forward and backward SDEs are related through the following Nelson's relation:

**Lemma 2** (Nelson's relation (Nelson, 1967; Anderson, 1982)). *Given a BM $(B_t)_{t \in [0,T]}$ and its time-reversal $(B_t^{\leftarrow} = B_{T-t})_{t \in [0,T]}$, the following two SDEs*

$$\mathrm{d}X_t = a_t(X_t)\mathrm{d}t + \sigma \mathrm{d}B_t, \ X_0 \sim p_0; \quad and \quad \mathrm{d}Y_t = b_t(Y_t)\mathrm{d}t + \sigma \mathrm{d}B_t^{\leftarrow}, \ Y_T \sim q$$

*have the same path measure if and only if*

$$q = p_T, \quad and \quad b_t = a_t - \sigma^2 \nabla \log p_t, \ \forall t \in [0,T],$$

*where $p_t$ is the p.d.f. of $X_t$.*

*Proof.* The proof is by verifying the Fokker-Planck equation. For $X$, we have

$$\partial_t p_t = -\nabla \cdot (a_t p_t) + \frac{\sigma^2}{2} \Delta p_t.$$

Let $\star_t^\leftarrow := \star_{T-t}$. Then $p_t^\leftarrow$ satisfies

$$\partial_t p_t^\leftarrow = \nabla \cdot (a_t^\leftarrow p_t^\leftarrow) - \frac{\sigma^2}{2} \Delta p_t^\leftarrow = -\nabla \cdot ((-a_t^\leftarrow + \sigma^2 \nabla \log p_t^\leftarrow) p_t^\leftarrow) + \frac{\sigma^2}{2} \Delta p_t^\leftarrow,$$

which means $(X_t^\leftarrow)_{t \in [0,T]}$ has the same path measure as the following SDE:

$$\mathrm{d}Z_t = -(a_t^\leftarrow - \sigma^2 \nabla \log p_t^\leftarrow)(Z_t)\mathrm{d}t + \sigma \mathrm{d}B_t, \ Z_t \sim p_t^\leftarrow.$$

On the other hand, by definition, $(Y_t^\leftarrow)_{t \in [0,T]}$ satisfies the forward SDE

$$\mathrm{d}Y_t^\leftarrow = -b_t^\leftarrow(Y_t^\leftarrow)\mathrm{d}t + \sigma \mathrm{d}B_t, \ Y_0 \sim q,$$

and thus the claim is evident. $\qquad\square$

We now introduce the concept of backward stochastic integral, which allows us to represent the RN derivative between path measures of forward and backward SDEs.

**Definition 2** (Backward stochastic integral). *For two continuous stochastic processes $X$ and $Y$ on $C([0,T]; \mathbb{R}^d)$, the **backward stochastic integral** of $Y$ with respect to $X$ is defined as*

$$\int_0^T \langle Y_t, *\mathrm{d}X_t \rangle := \mathrm{Pr} \text{-} \lim_{\|\Pi\| \to 0} \sum_{i=0}^{n-1} \langle Y_{t_{i+1}}, X_{t_{i+1}} - X_{t_i} \rangle,$$

*where $\Pi = \{0 = t_0 < t_1 < ... < t_n = T\}$ is a partition of $[0,T]$, $\|\Pi\| := \max_{i \in [\![1,n]\!]} (t_{i+1} - t_i)$, and the convergence is in the probability sense. When both $X$ and $Y$ are continuous semi-martingales, one can equivalently define*

$$\int_0^T \langle Y_t, *\mathrm{d}X_t \rangle := \int_0^T \langle Y_t, \mathrm{d}X_t \rangle + [X, Y]_T, \tag{15}$$

*where $[X,Y].$ is the cross quadratic variation process[6] of the local martingale parts of $X$ and $Y$.*

**Remark 5.** *Although rarely used in practice, the backward stochastic integral is sometimes referred to as the Hänggi-Klimontovich integral in the literature. Recall that the Itô integral is defined as the limit of Riemann sums when the leftmost point of each interval is used, while the Stratonovich integral is based on the midpoint and the backward integral uses the rightmost point. The equivalence in (15) can be justified in Karatzas & Shreve (1991, Chap. 3.3).*

**Lemma 3** (Continuation of Lem. 1). *1. If we replace the SDEs in Lem. 1 with*

$$\mathrm{d}X_t = a_t(X_t)\mathrm{d}t + \sigma \mathrm{d}B_t^\leftarrow, \ X_T \sim \mu; \qquad \mathrm{d}Y_t = b_t(Y_t)\mathrm{d}t + \sigma \mathrm{d}B_t^\leftarrow, \ Y_T \sim \nu,$$

*while keeping other assumptions and notations unchanged, then for any trajectory $\xi \in \Omega$,*

$$\log \frac{\mathrm{d}\mathbb{P}^X}{\mathrm{d}\mathbb{P}^Y}(\xi) = \log \frac{\mathrm{d}\mu}{\mathrm{d}\nu}(\xi_T) + \frac{1}{\sigma^2} \int_0^T \langle a_t(\xi_t) - b_t(\xi_t), *\mathrm{d}\xi_t \rangle - \frac{1}{2\sigma^2} \int_0^T (\|a_t(\xi_t)\|^2 - \|b_t(\xi_t)\|^2)\mathrm{d}t,$$

*and consequently,*

$$\mathrm{KL}(\mathbb{P}^X \| \mathbb{P}^Y) = \mathrm{KL}(\mu \| \nu) + \frac{1}{2\sigma^2} \int_0^T \mathbb{E}_{\mathbb{P}^X} \|a_t(X_t) - b_t(X_t)\|^2 \mathrm{d}t.$$

*2. Define the following two SDEs from $0$ to $T$:*

$$\mathrm{d}X_t = a_t(X_t)\mathrm{d}t + \sigma \mathrm{d}B_t, \ X_0 \sim \mu; \qquad \mathrm{d}Y_t = b_t(Y_t)\mathrm{d}t + \sigma \mathrm{d}B_t^\leftarrow, \ Y_T \sim \nu.$$

*Denote the path measures of $X$ and $Y$ as $\mathbb{P}^X$ and $\mathbb{P}^Y$, respectively. Then for any trajectory $\xi \in \Omega$,*

$$\log \frac{\mathrm{d}\mathbb{P}^X}{\mathrm{d}\mathbb{P}^Y}(\xi) = \log \frac{\mu(\xi_0)}{\nu(\xi_T)} + \frac{1}{\sigma^2} \int_0^T (\langle a_t(\xi_t), \mathrm{d}\xi_t \rangle - \langle b_t(\xi_t), *\mathrm{d}\xi_t \rangle) - \frac{1}{2\sigma^2} \int_0^T (\|a_t(\xi_t)\|^2 - \|b_t(\xi_t)\|^2)\mathrm{d}t.$$

---

[6]The notation used in Karatzas & Shreve (1991) is $\langle \cdot, \cdot \rangle.$. We use square brackets here to avoid conflict with the notation for inner product.

*Proof.* **1.** Only in the proof of this theorem, we use the notation $\star_t^{\leftarrow} := \star_{T-t}$ to represent the time reversal. We know that

$$\mathrm{d}X_t^{\leftarrow} = -a_t^{\leftarrow}(X_t^{\leftarrow})\mathrm{d}t + \sigma \mathrm{d}B_t, \ X_0^{\leftarrow} \sim \mu; \qquad \mathrm{d}Y_t^{\leftarrow} = -b_t^{\leftarrow}(Y_t^{\leftarrow})\mathrm{d}t + \sigma \mathrm{d}B_t, \ Y_0^{\leftarrow} \sim \nu.$$

Let $\mathbb{P}^{X^{\leftarrow}}$ and $\mathbb{P}^{Y^{\leftarrow}}$ be the path measures of $X^{\leftarrow}$ and $Y^{\leftarrow}$, respectively. From Lem. 1, we know that

$$\log \frac{\mathrm{d}\mathbb{P}^{X^{\leftarrow}}}{\mathrm{d}\mathbb{P}^{Y^{\leftarrow}}}(\xi) = \log \frac{\mathrm{d}\mu}{\mathrm{d}\nu}(\xi_0) - \frac{1}{\sigma^2}\int_0^T \langle a_t^{\leftarrow}(\xi_t) - b_t^{\leftarrow}(\xi_t), \mathrm{d}\xi_t \rangle - \frac{1}{2\sigma^2}\int_0^T (\|a_t^{\leftarrow}(\xi_t)\|^2 - \|b_t^{\leftarrow}(\xi_t)\|^2)\mathrm{d}t.$$

Since $\mathbb{P}^{X^{\leftarrow}}(\mathrm{d}\xi) = \Pr(X^{\leftarrow} \in \mathrm{d}\xi) = \Pr(X \in \mathrm{d}\xi^{\leftarrow}) = \mathbb{P}^X(\mathrm{d}\xi^{\leftarrow})$, we obtain

$$\log \frac{\mathrm{d}\mathbb{P}^X}{\mathrm{d}\mathbb{P}^Y}(\xi) = \log \frac{\mathrm{d}\mathbb{P}^{X^{\leftarrow}}}{\mathrm{d}\mathbb{P}^{Y^{\leftarrow}}}(\xi^{\leftarrow})$$

$$= \log \frac{\mathrm{d}\mu}{\mathrm{d}\nu}(\xi_0^{\leftarrow}) - \frac{1}{\sigma^2}\int_0^T \langle a_t^{\leftarrow}(\xi_t^{\leftarrow}) - b_t^{\leftarrow}(\xi_t^{\leftarrow}), \mathrm{d}\xi_t^{\leftarrow} \rangle - \frac{1}{2\sigma^2}\int_0^T (\|a_t^{\leftarrow}(\xi_t^{\leftarrow})\|^2 - \|b_t^{\leftarrow}(\xi_t^{\leftarrow})\|^2)\mathrm{d}t$$

$$= \log \frac{\mathrm{d}\mu}{\mathrm{d}\nu}(\xi_T) + \frac{1}{\sigma^2}\int_0^T \langle a_t(\xi_t) - b_t(\xi_t), *\mathrm{d}\xi_t \rangle - \frac{1}{2\sigma^2}\int_0^T (\|a_t(\xi_t)\|^2 - \|b_t(\xi_t)\|^2)\mathrm{d}t.$$

To justify the last equality, if $\xi, \eta$ are two continuous stochastic processes, then by definition,

$$\int_0^T \langle \xi_t^{\leftarrow}, \mathrm{d}\eta_t^{\leftarrow} \rangle = \text{Pr -} \lim_{\|\Pi\| \to 0} \sum_{i=0}^{n-1} \left\langle \xi_{t_{i-1}}^{\leftarrow}, \eta_{t_i}^{\leftarrow} - \eta_{t_{i-1}}^{\leftarrow} \right\rangle$$

$$= \text{Pr -} \lim_{\|\Pi\| \to 0} \sum_{i=0}^{n-1} \left\langle \xi_{T-t_{i-1}}, \eta_{T-t_i} - \eta_{T-t_{i-1}} \right\rangle$$

$$= \text{Pr -} \lim_{\|\Pi\| \to 0} -\sum_{i=0}^{n-1} \left\langle \xi_{T-t_{i-1}}, \eta_{T-t_{i-1}} - \eta_{T-t_i} \right\rangle$$

$$= -\int_0^T \langle \xi_t, *\mathrm{d}\eta_t \rangle. \tag{16}$$

On the other hand,

$$\int_0^T \xi_t^{\leftarrow} \mathrm{d}t = \int_0^T \xi_{T-t} \mathrm{d}t = \int_0^T \xi_t \mathrm{d}t.$$

Therefore, the equality of RN derivative holds. Plugging in $\xi \leftarrow X$, we have

$$\log \frac{\mathrm{d}\mathbb{P}^X}{\mathrm{d}\mathbb{P}^Y}(X) = \log \frac{\mathrm{d}\mu}{\mathrm{d}\nu}(X_T) + \frac{1}{\sigma}\int_0^T \langle a_t(X_t) - b_t(X_t), *\mathrm{d}B_t^{\leftarrow} \rangle + \frac{1}{2\sigma^2}\int_0^T \|a_t(X_t) - b_t(X_t)\|^2 \mathrm{d}t.$$

To obtain the KL divergence, it suffices to show the expectation of the second term is zero. Let

$$M_t := \int_t^T \langle a_r(X_r) - b_r(X_r), *\mathrm{d}B_r^{\leftarrow} \rangle, \ t \in [0, T].$$

By (16), we have

$$M_t^{\leftarrow} = -\int_0^t \langle a_r^{\leftarrow}(X_r^{\leftarrow}) - b_r^{\leftarrow}(X_r^{\leftarrow}), \mathrm{d}B_r \rangle.$$

Since $\mathrm{d}X_t^{\leftarrow} = -a_t^{\leftarrow}(X_t^{\leftarrow})\mathrm{d}t + \sigma \mathrm{d}B_t$, we conclude that $M_t^{\leftarrow}$ is a (forward) martingale, and thus $M$ is a *backward* martingale with $\mathbb{E} M_t = \mathbb{E} M_{T-t}^{\leftarrow} = 0$.
**2.** We present a formal proof by considering the process $\mathrm{d}Z_t = \sigma \mathrm{d}B_t$ and $Z_0 \sim \lambda$, the Lebesgue measure. As a result, formally $Z_t \sim \lambda$ for all $t$, so it can also be written as $\mathrm{d}Z_t = \sigma \mathrm{d}B_t^{\leftarrow}$, $Z_T \sim \lambda$. The result follows by applying Lem. 1 to $X$ and $Z$ and **1.** to $Y$ and $Z$.

$\square$

## A.2 Optimal Transport: Wasserstein Geometry and Metric Derivative

We provide a concise overview of essential concepts in optimal transport (OT) that will be used in the paper. See standard textbooks (Villani, 2003; 2008; Ambrosio et al., 2008; 2021) for details.

For two probability measures $\mu, \nu$ on $\mathbb{R}^d$ with finite second-order moments (i.e., $\mathbb{E}_\mu \| \cdot \|^2, \mathbb{E}_\nu \| \cdot \|^2 < \infty$), the **Wasserstein-2 (W$_2$) distance** between $\mu$ and $\nu$ is defined as $\mathrm{W}_2(\mu, \nu) = \inf_{\gamma \in \Pi(\mu, \nu)} \left( \int \|x - y\|^2 \gamma(\mathrm{d}x, \mathrm{d}y) \right)^{1/2}$, where $\Pi(\mu, \nu)$ is the set of all couplings of $(\mu, \nu)$. The Brenier's theorem states that when $\mu$ has a Lebesgue density, then there exists a unique coupling $(\mathrm{id} \times T_{\mu \to \nu})_\sharp \mu$ that reaches the infimum. Here, $\sharp$ stands for the push-forward of a measure (defined by $T_\sharp \mu(\cdot) = \mu(\{\omega : T(\omega) \in \cdot\})$), and $T_{\mu \to \nu}$ is known as the **OT map** from $\mu$ to $\nu$, which can be written as the gradient of a convex function.

Given a vector field $v = (v_t)_{t \in [a,b]}$ and a curve of probability measures $\rho = (\rho_t)_{t \in [a,b]}$ with finite second-order moment on $\mathbb{R}^d$, we say that $v$ **generates** $\rho$ if the continuity equation $\partial_t \rho_t + \nabla \cdot (\rho_t v_t) = 0$, $t \in [a,b]$ holds in the weak sense. The **metric derivative** of $\rho$ at $t \in [a,b]$ is defined as

$$|\dot\rho|_t := \lim_{\delta \to 0} \frac{\mathrm{W}_2(\rho_{t+\delta}, \rho_t)}{|\delta|},$$

which can be interpreted as the speed of this curve. We say $\rho$ is **absolutely continuous** if $|\dot\rho|_t$ exists and is finite for Lebesgue-a.e. $t \in [a,b]$. The metric derivative and the continuity equation are related through the following fact (Ambrosio et al., 2008, Thm. 8.3.1 & Prop. 8.4.5):

**Lemma 4.** *For an absolutely continuous curve of probability measures $(\rho_t)_{t \in [a,b]}$, any vector field $(v_t)_{t \in [a,b]}$ that generates $(\rho_t)_{t \in [a,b]}$ satisfies $|\dot\rho|_t \leq \|v_t\|_{L^2(\rho_t)}$ for Lebesgue-a.e. $t \in [a,b]$. Moreover, there exists an a.s. unique vector field $(v_t^* \in L^2(\rho_t))_{t \in [a,b]}$ that generates $(\rho_t)_{t \in [a,b]}$ and satisfies $|\dot\rho|_t = \|v_t^*\|_{L^2(\rho_t)}$ for Lebesgue-a.e. $t \in [a,b]$, which is $v_t^* = \lim_{\delta \to 0} \frac{T_{\rho_t \to \rho_{t+\delta}} - \mathrm{id}}{\delta}$.*

Finally, we define the **action** of an absolutely continuous curve of probability measures $(\rho_t)_{t \in [a,b]}$ as $\int_a^b |\dot\rho|_t^2 \mathrm{d}t$, which plays a key role in characterizing the efficiency of a curve for normalizing constant estimation. For basic properties of the action and its relation to isoperimetric inequalities such as log-Sobolev and Poincaré inequalities (see definitions below), we refer the reader to Guo et al. (2025, Lem. 3 & Ex. 1).

**Definition 3** (Isoperimetric inequalities)**.** *A probability measure $\pi$ on $\mathbb{R}^d$ satisfies a **Poincaré inequality (PI)** with constant $C$, or $C$-PI, if for all $f \in C_c^\infty(\mathbb{R}^d)$,*

$$\mathrm{Var}_\pi f \leq C \, \mathbb{E}_\pi \|\nabla f\|^2.$$

*It satisfies a **log-Sobolev inequality (LSI)** with constant $C$, or $C$-LSI, if for all $0 \not\equiv f \in C_c^\infty(\mathbb{R}^d)$,*

$$\mathbb{E}_\pi f^2 \log \frac{f^2}{\mathbb{E}_\pi f^2} \leq 2C \, \mathbb{E}_\pi \|\nabla f\|^2.$$

*Furthermore, $\alpha$-strong-log-concavity implies $\frac{1}{\alpha}$-LSI, and $C$-LSI implies $C$-PI (Bakry et al., 2014).*

## B Pseudo-codes of the Algorithms

See Algs. 1 and 2 for the detailed implementation of the AIS and RDS algorithms, respectively.

## C Proofs for Sec. 3

### C.1 A Complete Proof of Thm. 1

*Proof.* By Girsanov's theorem (Lem. 3), we have

$$\log \frac{\mathrm{d}\mathbb{P}^\rightarrow}{\mathrm{d}\mathbb{P}^\leftarrow}(\xi) = \log \frac{\widetilde{\pi}_0(\xi_0)}{\widetilde{\pi}_T(\xi_T)} + \frac{1}{2} \int_0^T (\langle \nabla \log \widetilde{\pi}_t(\xi_t), \mathrm{d}\xi_t \rangle + \langle \nabla \log \widetilde{\pi}_t(\xi_t), *\mathrm{d}\xi_t \rangle).$$

We first prove the following result (Vargas et al., 2024, Eq. (15)): if $\mathrm{d}x_t = a_t(x_t)\mathrm{d}t + \sqrt{2}\mathrm{d}B_t$, then

$$\int_0^T \langle a_t(x_t), *\mathrm{d}x_t \rangle = \int_0^T \langle a_t(x_t), \mathrm{d}x_t \rangle + 2 \int_0^T \mathrm{tr}\, \nabla a_t(X_t)\mathrm{d}t.$$

---

**Algorithm 1:** Normalizing constant estimation via AIS.

---

**Input:** The target distribution $\pi \propto e^{-V}$, smoothness parameter $\beta$, total time $T$; **TI** number of intermediate distributions $K$, annealing schedule $\lambda_0 > ... > \lambda_K = 0$, number of particles $N$; **AIS** steps $M$, annealing schedule $\lambda(\cdot)$ with $\lambda(0) = 2\beta$, time points $0 = \theta_0 < ... < \theta_M = 1$.

**Output:** $\widehat{Z}$, an estimation of $Z = \int_{\mathbb{R}^d} e^{-V(x)} dx$.

1 // *Phase 1: estimate $Z_0$ via **TI**.*

2 Define $V_0 := V + \beta \|\cdot\|^2$, $\rho_k :\propto \exp\left(-V_0 - \frac{\lambda_k}{2}\|\cdot\|^2\right)$, and $g_k := \exp\left(\frac{\lambda_k - \lambda_{k+1}}{2}\|\cdot\|^2\right)$, for $k \in [\![0, K-1]\!]$;

3 Initialize $\widehat{Z}_0 \leftarrow \exp\left(-V_0(0) + \frac{\|\nabla V_0(0)\|^2}{2(3\beta + \lambda_0)}\right)\left(\frac{2\pi}{3\beta + \lambda_0}\right)^{\frac{d}{2}}$;

4 **for** $k = 0$ **to** $K - 1$ **do**

5      Obtain $N$ i.i.d. approximate samples $x_1^{(k)}, ..., x_N^{(k)}$ from $\rho_k$ (e.g., using LMC or proximal sampler);

6      Update $\widehat{Z}_0 \leftarrow \left(\frac{1}{N}\sum_{n=1}^N g_k(X_n^{(k)})\right)\widehat{Z}_0$;

7 **end**

8 // *Phase 2: estimate $Z$ via **AIS**.*

9 Approximately sample $x_0$ from $\pi_0$ (e.g., using LMC or proximal sampler);

10 Initialize $W \leftarrow -\frac{1}{2}(\lambda(\theta_0) - \lambda(\theta_1))\|x_0\|^2$;

11 **for** $\ell = 1$ **to** $M - 1$ **do**

12      Sample an independent $\xi \sim \mathcal{N}(0, I_d)$;

13      Define $\Lambda(t) := \int_0^t \lambda\left(\theta_{\ell-1} + \frac{\tau}{T_\ell}(\theta_\ell - \theta_{\ell-1})\right)d\tau$, where $T_\ell := T(\theta_\ell - \theta_{\ell-1})$;

14      Update

         $x_\ell \leftarrow e^{-\Lambda(T_\ell)}x_{\ell-1} - \left(\int_0^{T_\ell} e^{-(\Lambda(T_\ell) - \Lambda(t))}dt\right)\nabla V(x_{\ell-1}) + \left(2\int_0^{T_\ell} e^{-2(\Lambda(T_\ell) - \Lambda(t))}dt\right)^{\frac{1}{2}}\xi$;

         // *see Lem. 11 for the derivation.*

15      Update $W \leftarrow W - \frac{1}{2}(\lambda(\theta_\ell) - \lambda(\theta_{\ell+1}))\|x_\ell\|^2$;

16 **end**

17 **return** $\widehat{Z} = \widehat{Z}_0 e^{-W}$

---

*Proof.* Due to (15), it suffices to calculate $[a(X), X]_T$. By Itô's formula, we have

$$da_t(x_t) = (\partial_t a_t(x_t) + \langle \nabla a_t(x_t), a_t(x_t)\rangle + \Delta a_t(x_t))dt + \sqrt{2}\nabla a_t dB_t,$$

and hence

$$[a(X), X]_T = \left[\int_0^{\cdot} \sqrt{2}\nabla a_t(x_t)dB_t, \int_0^{\cdot} \sqrt{2}dB_t\right]_T = \text{tr}\int_0^T 2\nabla a_t(x_t)dt.$$

$\square$

Therefore, for $X \sim \mathbb{P}^{\to}$, we have

$$\log\frac{d\mathbb{P}^{\to}}{d\mathbb{P}^{\leftarrow}}(X) = \log\frac{\widetilde{\pi}_0(X_0)}{\widetilde{\pi}_T(X_T)} + \int_0^T (\langle\nabla\log\widetilde{\pi}_t(X_t), dX_t\rangle + \Delta\log\widetilde{\pi}_t(X_t)dt).$$

On the other hand, by Itô's formula, we have

$$d\log\widetilde{\pi}_t(X_t) = \partial_t\log\widetilde{\pi}_t(X_t) + \langle\nabla\log\widetilde{\pi}_t(X_t), dX_t\rangle + \Delta\log\widetilde{\pi}_t(X_t)dt.$$

Taking the integral, we immediately obtain (4), and the proof is complete. $\square$

### C.2 PROOF OF THM. 2

*Proof.* The proof builds on the techniques developed in Guo et al. (2025, Thm. 1), yet with new components including backward SDE and the corresponding version of the Girsanov's theorem.

---

**Algorithm 2:** Normalizing constant estimation via RDS.

---

**Input:** The target distribution $\pi \propto e^{-V}$, total time duration $T$, early stopping time $\delta \geq 0$, time points $0 = t_0 < t_1 < ... < t_N = T - \delta$; non-parametric score estimator $s_t(\cdot) \approx \nabla \log \overline{\pi}_t(\cdot)$ based on {RDMC, RSDMC, ZODMC, or SNDMC} algorithms.

**Output:** $\widehat{Z}$, an estimation of $Z = \int_{\mathbb{R}^d} e^{-V(x)} \mathrm{d}x$.

1 Sample $X_0 \sim \mathcal{N}(0, I)$, and initialize $W := -\frac{\|X_0\|^2}{2} - \frac{d}{2} \log 2\pi$;

2 **for** $k = 0$ **to** $N - 1$ **do**

3    Sample an independent pair of $\begin{pmatrix} \xi_1 \\ \xi_2 \end{pmatrix} \sim \mathcal{N}\left(0, \begin{pmatrix} 1 & \rho_k \\ \rho_k & 1 \end{pmatrix} \otimes I\right)$, where the correlation is

   $\rho_k = \frac{\sqrt{2}(e^{t_{k+1} - t_k} - 1)}{\sqrt{(e^{2(t_{k+1} - t_k)} - 1)(t_{k+1} - t_k)}}$, and $\otimes$ stands for the Kronecker product;     *// this can be done by sampling* $\xi_1, \widetilde{\xi}_2 \overset{\text{i.i.d.}}{\sim} \mathcal{N}(0, I)$ *and setting* $\xi_2 = \rho_k \xi_1 + \sqrt{1 - \rho_k^2} \widetilde{\xi}_2$

4    Update $X_{t_{k+1}} \leftarrow e^{t_{k+1} - t_k} X_{t_k} + 2(e^{t_{k+1} - t_k} - 1) s_{T - t_k}(X_{t_k}) + \sqrt{e^{2(t_{k+1} - t_k)} - 1} \xi_1$; *// see Lem. 12 for the derivation*

5    Update $W \leftarrow W + (t_{k+1} - t_k)\|s_{T - t_k}(X_{t_k})\|^2 + \sqrt{2(t_{k+1} - t_k)} \langle s_{T - t_k}(X_{t_k}), \xi_2 \rangle$; *// see Lem. 12 for the derivation*

6 **end**

7 Update $W \leftarrow W + V(X_{t_N}) - (T - \delta)d$;

8 **return** $\widehat{Z} = e^{-W}$.

---

We define $\mathbb{P}$ as the path measure of the following SDE:

$$\mathrm{d}X_t = (\nabla \log \widetilde{\pi}_t + v_t)(X_t)\mathrm{d}t + \sqrt{2}\mathrm{d}B_t, \ t \in [0, T]; \ X_0 \sim \widetilde{\pi}_0, \tag{17}$$

where the vector field $(v_t)_{t \in [0,T]}$ is chosen such that $X_t \sim \widetilde{\pi}_t$ under $\mathbb{P}$ for all $t \in [0, T]$. According to the Fokker-Planck equation,[7] $(v_t)_{t \in [0,T]}$ must satisfy the PDE

$$\partial_t \widetilde{\pi}_t = -\nabla \cdot (\widetilde{\pi}_t(\nabla \log \widetilde{\pi}_t + v_t)) + \Delta \widetilde{\pi}_t = -\nabla \cdot (\widetilde{\pi}_t v_t), \ t \in [0, T],$$

which means that $(v_t)_{t \in [0,T]}$ generates $(\widetilde{\pi}_t)_{t \in [0,T]}$. The Nelson's relation (Lem. 2) implies an equivalent definition of $\mathbb{P}$ as the path measure of the following backward SDE with an independent time-reversed BM $(B_t^\leftarrow)_{t \in [0,T]}$:

$$\mathrm{d}X_t = (-\nabla \log \widetilde{\pi}_t + v_t)(X_t)\mathrm{d}t + \sqrt{2}\mathrm{d}B_t^\leftarrow, \ t \in [0, T]; \ X_T \sim \widetilde{\pi}_T.$$

Now we bound the probability of $\varepsilon$ relative error:

$$\Pr\left(\left|\frac{\widehat{Z}}{Z} - 1\right| \geq \varepsilon\right) = \mathbb{P}^\rightarrow\left(\left|\frac{e^{-W}}{e^{-\Delta F}} - 1\right| \geq \varepsilon\right) = \mathbb{P}^\rightarrow\left(\left|\frac{\mathrm{d}\mathbb{P}^\leftarrow}{\mathrm{d}\mathbb{P}^\rightarrow} - 1\right| \geq \varepsilon\right)$$

$$\leq \frac{1}{\varepsilon} \mathbb{E}_{\mathbb{P}^\rightarrow}\left|\frac{\mathrm{d}\mathbb{P}^\leftarrow}{\mathrm{d}\mathbb{P}^\rightarrow} - 1\right| = \frac{2}{\varepsilon} \mathrm{TV}(\mathbb{P}^\leftarrow, \mathbb{P}^\rightarrow)$$

$$\leq \frac{2}{\varepsilon}(\mathrm{TV}(\mathbb{P}, \mathbb{P}^\rightarrow) + \mathrm{TV}(\mathbb{P}, \mathbb{P}^\leftarrow))$$

$$\leq \frac{\sqrt{2}}{\varepsilon}\left(\sqrt{\mathrm{KL}(\mathbb{P}\|\mathbb{P}^\rightarrow)} + \sqrt{\mathrm{KL}(\mathbb{P}\|\mathbb{P}^\leftarrow)}\right). \tag{18}$$

In the second line above, we apply Markov inequality along with an equivalent definition of the TV distance: $\mathrm{TV}(\mu, \nu) = \frac{1}{2} \int \left|\frac{\mathrm{d}\mu}{\mathrm{d}\lambda} - \frac{\mathrm{d}\nu}{\mathrm{d}\lambda}\right| \mathrm{d}\lambda$, where $\lambda$ is a measure that dominates both $\mu$ and $\nu$. The third line follows from the triangle inequality for TV distance, while the final line is a consequence of Pinsker's inequality $\mathrm{KL} \geq 2\mathrm{TV}^2$.

---

[7]We assume the existence of a unique curve of probability measures solving the Fokker-Planck equation given the drift and diffusion terms, guaranteed under mild regularity conditions (Le Bris & Lions, 2008).

By Girsanov's theorem (Lems. 1 and 3), it is straightforward to see that

$$\mathrm{KL}(\mathbb{P}\|\mathbb{P}^{\leftarrow}) = \mathrm{KL}(\mathbb{P}\|\mathbb{P}^{\rightarrow}) = \frac{1}{4}\,\mathbb{E}_{\mathbb{P}}\int_0^T \|v_t(X_t)\|^2 \mathrm{d}t = \frac{1}{4}\int_0^T \|v_t\|_{L^2(\widetilde{\pi}_t)}^2 \mathrm{d}t.$$

Leveraging the relation between metric derivative and continuity equation (Lem. 4), among all vector fields $(v_t)_{t\in[0,T]}$ that generate $(\widetilde{\pi}_t)_{t\in[0,T]}$, we can choose the one that minimizes $\|v_t\|_{L^2(\widetilde{\pi}_t)}$, thereby making $\|v_t\|_{L^2(\widetilde{\pi}_t)} = |\dot{\widetilde{\pi}}|_t$, the metric derivative. With the reparameterization $\widetilde{\pi}_t = \pi_{t/T}$, we have the following relation by chain rule:

$$|\dot{\widetilde{\pi}}|_t = \lim_{\delta\to 0}\frac{\mathrm{W}_2(\widetilde{\pi}_{t+\delta},\widetilde{\pi}_t)}{|\delta|} = \lim_{\delta\to 0}\frac{\mathrm{W}_2(\pi_{(t+\delta)/T},\pi_{t/T})}{T|\delta/T|} = \frac{1}{T}|\dot{\pi}|_{t/T}.$$

Employing the change-of-variable formula leads to

$$\mathrm{KL}(\mathbb{P}\|\mathbb{P}^{\leftarrow}) = \mathrm{KL}(\mathbb{P}\|\mathbb{P}^{\rightarrow}) = \frac{1}{4}\int_0^T |\dot{\widetilde{\pi}}|_t^2 \mathrm{d}t = \frac{1}{4T}\int_0^1 |\dot{\pi}|_\theta^2 \mathrm{d}\theta = \frac{\mathcal{A}}{4T}.$$

Therefore, it suffices to choose $T = \frac{32\mathcal{A}}{\varepsilon^2}$ to make the r.h.s. of (18) less than $\frac{1}{4}$. $\qquad\square$

## D  PROOF OF THM. 4

With the forward and backward path measures $\mathbb{P}^{\rightarrow}$ and $\mathbb{P}^{\leftarrow}$ defined in (5) and (6), we further define the reference path measure

$$\mathbb{P}(x_{0:M}) = \pi_0(x_0)\prod_{\ell=1}^M F_\ell^*(x_{\ell-1},x_\ell), \tag{19}$$

where $F_\ell^*$ can be an arbitrary transition kernel transporting $\pi_{\theta_{\ell-1}}$ to $\pi_{\theta_\ell}$, i.e., it satisfies

$$\pi_{\theta_\ell}(y) = \int F_\ell^*(x,y)\pi_{\theta_{\ell-1}}(x)\mathrm{d}x,\ \forall y\in\mathbb{R}^d \implies x_\ell \sim \pi_{\theta_\ell},\ \forall\ell\in[\![0,M]\!].$$

Define the backward transition kernel of $F_\ell^*$ as

$$B_\ell^*(x,x') = \frac{\pi_{\theta_{\ell-1}}(x')}{\pi_{\theta_\ell}(x)}F_\ell^*(x',x),\ \ell\in[\![1,M]\!],$$

which transports $\pi_{\theta_\ell}$ to $\pi_{\theta_{\ell-1}}$. Equivalently, we can write

$$\mathbb{P}(x_{0:M}) = \pi_1(x_M)\prod_{\ell=1}^M B_\ell^*(x_\ell,x_{\ell-1}).$$

Straightforward calculations yield

$$\mathrm{KL}(\mathbb{P}\|\mathbb{P}^{\rightarrow}) = \sum_{\ell=1}^M \mathbb{E}_{\pi_{\theta_{\ell-1}}(x_{\ell-1})}\mathrm{KL}(F_\ell^*(x_{\ell-1},\cdot)\|F_\ell(x_{\ell-1},\cdot)),$$

$$\mathrm{KL}(\mathbb{P}\|\mathbb{P}^{\leftarrow}) = \sum_{\ell=1}^M \mathbb{E}_{\pi_{\theta_\ell}(x_\ell)}\mathrm{KL}(B_\ell^*(x_\ell,\cdot)\|B_\ell(x_\ell,\cdot))$$

$$= \sum_{\ell=1}^M \mathrm{KL}(\pi_{\theta_{\ell-1}}(x_{\ell-1})F_\ell^*(x_{\ell-1},x_\ell)\|\pi_{\theta_\ell}(x_{\ell-1})F_\ell(x_{\ell-1},x_\ell)) \tag{20}$$

$$= \mathrm{KL}(\mathbb{P}\|\mathbb{P}^{\rightarrow}) + \sum_{\ell=1}^M \mathrm{KL}(\pi_{\theta_{\ell-1}}\|\pi_{\theta_\ell}). \tag{21}$$

Also, recall that the sampling path measure $\widehat{\mathbb{P}}^{\rightarrow}$ is defined in (9) starts at $\widehat{\pi}_0$, the distribution of an approximate sample of $\pi_0$. For convenience, we define the following path measure, which differs from $\widehat{\mathbb{P}}^{\rightarrow}$ only from the initial distribution:

$$\overline{\mathbb{P}}^{\rightarrow}(x_{0:M}) = \pi_0(x_0)\prod_{\ell=1}^M \widehat{F}_\ell(x_{\ell-1},x_\ell). \tag{22}$$

Equipped with these definitions, we first prove a lemma about a necessary condition for the estimator $\widehat{Z}$ to satisfy the desired accuracy (1).

**Lemma 5.** *Define the estimator $\widehat{Z} := \widehat{Z}_0 e^{-W(x_{0:M})}$, where $x_{0:M} \sim \widehat{\mathbb{P}}^{\rightarrow}$, and $\widehat{Z}_0$ is independent of $x_{0:M}$. To make $\widehat{Z}$ satisfy the criterion (1), it suffices to meet the following four conditions:*

$$\Pr\left(\left|\frac{\widehat{Z}_0}{Z_0} - 1\right| \geq \frac{\varepsilon}{8}\right) \leq \frac{1}{8}, \tag{23}$$

$$\mathrm{TV}(\widehat{\pi}_0, \pi_0) \lesssim 1, \tag{24}$$

$$\mathrm{KL}(\mathbb{P}\|\mathbb{P}^{\leftarrow}) \lesssim \varepsilon^2, \tag{25}$$

$$\mathrm{KL}(\mathbb{P}\|\overline{\mathbb{P}}^{\rightarrow}) \lesssim 1. \tag{26}$$

*Proof.* Recall that $Z = Z_0 e^{-\Delta F}$. Using Lem. 8, we have

$$\Pr\left(\left|\frac{\widehat{Z}}{Z} - 1\right| \geq \varepsilon\right) \leq \Pr\left(\left|\log\frac{\widehat{Z}}{Z}\right| \geq \frac{\varepsilon}{2}\right) = \Pr_{x_{0:M}\sim\widehat{\mathbb{P}}^{\rightarrow}}\left(\left|\log\frac{\widehat{Z}_0}{Z_0} + \log\frac{e^{-W(x_{0:M})}}{e^{-\Delta F}}\right| \geq \frac{\varepsilon}{2}\right)$$

$$\leq \Pr\left(\left|\log\frac{\widehat{Z}_0}{Z_0}\right| \geq \frac{\varepsilon}{4}\right) + \widehat{\mathbb{P}}^{\rightarrow}\left(\left|\log\frac{e^{-W}}{e^{-\Delta F}}\right| \geq \frac{\varepsilon}{4}\right)$$

$$\leq \Pr\left(\left|\frac{\widehat{Z}_0}{Z_0} - 1\right| \geq \frac{\varepsilon}{8}\right) + \widehat{\mathbb{P}}^{\rightarrow}\left(\left|\frac{e^{-W}}{e^{-\Delta F}} - 1\right| \geq \frac{\varepsilon}{8}\right).$$

The first term is $\leq \frac{1}{8}$ if (23) holds. To bound the second term, using the definition of TV distance and the triangle inequality, we have

$$\widehat{\mathbb{P}}^{\rightarrow}\left(\left|\frac{e^{-W}}{e^{-\Delta F}} - 1\right| \geq \frac{\varepsilon}{8}\right)$$

$$\leq \mathrm{TV}(\widehat{\mathbb{P}}^{\rightarrow}, \mathbb{P}^{\rightarrow}) + \mathbb{P}^{\rightarrow}\left(\left|\frac{e^{-W}}{e^{-\Delta F}} - 1\right| \geq \frac{\varepsilon}{8}\right)$$

$$\leq \mathrm{TV}(\widehat{\mathbb{P}}^{\rightarrow}, \overline{\mathbb{P}}^{\rightarrow}) + \mathrm{TV}(\overline{\mathbb{P}}^{\rightarrow}, \mathbb{P}) + \mathrm{TV}(\mathbb{P}, \mathbb{P}^{\rightarrow}) + \mathbb{P}^{\rightarrow}\left(\left|\frac{d\mathbb{P}^{\leftarrow}}{d\mathbb{P}^{\rightarrow}} - 1\right| \geq \frac{\varepsilon}{8}\right).$$

Recall that by definition (9) and (22), the distributions of $x_{1:M}$ conditional on $x_0$ are the same under $\widehat{\mathbb{P}}^{\rightarrow}$ and $\overline{\mathbb{P}}^{\rightarrow}$. Hence, $\mathrm{TV}(\widehat{\mathbb{P}}^{\rightarrow}, \overline{\mathbb{P}}^{\rightarrow}) = \mathrm{TV}(\widehat{\pi}_0, \pi_0)$. Applying Pinsker's inequality and leveraging (18), we have

$$\widehat{\mathbb{P}}^{\rightarrow}\left(\left|\frac{e^{-W}}{e^{-\Delta F}} - 1\right| \geq \frac{\varepsilon}{8}\right)$$

$$\lesssim \mathrm{TV}(\widehat{\pi}_0, \pi_0) + \sqrt{\mathrm{KL}(\mathbb{P}\|\overline{\mathbb{P}}^{\rightarrow})} + \sqrt{\mathrm{KL}(\mathbb{P}\|\mathbb{P}^{\rightarrow})} + \frac{\sqrt{\mathrm{KL}(\mathbb{P}\|\mathbb{P}^{\rightarrow})} + \sqrt{\mathrm{KL}(\mathbb{P}\|\mathbb{P}^{\leftarrow})}}{\varepsilon}.$$

Note that from (21) we know that $\mathrm{KL}(\mathbb{P}\|\mathbb{P}^{\rightarrow}) \leq \mathrm{KL}(\mathbb{P}\|\mathbb{P}^{\leftarrow})$, so if (24) to (26) hold up to some small enough absolute constants, we can achieve $\widehat{\mathbb{P}}^{\rightarrow}\left(\left|\frac{e^{-W}}{e^{-\Delta F}} - 1\right| \geq \frac{\varepsilon}{8}\right) \leq \frac{1}{8}$, and therefore $\Pr\left(\left|\frac{\widehat{Z}}{Z} - 1\right| \geq \varepsilon\right) \leq \frac{1}{4}$. $\square$

**Remark 6.** *As the TV distance is always upper bounded by 1, one can in fact write* (24) *to* (26) *in a more precise way:* $\mathrm{TV}(\widehat{\pi}_0, \pi_0) \leq 2^{-5}$, $\mathrm{KL}(\mathbb{P}\|\mathbb{P}^{\leftarrow}) \leq 2^{-13}\varepsilon^2$, *and* $\mathrm{KL}(\mathbb{P}\|\overline{\mathbb{P}}^{\rightarrow}) \leq 2^{-8}$.

Next, we study how to satisfy the conditions in (25) and (26) while minimizing oracle complexity. Given that we already have an approximate sample from $\pi_0$ and an accurate estimate of $Z_0$, we proceed to the next step of the AIS algorithm. Since each transition kernel requires one call to the oracle of $\nabla V$, and by plugging in $f_\theta \leftarrow V + \frac{\lambda(\theta)}{2}\|\cdot\|^2$ in AIS (Thm. 3), the work function $W(x_{0:M})$ is independent of $V$, it follows that the remaining oracle complexity is $M$. The result is formalized in the following lemma.

**Lemma 6.** *To minimize the oracle complexity, it suffices to find the minimal $M$ such that there exists a sequence $0 = \theta_0 < \theta_1 < ... < \theta_M = 1$ satisfying the following three constraints:*

$$\sum_{\ell=1}^{M} \int_{\theta_{\ell-1}}^{\theta_\ell} (\lambda(\theta) - \lambda(\theta_\ell))^2 \mathrm{d}\theta \lesssim \frac{\varepsilon^4}{m^2 \mathcal{A}}, \tag{27}$$

$$\sum_{\ell=1}^{M} (\theta_\ell - \theta_{\ell-1})^2 \lesssim \frac{\varepsilon^4}{d\beta^2 \mathcal{A}^2}, \tag{28}$$

$$\max_{\ell \in [\![1,M]\!]} (\theta_\ell - \theta_{\ell-1}) \lesssim \frac{\varepsilon^2}{\beta \mathcal{A}}. \tag{29}$$

*Proof.* We break down the argument into two steps.

**Step 1.** We first consider (25).

Note that when defining the reference path measure $\mathbb{P}$, the only requirement for the transition kernel $F_\ell^*$ is that it should transport $\pi_{\theta_{\ell-1}}$ to $\pi_{\theta_\ell}$. Our aim is to find the "optimal" $F_\ell^*$'s in order to minimize the sum of KL divergences, which can be viewed as a *static Schrödinger bridge problem* (Léonard, 2014; Chen et al., 2016; 2021). By data-processing inequality,

$$C_\ell := \inf_{F_\ell^*} \mathrm{KL}(\pi_{\theta_{\ell-1}}(x_{\ell-1}) F_\ell^*(x_{\ell-1}, x_\ell) \| \pi_{\theta_\ell}(x_{\ell-1}) F_\ell(x_{\ell-1}, x_\ell)) \leq \inf_{\mathbf{P}^\ell} \mathrm{KL}(\mathbf{P}^\ell \| \mathbf{Q}^\ell),$$

where the infimum is taken among all path measures from $0$ to $T_\ell$ with the marginal constraints $\mathbf{P}_0^\ell = \pi_{\theta_{\ell-1}}$ and $\mathbf{P}_{T_\ell}^\ell = \pi_{\theta_\ell}$; $\mathbf{Q}^\ell$ is the path measure of (8) (i.e., LD with target distribution $\pi_{\theta_\ell}$) initialized at $X_0 \sim \pi_{\theta_\ell}$.

For each $\ell \in [\![1, M]\!]$, define the following interpolation between $\pi_{\theta_{\ell-1}}$ and $\pi_{\theta_\ell}$:

$$\mu_t^\ell := \pi_{\theta_{\ell-1} + \frac{t}{T_\ell}(\theta_\ell - \theta_{\ell-1})}, \ t \in [0, T_\ell].$$

Let $\mathbf{P}^\ell$ be the path measure of

$$\mathrm{d}X_t = (\nabla \log \mu_t^\ell + u_t^\ell)(X_t)\mathrm{d}t + \sqrt{2}\mathrm{d}B_t, \ t \in [0, T_\ell]; \ X_0 \sim \pi_{\theta_{\ell-1}},$$

where the vector field $(u_t^\ell)_{t \in [0,T_\ell]}$ is chosen such that $X_t \sim \mu_t^\ell$ under $\mathbf{P}^\ell$, and in particular, the marginal distributions at $0$ and $T_\ell$ are $\pi_{\theta_{\ell-1}}$ and $\pi_{\theta_\ell}$, respectively. By verifying the Fokker-Planck equation, the following PDE needs to be satisfied:

$$\partial_t \mu_t^\ell = -\nabla \cdot (\mu_t^\ell (\nabla \log \mu_t^\ell + u_t^\ell)) + \Delta \mu_t^\ell = -\nabla \cdot (\mu_t^\ell u_t^\ell), \ t \in [0, T_\ell],$$

meaning that $(u_t^\ell)_{t \in [0,T_\ell]}$ generates $(\mu_t^\ell)_{t \in [0,T_\ell]}$. Similar to the proof of JE (Thm. 2), using the relation between metric derivative and continuity equation (Lem. 4), among all vector fields generating $(\mu_t^\ell)_{t \in [0,T_\ell]}$, we choose $(u_t^\ell)_{t \in [0,T_\ell]}$ to be the a.s.-unique vector field that satisfies $\|u_t^\ell\|_{L^2(\mu_t^\ell)} = |\dot{\mu}^\ell|_t$ for Lebesgue-a.e. $t \in [0, T_\ell]$, which implies (using the chain rule)

$$\int_0^{T_\ell} \|u_t^\ell\|_{L^2(\mu_t^\ell)}^2 \mathrm{d}t = \int_0^{T_\ell} |\dot{\mu}^\ell|_t^2 \mathrm{d}t$$

$$= \int_0^{T_\ell} \left( \frac{\theta_\ell - \theta_{\ell-1}}{T_\ell} |\dot{\pi}|_{\theta_{\ell-1} + \frac{t}{T_\ell}(\theta_\ell - \theta_{\ell-1})} \right)^2 \mathrm{d}t = \frac{\theta_\ell - \theta_{\ell-1}}{T_\ell} \int_{\theta_{\ell-1}}^{\theta_\ell} |\dot{\pi}|_\theta^2 \mathrm{d}\theta.$$

By Lem. 2, we can equivalently write $\mathbf{P}^\ell$ as the path measure of the following backward SDE:

$$\mathrm{d}X_t = (-\nabla \log \mu_t^\ell + u_t^\ell)(X_t)\mathrm{d}t + \sqrt{2}\mathrm{d}B_t^{\leftarrow}, \ t \in [0, T_\ell]; \ X_T \sim \pi_{\theta_\ell}.$$

Recall that $\mathbf{Q}^\ell$ is the path measure of (8) initialized at $X_0 \sim \pi_{\theta_\ell}$, so $X_t \sim \pi_{\theta_\ell}$ for all $t \in [0, T_\ell]$. By Nelson's relation (Lem. 2), we can equivalently write $\mathbf{Q}^\ell$ as the path measure of

$$\mathrm{d}X_t = -\nabla \log \pi_{\theta_\ell}(X_t)\mathrm{d}t + \sqrt{2}\mathrm{d}B_t^{\leftarrow}, \ t \in [0, T_\ell]; \ X_{T_\ell} \sim \pi_{\theta_\ell}.$$

The purpose of writing these two path measures in the way of backward SDEs is to avoid the extra term of the KL divergence between the initialization distributions $\pi_{\theta_{\ell-1}}$ and $\pi_{\theta_\ell}$ at time $0$ when calculating $\mathrm{KL}(\mathbf{P}^\ell \| \mathbf{Q}^\ell)$. To see this, by Girsanov's theorem (Lem. 3), the triangle inequality, and the change-of-variable formula, we have

$$
\begin{aligned}
C_\ell \leq \mathrm{KL}(\mathbf{P}^\ell \| \mathbf{Q}^\ell) &= \frac{1}{4} \int_0^{T_\ell} \left\| u_t^\ell - \nabla \log \frac{\mu_t^\ell}{\pi_{\theta_\ell}} \right\|_{L^2(\mu_t^\ell)}^2 \mathrm{d}t \\
&\lesssim \int_0^{T_\ell} \|u_t^\ell\|_{L^2(\mu_t^\ell)}^2 \mathrm{d}t + \int_0^{T_\ell} \left\| \nabla \log \frac{\mu_t^\ell}{\pi_{\theta_\ell}} \right\|_{L^2(\mu_t^\ell)}^2 \mathrm{d}t \\
&= \frac{\theta_\ell - \theta_{\ell-1}}{T_\ell} \int_{\theta_{\ell-1}}^{\theta_\ell} |\dot\pi|_\theta^2 \mathrm{d}\theta + \frac{T_\ell}{\theta_\ell - \theta_{\ell-1}} \int_{\theta_{\ell-1}}^{\theta_\ell} \left\| \nabla \log \frac{\pi_\theta}{\pi_{\theta_\ell}} \right\|_{L^2(\pi_\theta)}^2 \mathrm{d}\theta.
\end{aligned}
$$

**Remark 7.** *Our bound above is based on a specific interpolation between $\pi_{\theta_{\ell-1}}$ and $\pi_{\theta_\ell}$ along the curve $(\pi_\theta)_{\theta \in [\theta_{\ell-1}, \theta_\ell]}$. This approach is inspired by, yet slightly differs from, Conforti & Tamanini (2021, Theorem 1.6), where the interpolation is based on the Wasserstein geodesic. As we will demonstrate shortly, our formulation simplifies the analysis of the second term (the Fisher divergence), making it more straightforward to bound.*

Now, summing over all $\ell \in [\![1, M]\!]$, we can see that in order to ensure $\mathrm{KL}(\mathbb{P} \| \mathbb{P}^\leftarrow) \leq \sum_{\ell=1}^M C_\ell \leq \varepsilon^2$, we only need the following two conditions to hold:

$$
\sum_{\ell=1}^M \frac{\theta_\ell - \theta_{\ell-1}}{T_\ell} \int_{\theta_{\ell-1}}^{\theta_\ell} |\dot\pi|_\theta^2 \mathrm{d}\theta \lesssim \varepsilon^2, \tag{30}
$$

$$
\sum_{\ell=1}^M \frac{T_\ell}{\theta_\ell - \theta_{\ell-1}} \int_{\theta_{\ell-1}}^{\theta_\ell} \left\| \nabla \log \frac{\pi_\theta}{\pi_{\theta_\ell}} \right\|_{L^2(\pi_\theta)}^2 \mathrm{d}\theta \lesssim \varepsilon^2. \tag{31}
$$

Since $\sum_{\ell=1}^M \int_{\theta_{\ell-1}}^{\theta_\ell} |\dot\pi|_\theta^2 \mathrm{d}\theta = \mathcal{A}$, it suffices to choose

$$
\frac{T_\ell}{\theta_\ell - \theta_{\ell-1}} =: T \asymp \frac{\mathcal{A}}{\varepsilon^2}, \ \forall \ell \in [\![1, M]\!] \tag{32}
$$

to make the l.h.s. of (30) $O(\varepsilon^2)$. Notably, $T$ is the summation over all $T_\ell$'s, which has the same order as the total time $T$ for running JE ((2)) in the continuous scenario, in Thm. 1. Plugging this $T_\ell$ into the second summation, and noticing that by (7) and Lem. 14,

$$
\left\| \nabla \log \frac{\pi_\theta}{\pi_{\theta'}} \right\|_{L^2(\pi_\theta)}^2 = \mathbb{E}_{x \sim \pi_\theta} \|(\lambda(\theta) - \lambda(\theta'))x\|^2 \leq (\lambda(\theta) - \lambda(\theta'))^2 m^2,
$$

we conclude that (27) implies (31).

**Step 2.** Now consider the other constraint (26). By chain rule and data-processing inequality,

$$
\mathrm{KL}(\mathbb{P} \| \overrightarrow{\overline{\mathbb{P}}}) = \sum_{\ell=1}^M \mathrm{KL}(\pi_{\theta_{\ell-1}}(x_{\ell-1}) F_\ell^*(x_{\ell-1}, x_\ell) \| \pi_{\theta_{\ell-1}}(x_{\ell-1}) \widehat{F}_\ell(x_{\ell-1}, x_\ell)) \leq \sum_{\ell=1}^M \mathrm{KL}(\mathbf{P}^\ell \| \widehat{\mathbf{Q}}^\ell),
$$

where $\mathbf{P}^\ell$ is the previously defined path measure of the SDE

$$
\begin{aligned}
\mathrm{d}X_t &= (\nabla \log \mu_t^\ell + u_t^\ell)(X_t)\mathrm{d}t + \sqrt{2}\mathrm{d}B_t \\
&= \left( -\nabla V(X_t) - \lambda \left( \theta_{\ell-1} + \frac{t}{T_\ell}(\theta_\ell - \theta_{\ell-1}) \right) X_t + u_t^\ell(X_t) \right) \mathrm{d}t + \sqrt{2}\mathrm{d}B_t, \ t \in [0, T_\ell]; \ X_0 \sim \pi_{\theta_{\ell-1}},
\end{aligned}
$$

and $\widehat{\mathbf{Q}}^\ell$ is the path measure of (10) initialized at $X_0 \sim \pi_{\theta_{\ell-1}}$, i.e.,

$$
\mathrm{d}X_t = \left( -\nabla V(X_0) - \lambda \left( \theta_{\ell-1} + \frac{t}{T_\ell}(\theta_\ell - \theta_{\ell-1}) \right) X_t \right) \mathrm{d}t + \sqrt{2}\mathrm{d}B_t, \ t \in [0, T_\ell]; \ X_0 \sim \pi_{\theta_{\ell-1}}.
$$

By Lem. 1, triangle inequality, and the smoothness of $V$, we have

$$\mathrm{KL}(\mathbf{P}^\ell\|\widehat{\mathbf{Q}}^\ell) = \frac{1}{4}\int_0^{T_\ell} \mathbb{E}_{\mathbf{P}^\ell}\|\nabla V(X_t) - \nabla V(X_0) - u_t^\ell(X_t)\|^2 \mathrm{d}t$$

$$\lesssim \int_0^{T_\ell} \mathbb{E}_{\mathbf{P}^\ell}\left[\|\nabla V(X_t) - \nabla V(X_0)\|^2 + \|u_t^\ell(X_t)\|^2\right]\mathrm{d}t$$

$$\le \beta^2 \int_0^{T_\ell} \mathbb{E}_{\mathbf{P}^\ell}\|X_t - X_0\|^2 \mathrm{d}t + \int_0^{T_\ell}\|u_t^\ell\|_{L^2(\mu_t^\ell)}^2 \mathrm{d}t$$

To bound the first part, note that under $\mathbf{P}^\ell$, we have

$$X_t - X_0 = \int_0^t (\nabla \log \mu_\tau^\ell + u_\tau^\ell)(X_\tau)\mathrm{d}\tau + \sqrt{2}B_t.$$

Thanks to the fact that $X_t \sim \mu_t^\ell$ under $\mathbf{P}^\ell$,

$$\mathbb{E}_{\mathbf{P}^\ell}\|X_t - X_0\|^2 \lesssim \mathbb{E}_{\mathbf{P}^\ell}\left\|\int_0^t (\nabla \log \mu_\tau^\ell + u_\tau^\ell)(X_\tau)\mathrm{d}\tau\right\|^2 + \mathbb{E}\|\sqrt{2}B_t\|^2$$

$$\lesssim t\int_0^t \mathbb{E}_{\mathbf{P}^\ell}\|(\nabla \log \mu_\tau^\ell + u_\tau^\ell)(X_\tau)\|^2 \mathrm{d}\tau + dt$$

$$\lesssim t\int_0^t \left(\|\nabla \log \mu_\tau^\ell\|_{L^2(\mu_\tau^\ell)}^2 + \|u_\tau^\ell\|_{L^2(\mu_\tau^\ell)}^2\right)\mathrm{d}\tau + dt$$

$$\lesssim T_\ell \int_0^{T_\ell}\left(\|\nabla \log \mu_\tau^\ell\|_{L^2(\mu_\tau^\ell)}^2 + \|u_\tau^\ell\|_{L^2(\mu_\tau^\ell)}^2\right)\mathrm{d}\tau + dT_\ell, \ \forall t \in [0, T_\ell],$$

where the second inequality follows from Jensen's inequality (Cheng et al., 2018, Sec. 4):

$$\left\|\int_0^t f_\tau \mathrm{d}\tau\right\|^2 = t^2\|\mathbb{E}_{\tau\sim\mathrm{Unif}(0,t)}f_\tau\|^2 \le t^2 \mathbb{E}_{\tau\sim\mathrm{Unif}(0,t)}\|f_\tau\|^2 = t\int_0^t \|f_\tau\|^2\mathrm{d}\tau.$$

Therefore,

$$\mathrm{KL}(\mathbf{P}^\ell\|\widehat{\mathbf{Q}}^\ell)$$

$$\le \beta^2 \int_0^{T_\ell}\mathbb{E}_{\mathbf{P}^\ell}\|X_t - X_0\|^2\mathrm{d}t + \int_0^{T_\ell}\|u_t^\ell\|_{L^2(\mu_t^\ell)}^2\mathrm{d}t$$

$$\le \beta^2 T_\ell^2 \int_0^{T_\ell}\|\nabla \log \mu_\tau^\ell\|_{L^2(\mu_\tau^\ell)}^2\mathrm{d}\tau + (\beta^2 T_\ell^2 + 1)\int_0^{T_\ell}\|u_\tau^\ell\|_{L^2(\mu_\tau^\ell)}^2\mathrm{d}\tau + d\beta^2 T_\ell^2$$

$$= \beta^2 T_\ell^2 \frac{T_\ell}{\theta_\ell - \theta_{\ell-1}}\int_{\theta_{\ell-1}}^{\theta_\ell}\|\nabla \log \pi_\theta\|_{L^2(\pi_\theta)}^2\mathrm{d}\theta + (\beta^2 T_\ell^2 + 1)\frac{\theta_\ell - \theta_{\ell-1}}{T_\ell}\int_{\theta_{\ell-1}}^{\theta_\ell}|\dot\pi|_\theta^2\mathrm{d}\theta + d\beta^2 T_\ell^2.$$

Recall that the potential of $\pi_\theta$ is $(\beta + \lambda(\theta))$-smooth. By Lem. 13 and the monotonicity of $\lambda(\cdot)$,

$$\int_{\theta_{\ell-1}}^{\theta_\ell}\|\nabla \log \pi_\theta\|_{L^2(\pi_\theta)}^2\mathrm{d}\theta \le \int_{\theta_{\ell-1}}^{\theta_\ell} d(\beta + \lambda(\theta))\mathrm{d}\theta \le d(\theta_\ell - \theta_{\ell-1})(\beta + \lambda(\theta_{\ell-1})).$$

Thus,

$$\mathrm{KL}(\mathbb{P}\|\overrightarrow{\mathbb{P}}) \le \sum_{\ell=1}^M \left(\beta^2 T_\ell^3 d(\beta + \lambda(\theta_{\ell-1})) + (\beta^2 T_\ell^2 + 1)\frac{\theta_\ell - \theta_{\ell-1}}{T_\ell}\int_{\theta_{\ell-1}}^{\theta_\ell}|\dot\pi|_\theta^2\mathrm{d}\theta + d\beta^2 T_\ell^2\right)$$

$$= \sum_{\ell=1}^M\left(\beta^2 dT_\ell^2\left(T_\ell(\beta + \lambda(\theta_{\ell-1})) + 1\right) + (\beta^2 T_\ell^2 + 1)\frac{\theta_\ell - \theta_{\ell-1}}{T_\ell}\int_{\theta_{\ell-1}}^{\theta_\ell}|\dot\pi|_\theta^2\mathrm{d}\theta\right)$$

Assume $\max_{\ell \in [\![1,M]\!]} T_\ell \lesssim \frac{1}{\beta}$, i.e., (29). so $\max_{\ell \in [\![1,M]\!]} T_\ell(\beta + \lambda(\theta_{\ell-1})) \lesssim 1$, due to $\lambda(\cdot) \le 2\beta$. We can further simplify the above expression to

$$\text{KL}(\mathbb{P}\|\overline{\mathbb{P}}^{\rightarrow}) \le \sum_{\ell=1}^{M} \left( \beta^2 dT_\ell^2 + \frac{\theta_\ell - \theta_{\ell-1}}{T_\ell} \int_{\theta_{\ell-1}}^{\theta_\ell} |\dot{\pi}|_\theta^2 d\theta \right) \lesssim \beta^2 d \left( \sum_{\ell=1}^{M} T_\ell^2 \right) + \varepsilon^2$$

$$= \beta^2 dT^2 \sum_{\ell=1}^{M} (\theta_\ell - \theta_{\ell-1})^2 + \varepsilon^2 \lesssim \beta^2 d \frac{\mathcal{A}^2}{\varepsilon^4} \sum_{\ell=1}^{M} (\theta_\ell - \theta_{\ell-1})^2 + \varepsilon^2.$$

So (29) implies that the r.h.s. of the above equation is $O(1)$.

$\square$

Finally, we have arrived at the last step of proving Thm. 4, that is to decide the schedule of $\theta_\ell$'s.

Define $\vartheta_\ell := 1 - \theta_\ell, \ell \in [\![0, M]\!]$. We consider the annealing schedule $\lambda(\theta) = 2\beta(1 - \theta)^r$ for some $1 \le r \lesssim 1$, and to emphasize the dependence on $r$, we use $\mathcal{A}_r$ to represent the action of $(\pi_\theta)_{\theta \in [0,1]}$. The l.h.s. of (27) is

$$\sum_{\ell=1}^{M} \int_{\theta_{\ell-1}}^{\theta_\ell} (\lambda(\theta) - \lambda(\theta_\ell))^2 d\theta \le \sum_{\ell=1}^{M} (\theta_\ell - \theta_{\ell-1})(2\beta(1-\theta_{\ell-1})^r - 2\beta(1-\theta_\ell)^r)^2$$

$$= \sum_{\ell=1}^{M} (\vartheta_{\ell-1} - \vartheta_\ell)(2\beta\vartheta_{\ell-1}^r - 2\beta\vartheta_\ell^r)^2$$

$$\lesssim \beta^2 \sum_{\ell=1}^{M} (\vartheta_{\ell-1} - \vartheta_\ell)(\vartheta_{\ell-1}^r - \vartheta_\ell^r)^2$$

$$\lesssim \beta^2 \sum_{\ell=1}^{M} (\vartheta_{\ell-1} - \vartheta_\ell)(\vartheta_{\ell-1} - \vartheta_\ell)^2 = \beta^2 \sum_{\ell=1}^{M} (\vartheta_{\ell-1} - \vartheta_\ell)^3,$$

where the last inequality comes from Lem. 9. So to satisfy (27), it suffices to ensure

$$\sum_{\ell=1}^{M} (\vartheta_{\ell-1} - \vartheta_\ell)^3 \lesssim \frac{\varepsilon^4}{m^2 \beta^2 \mathcal{A}_r},$$

while (28) and (29) are equivalent to

$$\sum_{\ell=1}^{M} (\vartheta_{\ell-1} - \vartheta_\ell)^2 \lesssim \frac{\varepsilon^4}{d\beta^2 \mathcal{A}_r^2}, \qquad \max_{\ell \in [\![1,M]\!]} (\vartheta_{\ell-1} - \vartheta_\ell) \lesssim \frac{\varepsilon^2}{\beta \mathcal{A}_r}.$$

Since we are minimizing the total number of oracle calls $M$, the Hölder's inequality implies that the optimal schedule of $\vartheta_\ell$'s is an arithmetic sequence, i.e., $\vartheta_\ell = 1 - \frac{\ell}{M}$. We need to ensure

$$\frac{1}{M^2} \lesssim \frac{\varepsilon^4}{m^2 \beta^2 \mathcal{A}_r}, \qquad \frac{1}{M} \lesssim \frac{\varepsilon^4}{d\beta^2 \mathcal{A}_r^2}, \qquad \frac{1}{M} \lesssim \frac{\varepsilon^2}{\beta \mathcal{A}_r}.$$

So it suffices to choose $\frac{1}{M} \asymp \frac{\varepsilon^2}{m\beta \mathcal{A}_r^{\frac{1}{2}}} \wedge \frac{\varepsilon^4}{d\beta^2 \mathcal{A}_r^2}$, which implies the oracle complexity

$$M \asymp \frac{m\beta \mathcal{A}_r^{\frac{1}{2}}}{\varepsilon^2} \vee \frac{d\beta^2 \mathcal{A}_r^2}{\varepsilon^4},$$

and the hyperparameter $T_\ell$ is thus $T_\ell \asymp \frac{\mathcal{A}_r^{\frac{1}{2}}}{m\beta} \wedge \frac{\varepsilon^2}{d\beta^2 \mathcal{A}_r}$ according to (32).

$\square$

**Remark 8.** *The work Guo et al. (2025) used similar methodologies to prove an $\widetilde{O}\left( \frac{d\beta^2 \mathcal{A}^2}{\varepsilon^6} \right)$ oracle complexity for obtaining a sample that is $\varepsilon^2$-close in KL divergence to the target distribution. While*

*our assumptions are mostly the same with Guo et al. (2025) except for some insignificant technical ones, and both proofs involve the standard discretization analysis through Girsanov's theorem, the improvement of the $\varepsilon$-dependency in Thm. 4 is due to the fact that Guo et al. (2025) requires $\mathrm{KL}(\mathbb{P}\|\overline{\mathbb{P}}^{\rightarrow}) \lesssim \varepsilon^2$ for sampling, which results in a $\widetilde{\Theta}(\varepsilon^4)$ step size in Guo et al. (2025), while our proof only needs $\mathrm{KL}(\mathbb{P}\|\overline{\mathbb{P}}^{\rightarrow}) \lesssim 1$ and $\mathrm{KL}(\mathbb{P}\|\mathbb{P}^{\leftarrow}) \lesssim \varepsilon^2$ for normalizing constant estimation, resulting in an improved $\widetilde{\Theta}(\varepsilon^2)$ step size.*

# E  PROOFS FOR SEC. 5

## E.1  PROOF OF PROP. 1

*Proof.* The claim of smoothness follows from Guo et al. (2025, Lem. 7). A similar approach for proving the lower bound of metric derivative was used independently in Chemseddine et al. (2025, App. B).

Throughout this proof, let $\phi$ and $\Phi$ denote the p.d.f. and c.d.f. of the standard normal distribution $\mathcal{N}(0,1)$, respectively. Unless otherwise specified, the integration ranges are assumed to be $(-\infty, \infty)$.

Note that

$$
\pi(x)\mathrm{e}^{-\frac{\lambda}{2}x^2} \propto \left(\mathrm{e}^{-\frac{x^2}{2}} + \mathrm{e}^{-\frac{(x-m)^2}{2}}\right)\mathrm{e}^{-\frac{\lambda}{2}x^2}
$$

$$
= \mathrm{e}^{-\frac{\lambda+1}{2}x^2} + \mathrm{e}^{-\frac{\lambda m^2}{2(\lambda+1)}}\mathrm{e}^{-\frac{\lambda+1}{2}\left(x-\frac{m}{\lambda+1}\right)^2}
$$

$$
= \frac{1}{1+\mathrm{e}^{-\frac{\lambda m^2}{2(\lambda+1)}}}\mathcal{N}\left(x\,\Big|\,0, \frac{1}{\lambda+1}\right) + \frac{\mathrm{e}^{-\frac{\lambda m^2}{2(\lambda+1)}}}{1+\mathrm{e}^{-\frac{\lambda m^2}{2(\lambda+1)}}}\mathcal{N}\left(x\,\Big|\,\frac{m}{\lambda+1}, \frac{1}{\lambda+1}\right).
$$

Define $S(\theta) := \frac{1}{1+m^2(1-\theta)^r}$, and let

$$
\underline{\pi}_s(x) :\propto \pi(x)\mathrm{e}^{-\frac{1/s-1}{2}x^2} = w(s)\mathcal{N}(x|0,s) + (1-w(s))\mathcal{N}(x|sm, s),
$$

where

$$
w(s) = \frac{1}{1+\mathrm{e}^{-(1-s)m^2/2}}, \quad w'(s) = -\frac{\mathrm{e}^{-(1-s)m^2/2}m^2/2}{(1+\mathrm{e}^{-(1-s)m^2/2})^2}.
$$

By definition, $\pi_\theta = \underline{\pi}_{S(\theta)}$. The p.d.f. of $\underline{\pi}_s$ is

$$
f_s(x) = \frac{w(s)}{\sqrt{s}}\phi\left(\frac{x}{\sqrt{s}}\right) + \frac{1-w(s)}{\sqrt{s}}\phi\left(\frac{x-sm}{\sqrt{s}}\right),
$$

and the c.d.f. of $\underline{\pi}_s$ is

$$
F_s(x) = w(s)\Phi\left(\frac{x}{\sqrt{s}}\right) + (1-w(s))\Phi\left(\frac{x-sm}{\sqrt{s}}\right).
$$

We now derive a formula for calculating the metric derivative. From Villani (2003, Thm. 2.18), $\mathrm{W}_2^2(\mu,\nu) = \int_0^1 (F_\mu^{-1}(y) - F_\nu^{-1}(y))^2\mathrm{d}y$, where $F_\mu, F_\nu$ stand for the c.d.f.s of $\mu, \nu$. Assuming regularity conditions hold, we have

$$
\lim_{\delta\to 0}\frac{\mathrm{W}_2^2(\underline{\pi}_s, \underline{\pi}_{s+\delta})}{\delta^2} = \lim_{\delta\to 0}\int_0^1\left(\frac{F_{s+\delta}^{-1}(y) - F_s^{-1}(y)}{\delta}\right)^2\mathrm{d}y = \int_0^1(\partial_s F_s^{-1}(y))^2\mathrm{d}y.
$$

Consider change of variable $y = F_s(x)$, then $\frac{\mathrm{d}y}{\mathrm{d}x} = f_s(x)$. As $x = F_s^{-1}(y)$, $(F_s^{-1})'(y) = \frac{\mathrm{d}x}{\mathrm{d}y} = \frac{1}{f_s(x)}$. Taking the derivative of $s$ on both sides of the equation $x = F_s^{-1}(F_s(x))$ yields

$$
0 = \partial_s F_s^{-1}(F_s(x)) + (F_s^{-1})'(F_s(x))\partial_s F_s(x) = \partial_s F_s^{-1}(y) + \frac{1}{f_s(x)}\partial_s F_s(x).
$$

Therefore,

$$\int_0^1 (\partial_s F_s^{-1}(y))^2 \mathrm{d}y = \int \left( \frac{\partial_s F_s(x)}{f_s(x)} \right)^2 f_s(x) \mathrm{d}x = \int \frac{(\partial_s F_s(x))^2}{f_s(x)} \mathrm{d}x.$$

Consider the interval $x \in \left[ \frac{m}{2} - 0.1, \frac{m}{2} + 0.1 \right]$, and fix the range of $s$ to be $[0.9, 0.99]$. We have

$$\begin{cases} 1 - w(s) = \frac{1}{1+e^{(1-s)m^2/2}} \asymp \frac{1}{e^{(1-s)m^2/2}}, & \forall m \gtrsim 1 \\ -w'(s) = \frac{e^{(1-s)m^2/2}m^2/2}{(1+e^{(1-s)m^2/2})^2} \asymp \frac{m^2}{e^{(1-s)m^2/2}}, & \forall m \gtrsim 1 \end{cases}$$

First consider upper bounding $f_s(x)$. We have the following two bounds:

$$\frac{w(s)}{\sqrt{s}} \phi \left( \frac{x}{\sqrt{s}} \right) \lesssim e^{-\frac{x^2}{2s}} \le e^{-\frac{(m/2-0.1)^2}{2\times 0.99}} \le e^{-\frac{m^2}{8}}, \ \forall m \gtrsim 1,$$

$$\frac{1-w(s)}{\sqrt{s}} \phi \left( \frac{x-sm}{\sqrt{s}} \right) \lesssim \frac{1}{e^{(1-s)m^2/2}} e^{-\frac{(sm-x)^2}{2s}} = \exp\left( -\frac{1}{2} \left[ \frac{(sm-x)^2}{s} + (1-s)m^2 \right] \right).$$

The term in the square brackets above is

$$\frac{(sm-x)^2}{s} + (1-s)m^2 \ge \frac{1}{s} \left( sm - \frac{m}{2} - 0.1 \right)^2 + (1-s)m^2$$

$$= \frac{m^2}{4s} - 0.2 \left( 1 - \frac{1}{2s} \right) m + \frac{0.01}{s}$$

$$\ge \frac{m^2}{4 \times 0.99} - 0.1m + 0.1 \ge \frac{m^2}{4}, \ \forall m \gtrsim 1.$$

Hence, we conclude that $f_s(x) \lesssim e^{-\frac{m^2}{8}}$.

Next, we consider lower bounding the term $(\partial_s F_s(x))^2$. Note that

$$-\partial_s F_s(x) = -w'(s) \left( \Phi \left( \frac{x}{\sqrt{s}} \right) - \Phi \left( \frac{x-sm}{\sqrt{s}} \right) \right)$$

$$+ w(s) \phi \left( \frac{x}{\sqrt{s}} \right) \frac{x}{2s^{\frac{3}{2}}} + (1-w(s)) \phi \left( \frac{x-sm}{\sqrt{s}} \right) \left( \frac{x}{2s^{\frac{3}{2}}} + \frac{m}{2s^{\frac{1}{2}}} \right).$$

As $x \in \left[ \frac{m}{2} - 0.1, \frac{m}{2} + 0.1 \right]$ and $s \in [0.9, 0.99]$, all these three terms are positive. We only focus on the first term. Note the following two bounds:

$$\begin{cases} \Phi \left( \frac{x}{\sqrt{s}} \right) \ge \Phi \left( \frac{m}{2} - 0.1 \right) \ge \frac{3}{4}, & \forall m \gtrsim 1, \\ \Phi \left( \frac{x-sm}{\sqrt{s}} \right) \le \Phi \left( \frac{m/2+0.1-sm}{\sqrt{s}} \right) \le \Phi(-0.4m+0.1) \le \frac{1}{4}, & \forall m \gtrsim 1. \end{cases}$$

Therefore, we have

$$-\partial_s F_s(x) \gtrsim \frac{m^2}{e^{(1-s)m^2/2}}.$$

To summarize, we derive the following lower bound on the metric derivative:

$$|\dot{\pi}|_s^2 = \int \frac{(\partial_s F_s(x))^2}{f_s(x)} \mathrm{d}x \ge \int_{\frac{m}{2}-0.1}^{\frac{m}{2}+0.1} \frac{(\partial_s F_s(x))^2}{f_s(x)} \mathrm{d}x$$

$$\gtrsim \int_{\frac{m}{2}-0.1}^{\frac{m}{2}+0.1} \frac{m^4 e^{-(1-s)m^2}}{e^{-m^2/8}} \mathrm{d}x$$

$$\gtrsim m^4 e^{\left(s-\frac{7}{8}\right)m^2} \ge m^4 e^{\frac{m^2}{40}}, \ \forall s \in [0.9, 0.99].$$

Finally, recall that $S(\theta) := \frac{1}{1+m^2(1-\theta)^r}$, and $\pi_\theta = \pi_{S(\theta)}$. Hence, by chain rule of derivative, $|\dot{\pi}|_\theta = |\dot{\pi}|_{S(\theta)} |S'(\theta)|$. Let

$$\Theta := \{\theta \in [0,1] : S(\theta) \in [0.9, 0.99]\} = \left[ 1 - \left( \frac{1/0.9 - 1}{m^2} \right)^{\frac{1}{r}}, 1 - \left( \frac{1/0.99 - 1}{m^2} \right)^{\frac{1}{r}} \right].$$

We have

$$
\mathcal{A}_r = \int_0^1 |\dot{\pi}|_\theta^2 \mathrm{d}\theta = \int_0^1 |\underline{\dot{\pi}}|_{S(\theta)}^2 |S'(\theta)|^2 \mathrm{d}\theta \geq \int_\Theta |\underline{\dot{\pi}}|_{S(\theta)}^2 |S'(\theta)|^2 \mathrm{d}\theta
$$

$$
\geq \min_{\theta \in \Theta} |S'(\theta)| \cdot \int_\Theta |\underline{\dot{\pi}}|_{S(\theta)}^2 |S'(\theta)| \mathrm{d}\theta = \min_{\theta \in \Theta} |S'(\theta)| \cdot \int_{0.9}^{0.99} |\underline{\dot{\pi}}|_s^2 \mathrm{d}s.
$$

For any $\theta \in \Theta$,

$$
|S'(\theta)| = \frac{m^2 r (1-\theta)^{r-1}}{(1+m^2(1-\theta)^r)^2} \geq \frac{m^2 r \left(\frac{1/0.99-1}{m^2}\right)^{1-1/r}}{\left(1+m^2\left(\frac{1/0.9-1}{m^2}\right)\right)^2} = \frac{m^{2/r} r (1/99)^{1-1/r}}{(1/0.9)^2} \gtrsim m^{2/r} \gtrsim 1,
$$

where in the first "$\gtrsim$" we used the inequality $r\left(\frac{1}{99}\right)^{1-\frac{1}{r}} \geq \frac{1}{\mathrm{e}^4}$ that holds for all $r \geq 1$. Thus, the proof is complete. $\qquad\square$

**Remark 9.** *In the above theorem, we established an exponential lower bound on the metric derivative of the $W_2$ distance, given by $\lim_{\delta \to 0} \frac{W_2(\pi_s, \pi_{s+\delta})}{|\delta|}$. In OT, another useful distance, the **Wasserstein-1 ($W_1$) distance**, defined as $W_1(\mu, \nu) = \inf_{\gamma \in \Pi(\mu,\nu)} \int \|x-y\| \gamma(\mathrm{d}x, \mathrm{d}y)$, is a lower bound of the $W_2$ distance. Below, we present a surprising result regarding the metric derivative of $W_1$ distance on the same curve of probability distributions. This result reveals an exponentially large gap between the $W_1$ and $W_2$ metric derivatives on the same curve, which is of independent interest.*

**Theorem 6.** *Define the probability distributions $\pi_s$ as in the proof of Prop. 1, for some large enough $m \gtrsim 1$. Then, for all $s \in [0.9, 0.99]$, we have*

$$
\lim_{\delta \to 0} \frac{W_1(\pi_s, \pi_{s+\delta})}{|\delta|} \lesssim 1.
$$

*Proof.* Since $W_1(\mu, \nu) = \int |F_\mu(x) - F_\nu(x)| \mathrm{d}x$ (Villani, 2003, Thm. 2.18), by assuming regularity conditions, we have

$$
\lim_{\delta \to 0} \frac{W_1(\pi_s, \pi_{s+\delta})}{|\delta|} = \int |\partial_s F_s(x)| \mathrm{d}x
$$

$$
\leq \int \left| w'(s) \left( \Phi\left(\frac{x}{\sqrt{s}}\right) - \Phi\left(\frac{x-sm}{\sqrt{s}}\right) \right) \right| \mathrm{d}x
$$

$$
+ \int \left| w(s) \phi\left(\frac{x}{\sqrt{s}}\right) \frac{x}{2s^{\frac{3}{2}}} \right| \mathrm{d}x
$$

$$
+ \int \left| (1-w(s)) \phi\left(\frac{x-sm}{\sqrt{s}}\right) \left( \frac{x}{2s^{\frac{3}{2}}} + \frac{m}{2s^{\frac{1}{2}}} \right) \right| \mathrm{d}x.
$$

To bound the first term, notice that for any $\lambda > 0$,

$$
\Phi\left(\frac{x}{\sqrt{s}}\right) - \Phi\left(\frac{x-sm}{\sqrt{s}}\right) \lesssim \begin{cases} \sqrt{s}m\mathrm{e}^{-\frac{(x-sm)^2}{2s}}, & \frac{x-sm}{\sqrt{s}} \geq \lambda; \\ \sqrt{s}m\mathrm{e}^{-\frac{x^2}{2s}}, & \frac{x}{\sqrt{s}} \leq -\lambda; \\ 1, & \text{otherwise.} \end{cases}
$$

Therefore, using Gaussian tail bound $1 - \Phi(\lambda) \leq \frac{1}{2}\mathrm{e}^{-\frac{\lambda^2}{2}}$, the first term is bounded by

$$
\lesssim \frac{m^2}{\mathrm{e}^{(1-s)m^2/2}} \left[ 2\sqrt{s}\lambda + sm + sm(1-\Phi(\lambda)) + sm\Phi(-\lambda) \right]
$$

$$
\lesssim \frac{m^2}{\mathrm{e}^{(1-s)m^2/2}} [\lambda + m + \mathrm{e}^{-\frac{\lambda^2}{2}}] \overset{\lambda \leftarrow \Theta(m)}{\lesssim} \frac{m^3}{\mathrm{e}^{(1-s)m^2/2}} = o(1).
$$

The second term is bounded by

$$\lesssim \int \phi\left(\frac{x}{\sqrt{s}}\right)|x|\mathrm{d}x = s\int\phi(u)|u|\mathrm{d}u \lesssim 1.$$

Finally, the third term is bounded by

$$\lesssim \frac{1}{e^{(1-s)m^2/2}}\int\phi\left(\frac{x-sm}{\sqrt{s}}\right)(|x|+m)\mathrm{d}x$$

$$\lesssim \frac{1}{e^{(1-s)m^2/2}}\int\phi(u)(|u|+m)\mathrm{d}u \lesssim \frac{m}{e^{(1-s)m^2/2}} = o(1).$$

$\square$

### E.2 PROOF OF PROP. 2

*Proof.* We first prove a more general result with $\phi$ being *any* distribution with weak regularity condition, and then focus on the special case where $\phi = \mathcal{N}(0, I)$.

Note that the LD with target distribution $\phi$,

$$\mathrm{d}Y_t = \nabla\log\phi(Y_t)\mathrm{d}t + \sqrt{2}\mathrm{d}B_t,\ Y_t \sim \overline{\pi}_t,$$

can be perceived as the Wasserstein gradient flow of $\mathrm{KL}(\cdot\|\phi)$. $\overline{\pi}_t$ satisfies the Fokker-Planck equation $\partial_t\overline{\pi}_t = \nabla\cdot\left(\overline{\pi}_t\nabla\log\frac{\overline{\pi}_t}{\phi}\right)$. Hence, the vector field $\left(v_t := -\nabla\log\frac{\overline{\pi}_t}{\phi}\right)_{t\in[0,\infty)}$ generates $(\overline{\pi}_t)_{t\in[0,\infty)}$, and each $v_t$ can be written as a gradient field of a potential function. Thus, by the uniqueness result in Lem. 4, we conclude that

$$|\dot{\overline{\pi}}|_t^2 = \left\|\nabla\log\frac{\overline{\pi}_t}{\phi}\right\|_{L^2(\overline{\pi}_t)}^2 = \mathrm{FI}(\overline{\pi}_t\|\phi) = -\partial_t\mathrm{KL}(\overline{\pi}_t\|\phi) \implies \int_0^\infty |\dot{\overline{\pi}}|_t^2\mathrm{d}t = \mathrm{KL}(\pi\|\phi),$$

where FI is the Fisher divergence.

For the special case where $\phi = \mathcal{N}(0, I)$, using the log-Sobolev equality (Def. 3), the smoothness of $V$, and Lem. 13, we can further bound the KL divergence as follows:

$$\mathrm{KL}(\pi\|\phi) \leq \frac{1}{2}\mathrm{FI}(\pi\|\phi) = \frac{1}{2}\mathbb{E}_{\pi(x)}\|-\nabla V(x)+x\|^2 \leq \mathbb{E}_\pi\|\nabla V\|^2 + \mathbb{E}_\pi\|\cdot\|^2 \leq d\beta + m^2.$$

$\square$

### E.3 PROOF OF THM. 5

*Proof.* By Nelson's relation (Lem. 2), $\mathbb{Q}$ is equivalent to the path measure of the following SDE:

$$\mathrm{d}X_t = X_t\mathrm{d}t + \sqrt{2}\mathrm{d}B_t^\leftarrow,\ t \in [0, T-\delta];\ X_{T-\delta} \sim \overline{\pi}_\delta.$$

Leveraging Girsanov's theorem (Lem. 3), we know that for a.s. $X \sim \mathbb{Q}^\dagger$:

$$\log\frac{\mathrm{d}\mathbb{Q}^\dagger}{\mathrm{d}\mathbb{Q}}(X) = \log\frac{\phi(X_0)}{\overline{\pi}_\delta(X_{T-\delta})} + \frac{1}{2}\int_0^{T-\delta}\left(\langle X_t + 2s_{T-t_-}(X_{t_-}), \mathrm{d}X_t\rangle - \langle X_t, *\mathrm{d}X_t\rangle\right)$$

$$-\frac{1}{4}\int_0^{T-\delta}\left(\|X_t + 2s_{T-t_-}(X_{t_-})\|^2 - \|X_t\|^2\right)\mathrm{d}t.$$

Note that for $X \sim \mathbb{Q}^\dagger$, $\int_0^{T-\delta}\langle X_t, *\mathrm{d}X_t\rangle = \int_0^{T-\delta}\langle X_t, \mathrm{d}X_t\rangle + [X, X]_{T-\delta}$ and $[X, X]_{T-\delta} = [\sqrt{2}B, \sqrt{2B}]_{T-\delta} = 2(T-\delta)d$. Some simple calculations yield

$$\log\frac{\mathrm{d}\mathbb{Q}^\dagger}{\mathrm{d}\mathbb{Q}}(X) = \log\frac{\phi(X_0)}{\overline{\pi}_\delta(X_{T-\delta})} - (T-\delta)d + \int_0^{T-\delta}\left(\|s_{T-t_-}(X_{t_-})\|^2\mathrm{d}t + \sqrt{2}\langle s_{T-t_-}(X_{t_-}), \mathrm{d}B_t\rangle\right)$$

$$= \log Z + W(X) + \log\frac{\mathrm{d}\pi}{\mathrm{d}\overline{\pi}_\delta}(X_{T-\delta}).$$

Thus, the equation $\mathbb{E}_{\mathbb{Q}^\dagger} \frac{\mathrm{d}\mathbb{Q}}{\mathrm{d}\mathbb{Q}^\dagger} = 1$ implies

$$Z = \mathbb{E}_{\mathbb{Q}^\dagger(X)} \, \mathrm{e}^{-W(X)} \frac{\mathrm{d}\overline{\pi}_\delta}{\mathrm{d}\pi}(X_{T-\delta}) \approx \mathbb{E}_{\mathbb{Q}^\dagger(X)} \, \mathrm{e}^{-W(X)} = \mathbb{E} \, \widehat{Z}.$$

Since $\frac{\widehat{Z}}{Z} = \frac{\mathrm{d}\mathbb{Q}}{\mathrm{d}\mathbb{Q}^\dagger}(X) \frac{\mathrm{d}\pi}{\mathrm{d}\overline{\pi}_\delta}(X_{T-\delta})$, we have

$$\Pr\left(\left|\frac{\widehat{Z}}{Z} - 1\right| \geq \varepsilon\right) = \Pr_{X \sim \mathbb{Q}^\dagger}\left(\left|\frac{\mathrm{d}\mathbb{Q}}{\mathrm{d}\mathbb{Q}^\dagger}(X)\frac{\mathrm{d}\pi}{\mathrm{d}\overline{\pi}_\delta}(X_{T-\delta}) - 1\right| \geq \varepsilon\right)$$

$$\leq \Pr_{X \sim \mathbb{Q}^\dagger}\left(\left|\frac{\mathrm{d}\mathbb{Q}}{\mathrm{d}\mathbb{Q}^\dagger}(X) - 1\right| \gtrsim \varepsilon\right) + \Pr_{X \sim \mathbb{Q}^\dagger}\left(\left|\frac{\mathrm{d}\pi}{\mathrm{d}\overline{\pi}_\delta}(X_{T-\delta}) - 1\right| \gtrsim \varepsilon\right).$$

The inequality is due to the fact that $|ab - 1| \geq \varepsilon$ implies $|a - 1| \geq \frac{\varepsilon}{3}$ or $|b - 1| \geq \frac{\varepsilon}{3}$ for $\varepsilon \in [0, 1]$. It suffices to make both terms above $O(1)$. To bound the first term, we use the similar approach as in the proof of (18) in Thm. 2:

$$\Pr_{X \sim \mathbb{Q}^\dagger}\left(\left|\frac{\mathrm{d}\mathbb{Q}}{\mathrm{d}\mathbb{Q}^\dagger}(X) - 1\right| \gtrsim \varepsilon\right) = \mathbb{Q}^\dagger\left(\left|\frac{\mathrm{d}\mathbb{Q}}{\mathrm{d}\mathbb{Q}^\dagger} - 1\right| \gtrsim \varepsilon\right) \lesssim \frac{\mathrm{TV}(\mathbb{Q}, \mathbb{Q}^\dagger)}{\varepsilon} \lesssim \frac{\sqrt{\mathrm{KL}(\mathbb{Q} \| \mathbb{Q}^\dagger)}}{\varepsilon}.$$

Hence, it suffices to let $\mathrm{TV}(\mathbb{Q}, \mathbb{Q}^\dagger)^2 \lesssim \mathrm{KL}(\mathbb{Q} \| \mathbb{Q}^\dagger) \lesssim \varepsilon^2$. To bound the second term, we have

$$\Pr_{X \sim \mathbb{Q}^\dagger}\left(\left|\frac{\mathrm{d}\pi}{\mathrm{d}\overline{\pi}_\delta}(X_{T-\delta}) - 1\right| \gtrsim \varepsilon\right) \leq \Pr_{X \sim \mathbb{Q}}\left(\left|\frac{\mathrm{d}\pi}{\mathrm{d}\overline{\pi}_\delta}(X_{T-\delta}) - 1\right| \gtrsim \varepsilon\right) + \mathrm{TV}(\mathbb{Q}, \mathbb{Q}^\dagger)$$

$$\leq \overline{\pi}_\delta\left(\left|\frac{\mathrm{d}\pi}{\mathrm{d}\overline{\pi}_\delta} - 1\right| \gtrsim \varepsilon\right) + \mathrm{TV}(\mathbb{Q}, \mathbb{Q}^\dagger)$$

$$\lesssim \frac{\mathrm{TV}(\overline{\pi}_\delta, \pi)}{\varepsilon} + \varepsilon.$$

Therefore, it suffices to make $\mathrm{TV}(\overline{\pi}_\delta, \pi) \lesssim \varepsilon$. $\qquad\square$

### E.4 An Upper Bound of the TV Distance along the OU Process

**Lemma 7.** *Assume that the target distribution $\pi \propto \mathrm{e}^{-V}$ satisfies Assump. 2. Let $\overline{\pi}_\delta$ be the distribution of $Y_\delta$ in the OU process (12) initialized at $Y_0 \sim \pi$, for some $\delta \lesssim 1$. Then,*

$$\mathrm{TV}(\pi, \overline{\pi}_\delta) \lesssim \delta(\beta m^2 + d + \beta d) + \delta^{\frac{1}{2}} d^{\frac{1}{2}} \beta m.$$

**Remark 10.** *Consider a simplified case where $\beta \gtrsim 1$ and $m^2 \asymp d$. Then it suffices to choose $\delta \lesssim \frac{\varepsilon^2}{\beta^2 d^2}$ in order to guarantee $\mathrm{TV}(\pi, \overline{\pi}_\delta) \lesssim \varepsilon$.*

*Proof.* Our proof is inspired by Lee et al. (2023, Lem. 6.4), which addresses the case where $V$ is Lipschitz.

Without loss of generality, suppose $\pi = \mathrm{e}^{-V}$. Let $\phi$ be the p.d.f. of $\mathcal{N}(0, I)$, and define $\sigma^2 := 1 - \mathrm{e}^{-2\delta} \asymp \delta$. We will use the following inequality: $|\mathrm{e}^a - \mathrm{e}^b| \leq (\mathrm{e}^a + \mathrm{e}^b)|a - b|$, which is due to the convexity of the exponential function. By the smoothness of $V$, $\|\nabla V(x)\| = \|\nabla V(x) - \nabla V(0)\| \leq \beta \|x\|$.

Define $\pi'(x) = \mathrm{e}^{d\delta} \pi(\mathrm{e}^\delta x)$, and thus $\overline{\pi}_\delta(x) = \int \pi'(x + \sigma u)\phi(u)\mathrm{d}u$. Using triangle inequality, we bound $\mathrm{TV}(\pi, \pi')$ and $\mathrm{TV}(\pi', \overline{\pi}_\delta)$ separately. First,

$$\mathrm{TV}(\pi, \pi') = \frac{1}{2} \int |\mathrm{e}^{-V(x)} - \mathrm{e}^{-V(\mathrm{e}^\delta x) + d\delta}| \mathrm{d}x$$

$$\lesssim \int (\pi(x) + \pi'(x))(|V(\mathrm{e}^\delta x) - V(x)| + d\delta)\mathrm{d}x.$$

By the smoothness,

$$|V(\mathrm{e}^{\delta}x) - V(x)| \leq \|\nabla V(x)\|(\mathrm{e}^{\delta} - 1)\|x\| + \frac{\beta}{2}(\mathrm{e}^{\delta} - 1)^2\|x\|^2$$
$$\lesssim \beta\|x\|\delta\|x\| + \beta\delta^2\|x\|^2 \lesssim \beta\delta\|x\|^2.$$
$$\implies \mathrm{TV}(\pi, \pi') \lesssim \delta \int (\pi(x) + \pi'(x))(\beta\|x\|^2 + d)\mathrm{d}x.$$

Note that $\int \pi(x)(\beta\|x\|^2 + d)\mathrm{d}x = \beta m^2 + d$. Since $\mathbb{E}_{\pi'} \varphi = \mathbb{E}_{\pi} \varphi(\mathrm{e}^{-\delta}\cdot)$, we also have

$$\int \pi'(x)(\beta\|x\|^2 + d)\mathrm{d}x = \mathrm{e}^{-2\delta}\beta m^2 + d \leq \beta m^2 + d.$$

We thus conclude that

$$\mathrm{TV}(\pi, \pi') \lesssim \delta(\beta m^2 + d).$$

Next,

$$\mathrm{TV}(\pi', \overline{\pi}_{\delta}) = \frac{1}{2}\int \left| \int (\pi'(x + \sigma u) - \pi'(x))\phi(u)\mathrm{d}u \right| \mathrm{d}x$$
$$\lesssim \iint |\pi'(x + \sigma u) - \pi'(x)|\phi(u)\mathrm{d}u\mathrm{d}x$$
$$\lesssim \iint (\pi'(x + \sigma u) + \pi'(x))|V(\mathrm{e}^{\delta}(x + \sigma u)) - V(\mathrm{e}^{\delta}x)|\phi(u)\mathrm{d}u\mathrm{d}x.$$

Again, by smoothness,

$$V(\mathrm{e}^{\delta}(x + \sigma u)) - V(\mathrm{e}^{\delta}x) \leq \|\nabla V(\mathrm{e}^{\delta}x)\|\mathrm{e}^{\delta}\sigma\|u\| + \frac{\beta}{2}\mathrm{e}^{2\delta}\sigma^2\|u\|^2$$
$$\lesssim \beta\mathrm{e}^{\delta}\|x\|\mathrm{e}^{\delta}\sigma\|u\| + \beta\mathrm{e}^{2\delta}\sigma^2\|u\|^2$$
$$\lesssim \beta\|x\|\delta^{\frac{1}{2}}\|u\| + \beta\delta\|u\|^2.$$

Therefore,

$$\mathrm{TV}(\pi', \overline{\pi}_{\delta}) \lesssim \beta\delta^{\frac{1}{2}} \iint (\pi'(x + \sigma u) + \pi'(x))(\|u\|\|x\| + \delta^{\frac{1}{2}}\|u\|^2)\phi(u)\mathrm{d}u\mathrm{d}x.$$

Note that, first,

$$\iint \pi'(x)(\|u\|\|x\| + \delta^{\frac{1}{2}}\|u\|^2)\phi(u)\mathrm{d}u\mathrm{d}x \lesssim \mathbb{E}_{\pi'} \|\cdot\| d^{\frac{1}{2}} + \delta^{\frac{1}{2}}d \leq m d^{\frac{1}{2}} + \delta^{\frac{1}{2}}d;$$

second,

$$\iint \pi'(x + \sigma u)(\|u\|\|x\| + \delta^{\frac{1}{2}}\|u\|^2)\phi(u)\mathrm{d}u\mathrm{d}x$$
$$= \iint \pi'(y)(\|u\|\|y - \sigma u\| + \delta^{\frac{1}{2}}\|u\|^2)\phi(u)\mathrm{d}u\mathrm{d}y$$
$$\lesssim \iint \pi'(y)(\|u\|\|y\| + \delta^{\frac{1}{2}}\|u\|^2)\phi(u)\mathrm{d}u\mathrm{d}y \lesssim m d^{\frac{1}{2}} + \delta^{\frac{1}{2}}d.$$

Therefore, $\mathrm{TV}(\pi', \overline{\pi}_{\delta}) \lesssim \beta\delta^{\frac{1}{2}}d^{\frac{1}{2}}(m + \delta^{\frac{1}{2}}d^{\frac{1}{2}})$. The proof is complete. $\square$

### E.5 DISCUSSION ON THE OVERALL COMPLEXITY OF RDS

In RDS, an accurate score estimate $s. \approx \nabla \log \overline{\pi}.$ is critical for the algorithmic efficiency. Existing methods estimate scores through different ways. Here, we review the existing methods and their complexity guarantees for sampling, and leverage Thm. 5 to derive the complexity of normalizing constant estimation. Throughout this section, we always assume that the target distribution $\pi \propto \mathrm{e}^{-V}$ satisfies $m^2 := \mathbb{E}_{\pi} \|\cdot\|^2 < \infty$ and that $V$ is $\beta$-smooth.

**(I) Reverse diffusion Monte Carlo.** The seminal work directly leveraged the following Tweedie's formula (Robbins, 1992) to estimate the score:

$$\nabla \log \overline{\pi}_t(x) = \mathbb{E}_{\overline{\pi}_{0|t}(x_0|x)} \frac{\mathrm{e}^{-t} x_0 - x}{1 - \mathrm{e}^{-2t}}, \tag{33}$$

where

$$\overline{\pi}_{0|t}(x_0|x) \propto_{x_0} \exp\left(-V(x_0) - \frac{\|x_0 - \mathrm{e}^t x\|^2}{2(\mathrm{e}^{2t} - 1)}\right) \tag{34}$$

is the posterior distribution of $Y_0$ conditional on $Y_t = x$ in the OU process (12). The paper proposed to sample from $\overline{\pi}_{0|t}(\cdot|x)$ by LMC and estimate the score via empirical mean, which has a provably better LSI constant than the target distribution $\pi$ (see Huang et al. (2024a, Lem. 2)). They show that if the scores $\nabla \log \overline{\pi}_t$ are uniformly $\beta$-Lipschitz, and assume that there exists some $c > 0$ and $n > 0$ such that for any $r > 0$, $V + r\|\cdot\|^2$ is convex for $\|x\| \geq \frac{c}{r^n}$, then w.p. $\geq 1 - \zeta$, the overall complexity for guaranteeing $\mathrm{KL}(\mathbb{Q}\|\mathbb{Q}^\dagger) \lesssim \varepsilon^2$ is

$$O\left(\operatorname{poly}\left(d, \frac{1}{\zeta}\right) \exp\left(\frac{1}{\varepsilon}\right)^{O(n)}\right),$$

which is also the complexity of obtaining a $\widehat{Z}$ satisfying (1).

**(II) Recursive score diffusion-based Monte Carlo.** A second work Huang et al. (2024b) proposed to estimate the scores in a recursive scheme. Assuming the scores $\nabla \log \overline{\pi}_t$ are uniformly $\beta$-Lipschitz, they established a complexity

$$\exp\left(\beta^3 \log^3 \operatorname{poly}\left(\beta, d, m^2, \frac{1}{\zeta}\right)\right)$$

in order to guarantee $\mathrm{KL}(\mathbb{Q}\|\mathbb{Q}^\dagger) \lesssim \varepsilon^2$ w.p. $\geq 1 - \zeta$.

**(III) Zeroth-order diffusion Monte Carlo.** The following work He et al. (2024) proposed a zeroth-order method that leverages rejection sampling to sample from $\overline{\pi}_{0|t}(\cdot|x)$. When $V$ is $\beta$-smooth, they showed that with a small early stopping time $\delta$, the overall complexity for guaranteeing $\mathrm{KL}(\mathbb{Q}\|\mathbb{Q}^\dagger) \lesssim \varepsilon^2$ is

$$\exp\left(\widetilde{O}(d) \log \beta \log \frac{1}{\varepsilon}\right).$$

**(IV) Self-normalized estimator.** Finally, a recent work Vacher et al. (2025) proposed to estimate the scores in a different approach:

$$\nabla \log \overline{\pi}_t(x) = -\frac{1}{1 - \mathrm{e}^{-2t}} \frac{\mathbb{E}[\xi \mathrm{e}^{-V(\mathrm{e}^t(x - \xi))}]}{\mathbb{E}[\mathrm{e}^{-V(\mathrm{e}^t(x - \xi))}]}, \quad \text{where } \xi \sim \mathcal{N}\left(0, (1 - \mathrm{e}^{-2t})I\right).$$

Assume that $V$ is $\beta$-smooth, and the distributions along the OU process starting from $\pi \propto \mathrm{e}^{-V}$ and $\pi' \propto \mathrm{e}^{-2V}$ have potentials whose Hessians are uniformly $\succeq -\beta I$, then the complexity for guaranteeing $\mathbb{E}\,\mathrm{KL}(\mathbb{Q}\|\mathbb{Q}^\dagger) \lesssim \varepsilon^2$ is

$$O\left(\left(\frac{\beta(m^2 \vee d)}{\varepsilon}\right)^{O(d)}\right).$$

# F  SUPPLEMENTARY LEMMAS

**Lemma 8.** *For $x > 0$ and $\varepsilon \in \left(0, \frac{1}{2}\right)$, define $x_0 := |\log x|$ and $x_1 := |x - 1|$. Then $x_i \geq \varepsilon$ implies $x_{1-i} \geq \frac{\varepsilon}{2}$, and $x_i \leq \varepsilon$ implies $x_{1-i} \leq 2\varepsilon$, for both $i = 0, 1$.*

This follows from the standard calculus approximation $\log x \approx x - 1$ when $x \approx 1$. The proof is straightforward and is left as an exercise for the reader.

**Lemma 9.** *For any $0 \leq a \leq b \leq 1$ and $r \geq 1$, $b^r - a^r \leq r(b - a)$.*

*Proof.* This is immediate from the decreasing property of the function $\varphi(x) := x^r - rx$, $x \in [0, 1]$, since $\varphi'(x) = r(x^{r-1} - 1) \le 0$. $\square$

**Lemma 10** (The median trick (Jerrum et al., 1986)). *Let $\widehat{Z}_1, ..., \widehat{Z}_N$ be $N(\ge 3)$ i.i.d. random variables satisfying*

$$\Pr\left(\left|\frac{\widehat{Z}_n}{Z} - 1\right| \le \varepsilon\right) \ge \frac{3}{4}, \ \forall n \in [\![1, N]\!],$$

*and let $\widehat{Z}_*$ be the median of $\widehat{Z}_1, ..., \widehat{Z}_N$. Then*

$$\Pr\left(\left|\frac{\widehat{Z}_*}{Z} - 1\right| \le \varepsilon\right) \ge 1 - e^{-\frac{N}{72}}.$$

*In particular, for any $\zeta \in \left(0, \frac{1}{4}\right)$, choosing $N = \left\lceil 72 \log \frac{1}{\zeta} \right\rceil$, the probability is at least $1 - \zeta$.*

*Proof.* Let $A_n := \left\{\left|\frac{\widehat{Z}_n}{Z} - 1\right| > \varepsilon\right\}$, which are i.i.d. events happening w.p. $p \le \frac{1}{4}$. If $\left|\frac{\widehat{Z}_*}{Z} - 1\right| > \varepsilon$, then there are at least $\left\lfloor\frac{N}{2}\right\rfloor$ $A_n$'s happening, i.e., $S_N := \sum_{n=1}^N 1_{A_n} \ge \left\lfloor\frac{N}{2}\right\rfloor$. Then,

$$\Pr\left(\left|\frac{\widehat{Z}_*}{Z} - 1\right| > \varepsilon\right) \le \Pr\left(S_N \ge \left\lfloor\frac{N}{2}\right\rfloor\right) = \Pr\left(S_N - \mathbb{E}\,S_N \ge \left\lfloor\frac{N}{2}\right\rfloor - pN\right)$$

$$\le \Pr\left(S_N - \mathbb{E}\,S_N \ge \frac{N}{12}\right) \le e^{-\frac{N}{72}},$$

where the first inequality on the second line follows from the fact that $\left\lfloor\frac{N}{2}\right\rfloor \ge \frac{N-1}{2} \ge \frac{N}{3}$ for all $N \ge 3$, and the last inequality is due to the Hoeffding's inequality. $\square$

**Lemma 11.** *The update rule of AIS (10) is:*

$$X_{T_\ell} = e^{-\Lambda(T_\ell)}X_0 - \left(\int_0^{T_\ell} e^{-(\Lambda(T_\ell)-\Lambda(t))}\mathrm{d}t\right)\nabla V(X_0) + \left(2\int_0^{T_\ell} e^{-2(\Lambda(T_\ell)-\Lambda(t))}\mathrm{d}t\right)^{\frac{1}{2}}\xi,$$

*where $\Lambda(t) := \int_0^t \lambda\left(\theta_{\ell-1} + \frac{\tau}{T_\ell}(\theta_\ell - \theta_{\ell-1})\right)\mathrm{d}\tau$, and $\xi \sim \mathcal{N}(0, I)$ is independent of $X_0$.*

*Proof.* By Itô's formula, we have

$$\mathrm{d}(e^{\Lambda(t)}X_t) = e^{\Lambda(t)}\left(\Lambda'(t)X_t\mathrm{d}t + \mathrm{d}X_t\right) = e^{\Lambda(t)}(-\nabla V(X_0)\mathrm{d}t + \sqrt{2}\mathrm{d}B_t).$$

Integrating over $t \in [0, T_\ell]$, we obtain

$$e^{\Lambda(T_\ell)}X_{T_\ell} - X_0 = -\left(\int_0^{T_\ell} e^{\Lambda(t)}\mathrm{d}t\right)\nabla V(X_0) + \sqrt{2}\int_0^{T_\ell} e^{\Lambda(t)}\mathrm{d}B_t,$$

$$\implies X_{T_\ell} = e^{-\Lambda(T_\ell)}X_0 - \left(\int_0^{T_\ell} e^{-(\Lambda(T_\ell)-\Lambda(t))}\mathrm{d}t\right)\nabla V(X_0) + \sqrt{2}\int_0^{T_\ell} e^{-(\Lambda(T_\ell)-\Lambda(t))}\mathrm{d}B_t,$$

and $\sqrt{2}\int_0^{T_\ell} e^{-(\Lambda(T_\ell)-\Lambda(t))}\mathrm{d}B_t \sim \mathcal{N}\left(0, \left(2\int_0^{T_\ell} e^{-2(\Lambda(T_\ell)-\Lambda(t))}\mathrm{d}t\right)I\right)$ by Itô isometry. $\square$

**Lemma 12.** *The update rule of the RDS (14) is*

$$X_{t_{k+1}} = e^{t_{k+1}-t_k}X_{t_k} + 2(e^{t_{k+1}-t_k} - 1)s_{T-t_k}(X_{t_k}) + \Xi_k,$$

*where*

$$\Xi_k := \int_{t_k}^{t_{k+1}} \sqrt{2}e^{-(t-t_{k+1})}\mathrm{d}B_t \sim \mathcal{N}\left(0, (e^{2(t_{k+1}-t_k)} - 1)I\right),$$

*and the correlation matrix between $\Xi_k$ and $B_{t_{k+1}} - B_{t_k}$ is*

$$\mathrm{Corr}(\Xi_k, B_{t_{k+1}} - B_{t_k}) = \frac{\sqrt{2}(e^{t_{k+1}-t_k} - 1)}{\sqrt{(e^{2(t_{k+1}-t_k)} - 1)(t_{k+1} - t_k)}}I.$$

*Proof.* By applying Itô's formula to (14) for $t \in [t_k, t_{k+1}]$, we have

$$d(e^{-t}X_t) = e^{-t}(-X_t dt + dX_t) = e^{-t}(2s_{T-t_k}(X_{t_k})dt + \sqrt{2}dB_t)$$

$$\implies e^{-t_{k+1}}X_{t_{k+1}} - e^{-t_k}X_{t_k} = 2(e^{-t_k} - e^{-t_{k+1}})s_{T-t_k}(X_{t_k}) + \int_{t_k}^{t_{k+1}} \sqrt{2}e^{-t}dB_t.$$

The covariance between two zero-mean Gaussian random variables $\Xi_k$ and $B_{t_{k+1}} - B_{t_k}$ is

$$\begin{aligned}
\mathrm{Cov}(\Xi_k, B_{t_{k+1}} - B_{t_k}) &= \mathbb{E}\left[\Xi_k(B_{t_{k+1}} - B_{t_k})^{\mathrm{T}}\right] \\
&= \mathbb{E}\left[\left(\int_{t_k}^{t_{k+1}} \sqrt{2}e^{-(t-t_{k+1})}dB_t\right)\left(\int_{t_k}^{t_{k+1}} dB_t\right)^{\mathrm{T}}\right] \\
&= \int_{t_k}^{t_{k+1}} \sqrt{2}e^{-(t-t_{k+1})}dt \cdot I = \sqrt{2}(e^{t_{k+1}-t_k} - 1)I.
\end{aligned}$$

Finally, $\mathrm{Corr}(u, v) = \mathrm{diag}(\mathrm{Cov}\, u)^{-\frac{1}{2}} \mathrm{Cov}(u, v) \mathrm{diag}(\mathrm{Cov}\, v)^{-\frac{1}{2}}$ yields the correlation. $\qquad\square$

**Lemma 13** (Chewi (2022, Lemma 4.E.1)). *Consider a probability measure $\mu \propto e^{-U}$ on $\mathbb{R}^d$.*

*1. If $\nabla^2 U \succeq \alpha I$ for some $\alpha > 0$ and $x_\star$ is the global minimizer of $U$, then $\mathbb{E}_\mu \|\cdot - x_\star\|^2 \leq \frac{d}{\alpha}$.*
*2. If $\nabla^2 U \preceq \beta I$ for some $\beta > 0$, then $\mathbb{E}_\mu \|\nabla U\|^2 \leq \beta d$.*

**Lemma 14.** *Define $\widehat{\pi}_\lambda \propto \exp\left(-V - \frac{\lambda}{2}\|\cdot\|^2\right)$, $\lambda \geq 0$. Then under Assump. 2, $\mathbb{E}_{\widehat{\pi}_\lambda}\|\cdot\|^2 \leq m^2$ for all $\lambda \geq 0$.*

*Proof.* Let $V_\lambda := V + \frac{\lambda}{2}\|\cdot\|^2$, and $Z_\lambda = \int e^{-V_\lambda}dx$, so $\widehat{\pi}_\lambda = e^{-V_\lambda - \log Z_\lambda}$. We have

$$\frac{d}{d\lambda} \log Z_\lambda = \frac{Z'_\lambda}{Z_\lambda} = -\frac{1}{Z_\lambda} \int e^{-V_\lambda} V'_\lambda dx = -\frac{1}{2}\mathbb{E}_{\widehat{\pi}_\lambda}\|\cdot\|^2,$$

$$\implies \frac{d}{d\lambda} \log \widehat{\pi}_\lambda = -V'_\lambda - \frac{d}{d\lambda} \log Z_\lambda = \frac{1}{2}\left(\mathbb{E}_{\widehat{\pi}_\lambda}\|\cdot\|^2 - \|\cdot\|^2\right),$$

$$\implies \frac{d}{d\lambda} \mathbb{E}_{\widehat{\pi}_\lambda}\|\cdot\|^2 = \int \|\cdot\|^2 \left(\frac{d}{d\lambda} \log \widehat{\pi}_\lambda\right) d\widehat{\pi}_\lambda = \frac{1}{2}\left(\left(\mathbb{E}_{\widehat{\pi}_\lambda}\|\cdot\|^2\right)^2 - \mathbb{E}_{\widehat{\pi}_\lambda}\|\cdot\|^4\right) \leq 0.$$

$\qquad\square$

# G  REVIEW AND DISCUSSION ON THE ERROR GUARANTEE (1)

## G.1  LITERATURE REVIEW OF EXISTING BOUNDS

**Estimation of $Z$.** Traditionally, the statistical properties of an estimator are typically analyzed through its bias and variance. However, deriving closed-form expressions of the variance of $\widehat{Z}$ and $\widehat{F}$ in JE remains challenging. Recall that the estimator $\widehat{Z} = Z_0 e^{-W(X)}$, $X \sim \mathbb{P}^{\rightarrow}$ for $Z = Z_0 e^{-\Delta F}$, and that JE implies $\mathrm{Bias}\, \widehat{Z} = 0$. For general (sub-optimally) controlled SDEs, Hartmann & Richter (2024) established both upper and lower bounds of the relative error of the importance sampling estimator, yet bounds tailored for JE are not well-studied. Inspired by this, we establish an upper bound on the *normalized variance* $\mathrm{Var}\, \frac{\widehat{Z}}{Z}$ in Prop. 3 at the end of this section using techniques in Rényi divergence. However, we remark that connecting this upper bound to the properties of the curve (e.g., action) is non-trivial, which we leave for future work.

**Estimation of $F$.** Turning to the estimator $\widehat{F} = -\log \widehat{Z}$ for $F = -\log Z$, we have

$$\mathrm{Bias}\, \widehat{F} = \mathbb{E}_{\mathbb{P}^{\rightarrow}} W - \Delta F = \mathcal{W} - \Delta F = \mathcal{W}_{\mathrm{diss}}.$$

Bounding the average dissipated work $\mathcal{W}_{\mathrm{diss}} = \mathrm{KL}(\mathbb{P}^{\rightarrow}\|\mathbb{P}^{\leftarrow}) = -\mathbb{E}_{\mathbb{P}^{\rightarrow}} \int_0^T (\partial_t \log \widetilde{\pi}_t)(X_t)dt$ remains challenging as well, as the law of $X_t$ under $\mathbb{P}^{\rightarrow}$ is unknown, thus complicating the bounding of the expectation. To the best of our knowledge, Chen et al. (2020) established a lower bound in terms

of $W_2(\pi_0, \pi_1)$ via the Wasserstein gradient flow, but an upper bound remains elusive. Furthermore, $\mathbb{E}\,\widehat{F}^2 = \mathbb{E}_{\mathbb{P}^{\to}(X)}\left(\log Z_0 - W(X)\right)^2$ is similarly intractable to analyze.

For multiple estimators, i.e., $\widehat{F}_K := -\log\left(Z_0 \frac{1}{M}\sum_{k=1}^{K} e^{-W(X^{(k)})}\right)$ where $X^{(1)}, ..., X^{(K)} \overset{\text{i.i.d.}}{\sim} \mathbb{P}^{\to}$, Zuckerman & Woolf (2002; 2004) (see also Lelièvre et al. (2010, Sec. 4.1.5)) derived approximate asymptotic bounds on $\text{Bias}\,\widehat{F}_K$ and $\text{Var}\,\widehat{F}_K$ via the delta method (or equivalently, the central limit theorem and Taylor expansions). Precise and non-asymptotic bounds remain elusive to date.

## G.2 Equivalence in Complexities for Estimating $Z$ and $F$

We prove the claim in Rmk. 1 that estimating $Z$ with $O(\varepsilon)$ relative error and estimating $F$ with $O(\varepsilon)$ absolute error share the same complexity up to absolute constants. This follows directly from Lem. 8: for any $\varepsilon \in \left(0, \frac{1}{2}\right)$,

$$(1) \implies \Pr\left(|\widehat{F} - F| \le 2\varepsilon\right) \ge \frac{3}{4}, \quad \text{and} \quad (1) \impliedby \Pr\left(|\widehat{F} - F| \le \frac{\varepsilon}{2}\right) \ge \frac{3}{4}.$$

## G.3 (1) is Weaker than Bias and Variance

We demonstrate that (1) is a weaker criterion than controlling bias and variance, which is an immediate result from the Chebyshev inequality:

$$\Pr\left(\left|\frac{\widehat{Z}}{Z} - 1\right| \ge \varepsilon\right) \le \frac{1}{\varepsilon^2}\,\mathbb{E}\left(\frac{\widehat{Z}}{Z} - 1\right)^2 = \frac{\text{Bias}^2\,\widehat{Z} + \text{Var}\,\widehat{Z}}{\varepsilon^2 Z^2},$$

$$\Pr\left(|\widehat{F} - F| \ge \varepsilon\right) \le \frac{\mathbb{E}(\widehat{F} - F)^2}{\varepsilon^2} = \frac{\text{Bias}^2\,\widehat{F} + \text{Var}\,\widehat{F}}{\varepsilon^2}.$$

On the other hand, suppose one has established a bound in the following form:

$$\Pr\left(\left|\frac{\widehat{Z}}{Z} - 1\right| \ge \varepsilon\right) \le p(\varepsilon), \quad \text{for some } p : [0, \infty) \to [0, 1],$$

and assume that $\widehat{Z}$ is unbiased. Then this implies

$$\text{Var}\,\frac{\widehat{Z}}{Z} = \mathbb{E}\left(\frac{\widehat{Z}}{Z} - 1\right)^2 = \int_0^\infty \Pr\left(\left(\frac{\widehat{Z}}{Z} - 1\right)^2 \ge \varepsilon\right)\,d\varepsilon \le \int_0^\infty p(\sqrt{\varepsilon})\,d\varepsilon.$$

## G.4 An Upper Bound on the Normalized Variance of $\widehat{Z}$ in Jarzynski Equality

**Proposition 3.** *Under the setting of JE (Thm. 1), let $(v_t)_{t \in [0,T]}$ be any vector field that generates $(\widetilde{\pi}_t)_{t \in [0,T]}$, and define $\mathbb{P}$ as the path measure of (17). Then,*

$$\text{Var}\,\frac{\widehat{Z}}{Z} \le \left[\mathbb{E}_{\mathbb{P}}\exp\left(14\int_0^T \|v_t(X_t)\|^2\,dt\right)\right]^{\frac{1}{2}} - 1.$$

*Proof.* The proof is inspired by Chewi et al. (2022). Note that

$$\text{Var}\,\frac{\widehat{Z}}{Z} = \mathbb{E}\left(\frac{\widehat{Z}}{Z}\right)^2 - 1 = \mathbb{E}_{\mathbb{P}^{\to}}\left(e^{-W(X)+\Delta F}\right)^2 - 1 = \mathbb{E}_{\mathbb{P}^{\to}}\left(\frac{d\mathbb{P}^{\leftarrow}}{d\mathbb{P}^{\to}}\right)^2 - 1,$$

which is the $\chi^2$ divergence from $\mathbb{P}^{\leftarrow}$ to $\mathbb{P}^{\to}$. Recall the $q(> 1)$-Rényi divergence defined as $R_q(\mu\|\nu) = \frac{1}{q-1}\log\mathbb{E}_\nu\left(\frac{d\mu}{d\nu}\right)^q$, and that $\chi^2(\mathbb{P}^{\leftarrow}\|\mathbb{P}^{\to}) = e^{R_2(\mathbb{P}^{\leftarrow}\|\mathbb{P}^{\to})} - 1$. By the weak triangle inequality of Rényi divergence (Chewi, 2022, Lem. 6.2.5):

$$R_2(\mathbb{P}^{\leftarrow}\|\mathbb{P}^{\to}) \le \frac{3}{2}R_4(\mathbb{P}^{\leftarrow}\|\mathbb{P}) + R_3(\mathbb{P}\|\mathbb{P}^{\to}).$$

We now bound $\mathbb{E}_{\mathbb{P}}\left(\frac{d\mathbb{P}^{\rightarrow}}{d\mathbb{P}}\right)^q$ for any $q \in \mathbb{R}$. By Girsanov's theorem (Lem. 1),

$$\log \frac{d\mathbb{P}^{\rightarrow}}{d\mathbb{P}}(X) = \int_0^T \left(-\frac{1}{\sqrt{2}}\langle v_t(X_t), dB_t \rangle - \frac{1}{4}\|v_t(X_t)\|^2 dt\right), \text{ a.s. } X \sim \mathbb{P}.$$

Therefore,

$$\mathbb{E}_{\mathbb{P}}\left(\frac{d\mathbb{P}^{\rightarrow}}{d\mathbb{P}}\right)^q$$

$$= \mathbb{E}_{\mathbb{P}}\exp\int_0^T\left(-\frac{q}{\sqrt{2}}\langle v_t(X_t), dB_t\rangle - \frac{q}{4}\|v_t(X_t)\|^2 dt\right)$$

$$= \mathbb{E}_{\mathbb{P}}\exp\left[\int_0^T\left(-\frac{q}{\sqrt{2}}\langle v_t(X_t), dB_t\rangle - \frac{q^2}{2}\|v_t(X_t)\|^2 dt\right) + \int_0^T\left(\frac{q^2}{2} - \frac{q}{4}\right)\|v_t(X_t)\|^2 dt\right]$$

$$\leq \left(\mathbb{E}_{\mathbb{P}}\exp\left[\int_0^T\left(-\sqrt{2}q\langle v_t(X_t), dB_t\rangle - q^2\|v_t(X_t)\|^2 dt\right)\right]\right)^{\frac{1}{2}}$$

$$\cdot \left(\mathbb{E}_{\mathbb{P}}\exp\left[\left(q^2 - \frac{q}{2}\right)\int_0^T\|v_t(X_t)\|^2 dt\right]\right)^{\frac{1}{2}},$$

where the last line is by the Cauchy-Schwarz inequality. Let $M_t := -\sqrt{2}q\int_0^t\langle v_r(X_r), dB_r\rangle$, $X \sim \mathbb{P}$ be a continuous martingale with quadratic variation $[M]_t = \int_0^t 2q^2\|v_r(X_r)\|^2 dr$. By Karatzas & Shreve (1991, Chap. 3.5.D), the process $t \mapsto e^{M_t - \frac{1}{2}[M]_t}$ is a super martingale, and hence $\mathbb{E}\,e^{M_T - \frac{1}{2}[M]_T} \leq 1$. Thus, we have

$$\mathbb{E}_{\mathbb{P}}\left(\frac{d\mathbb{P}^{\rightarrow}}{d\mathbb{P}}\right)^q \leq \left(\mathbb{E}_{\mathbb{P}}\exp\left[\left(q^2 - \frac{q}{2}\right)\int_0^T\|v_t(X_t)\|^2 dt\right]\right)^{\frac{1}{2}}$$

From Girsanov's theorem (Lem. 3), we can similarly obtain the following RN derivative:

$$\log \frac{d\mathbb{P}^{\leftarrow}}{d\mathbb{P}}(X) = \int_0^T\left(-\frac{1}{\sqrt{2}}\langle v_t(X_t), *dB_t^{\leftarrow}\rangle - \frac{1}{4}\|v_t(X_t)\|^2 dt\right), \text{ a.s. } X \sim \mathbb{P}.$$

and use the same argument to show that $\mathbb{E}_{\mathbb{P}}\left(\frac{d\mathbb{P}^{\leftarrow}}{d\mathbb{P}}\right)^q$ has exactly the same upper bound as $\mathbb{E}_{\mathbb{P}}\left(\frac{d\mathbb{P}^{\rightarrow}}{d\mathbb{P}}\right)^q$. In particular, we can use the same martingale argument, whereas now the *backward* continuous martingale is defined as $M_t' := -\sqrt{2}q\int_t^T\langle v_r(X_r), *dB_r^{\leftarrow}\rangle$, $X \sim \mathbb{P}$, with quadratic variation $[M']_t = \int_t^T 2q^2\|v_r(X_r)\|^2 dr$. Therefore, we conclude that

$$R_2(\mathbb{P}^{\leftarrow}\|\mathbb{P}^{\rightarrow}) \leq \frac{1}{4}\log\mathbb{E}_{\mathbb{P}}\exp\left(14\int_0^T\|v_t(X_t)\|^2 dt\right) + \frac{1}{4}\log\mathbb{E}_{\mathbb{P}}\exp\left(5\int_0^T\|v_t(X_t)\|^2 dt\right)$$

$$\leq \frac{1}{2}\log\mathbb{E}_{\mathbb{P}}\exp\left(14\int_0^T\|v_t(X_t)\|^2 dt\right).$$

$\square$

## H    RELATED WORKS

### H.1    THERMODYNAMIC INTEGRATION

**(I) Review of TI.**    We first briefly review the thermodynamic integration (TI) algorithm. Its essence is to write the free-energy difference as an integral of the derivative of free energy. Consider the general curve of probability measures $(\pi_\theta)_{\theta \in [0,1]}$ defined in (7). Then,

$$\frac{d}{d\theta}\log Z_\theta = -\frac{1}{Z_\theta}\int e^{-V_\theta(x)}\partial_\theta V_\theta(x)dx = -\mathbb{E}_{\pi_\theta}\partial_\theta V_\theta \implies \log\frac{Z}{Z_0} = -\int_0^1\mathbb{E}_{\pi_\theta}\partial_\theta V_\theta d\theta. \quad (35)$$

One may choose time points $0 = \theta_0 < ... < \theta_M = 1$ and approximate (35) by a Riemann sum:

$$\log \frac{Z}{Z_0} \approx -\sum_{\ell=0}^{M-1} (\theta_{\ell+1} - \theta_\ell) \, \mathbb{E}_{\pi_{\theta_\ell}} \, \partial_\theta|_{\theta=\theta_\ell} V_\theta, \tag{36}$$

where the expectation under each $\pi_{\theta_\ell}$ can be estimated by sampling from $\pi_{\theta_\ell}$. Nevertheless, there is a way of writing the exact equality instead of the approximation in (36): since

$$\log \frac{Z_{\theta_{\ell+1}}}{Z_{\theta_\ell}} = \log \int \frac{1}{Z_{\theta_\ell}} e^{-V_{\theta_\ell}(x)} e^{-(V_{\theta_{\ell+1}}(x) - V_{\theta_\ell}(x))} \mathrm{d}x = \log \mathbb{E}_{\pi_{\theta_\ell}} e^{-(V_{\theta_{\ell+1}} - V_{\theta_\ell})},$$

by summing over $\ell = 0, ..., M-1$, we have

$$\log \frac{Z}{Z_0} = \sum_{\ell=0}^{M-1} \log \mathbb{E}_{\pi_{\theta_\ell}} e^{-(V_{\theta_{\ell+1}} - V_{\theta_\ell})}, \tag{37}$$

which constitutes the estimation framework used in Brosse et al. (2018); Ge et al. (2020); Chehab et al. (2023); Kook & Vempala (2025). Hence, we also use TI to name this algorithm.

**(II) TI as a special case of AIS.** We follow the notations used in Thm. 3 to demonstrate the following claim: *TI (37) is a special case of AIS with every transition kernel $F_\ell(x, \cdot)$ chosen as the perfect proposal $\pi_{\theta_\ell}$.*

*Proof.* In AIS, with $F_\ell(x, \cdot) = \pi_{\theta_\ell}$ in the forward path $\mathbb{P}^\to$, we have $\mathbb{P}^\to(x_{0:M}) = \prod_{\ell=0}^{M} \pi_{\theta_\ell}(x_\ell)$. In this special case,

$$W(x_{0:M}) = \log \prod_{\ell=0}^{M-1} \frac{e^{-V_{\theta_\ell}(x_\ell)}}{e^{-V_{\theta_{\ell+1}}(x_\ell)}},$$

and hence the AIS equality becomes the following identity, exactly the same as (35):

$$\frac{Z}{Z_0} = e^{-\Delta\mathcal{F}} = \mathbb{E}_{\mathbb{P}^\to} e^{-W} = \prod_{\ell=0}^{M-1} \mathbb{E}_{\pi_{\theta_\ell}} e^{-(V_{\theta_{\ell+1}} - V_{\theta_\ell})}. \tag{38}$$

$\square$

**(III) The distinction between *equilibrium* and *non-equilibrium* methods.** In our AIS framework, the distinction lies in the choice of the transition kernels $F_\ell(x, \cdot)$ within the AIS framework.

In equilibrium methods, the transition kernels are ideally set to the perfect proposal $\pi_{\theta_\ell}$. However, in practice, exact sampling from $\pi_{\theta_\ell}$ is generally infeasible. Instead, one can apply multiple MCMC iterations targeting $\pi_{\theta_\ell}$, leveraging the mixing properties of MCMC algorithms to gradually approach the desired distribution $\pi_{\theta_\ell}$. Nonetheless, unless using exact sampling methods such as rejection sampling – which is exponentially expensive in high dimensions – the resulting sample distribution inevitably remains biased with a finite number of MCMC iterations.

In contrast, non-equilibrium methods employ transition kernels specifically designed to transport $\pi_{\ell-1}$ toward $\pi_\ell$, often following a curve of probability measures. This distinguishes them as inherently non-equilibrium. A key advantage of this approach over the equilibrium one is its ability to provide unbiased estimates, as demonstrated in JE and AIS.

**(IV) Complexity bounds for TI.** For the TI algorithm in Alg. 1 used to estimate $Z_0 = \int e^{-V - \beta\|\cdot\|^2} \mathrm{d}x$, the analysis Ge et al. (2020) indicates that it suffices to choose $K = \widetilde{\Theta}(\sqrt{d})$ intermediate distributions and $N = \widetilde{\Theta}\left(\frac{\sqrt{d}}{\varepsilon^2}\right)$ particles with multilevel estimation, which leads to a total complexity of $\widetilde{O}\left(\frac{d^{\frac{4}{3}}}{\varepsilon^2}\right)$ to achieve the requirement in (23) (note that the condition number of the potential $V + \beta\|\cdot\|^2$ is $O(1)$).

## H.2 PATH INTEGRAL SAMPLER AND CONTROLLED MONTE CARLO DIFFUSION

In this section, we briefly discuss two learning-based samplers used for normalizing constant estimation and refer readers to the original papers for detailed derivations. The path integral sampler (PIS) shares structural similarities with the RDS framework discussed in Thm. 5, using the time-reversal of a universal noising process that transforms any distribution into a prior – such as the OU process in RDS that converges to the standard normal or the Brownian bridge in PIS that converges to the delta distribution at zero. In contrast, the controlled Monte Carlo diffusion (CMCD) extends the JE framework from Sec. 3, focusing on learning the compensatory drift term along an arbitrary interpolating curve $(\pi_\theta)_{\theta \in [0,1]}$, as long as the density of each intermediate distribution $\pi_\theta$ is known up to a constant.

**Path integral sampler (PIS, Zhang & Chen (2022)).** The PIS learns the drift term of a reference SDE that interpolates the delta distribution at 0 and the target distribution $\pi$, which is closely connected with the Brownian bridge and the Föllmer drift (Chewi, 2022).

Fix a time horizon $T > 0$. For any drift term $(u_t)_{t \in [0,T]}$, let $\mathcal{Q}^u$ be the path measure of the following SDE:

$$\mathrm{d}X_t = u_t(X_t)\mathrm{d}t + \mathrm{d}B_t,\ t \in [0,T];\ X_0 \overset{\text{a.s.}}{=} 0.$$

In particular, when $u \equiv 0$, the marginal distribution of $X_T$ under $\mathcal{Q}^0$ is $\mathcal{N}(0, TI) =: \phi_T$. Define another path measure $\mathcal{Q}^*$ by

$$\mathcal{Q}^*(\mathrm{d}\xi_{[0,T]}) := \mathcal{Q}^0(\mathrm{d}\xi_{[0,T)}|\xi_T)\pi(\mathrm{d}\xi_T) = \mathcal{Q}^0(\mathrm{d}\xi_{[0,T]})\frac{\mathrm{d}\pi}{\mathrm{d}\phi_T}(\xi_T),\ \forall \xi \in C([0,T];\mathbb{R}^d)$$

and consider the problem

$$u^* = \operatorname*{argmin}_u \mathrm{KL}(\mathcal{Q}^u \| \mathcal{Q}^*) \implies \mathcal{Q}^{u^*} = \mathcal{Q}^*.$$

One can calculate the KL divergence between these path measures via Girsanov's theorem (Lem. 1):

$$\log \frac{\mathrm{d}\mathcal{Q}^u}{\mathrm{d}\mathcal{Q}^*}(X) = W^u(X) + \log Z,\ \text{a.s. } X \sim \mathcal{Q}^u,\ \text{where}$$

$$W^u(X) = \int_0^T \langle u_t(X_t), \mathrm{d}B_t \rangle + \frac{1}{2}\int_0^T \|u_t(X_t)\|^2 \mathrm{d}t - \frac{\|X_T\|^2}{2T} + V(X_T) - \frac{d}{2}\log 2\pi T,$$

which implies $Z = \mathbb{E}_{\mathcal{Q}^u} e^{-W^u}$, and $\mathrm{KL}(\mathcal{Q}^u \| \mathcal{Q}^*) = \mathbb{E}_{\mathcal{Q}^u} W^u + \log Z$. On the other hand, directly applying Lem. 1 gives

$$\mathrm{KL}(\mathcal{Q}^u \| \mathcal{Q}^*) = \frac{1}{2}\int_0^T \mathbb{E}_{\mathcal{Q}^u} \|u_t(X_t) - u_t^*(X_t)\|^2 \mathrm{d}t.$$

In Zhang & Chen (2022, Theorem 3), the authors considered the effective sample size (ESS) defined by $\mathrm{ESS}^{-1} = \mathbb{E}_{\mathcal{Q}^u}\left(\frac{\mathrm{d}\mathcal{Q}^*}{\mathrm{d}\mathcal{Q}^u}\right)^2$ as the convergence criterion, and stated that $\mathrm{ESS} \geq 1 - \varepsilon$ as long as $\sup_{t \in [0,T]} \|u_t - u_t^*\|_{L^\infty}^2 \leq \frac{\varepsilon}{T}$. However, this condition is generally hard to verify since the closed-form expression of $u^*$ is unknown, and the $L^\infty$ bound might be too strong. Using the criterion ((1)) and the same methodology in proving the convergence of JE (Thm. 2), we can establish an improved result on the convergence guarantee of this estimator, relating the relative error to the training loss of $u$, which is defined as

$$\min_u L(u) := \mathbb{E}_{\mathcal{Q}^u}\left[\frac{1}{2}\int_0^T \|u_t(X_t)\|^2 \mathrm{d}t - \frac{\|X_T\|^2}{2T} + V(X_T)\right] = \mathrm{KL}(\mathcal{Q}^u \| \mathcal{Q}^*) - \log Z + \frac{d}{2}\log 2\pi T$$

**Proposition 4.** *Consider the estimator* $\widehat{Z} := e^{-W^u(X)}$, $X \sim \mathcal{Q}^u$ *for* $Z$. *To achieve both* $\mathrm{KL}(\mathcal{Q}_T^u \| \pi) \lesssim \varepsilon^2$ *(with* $\mathcal{Q}_T^u$ *representing the law of* $X_T$ *in the sampled trajectory* $X \sim \mathcal{Q}^u$*) and* $\Pr\left(\left|\frac{\widehat{Z}}{Z} - 1\right| \leq \varepsilon\right) \geq \frac{3}{4}$, *it suffices to choose* $u$ *that satisfies*

$$L(u) = -\log Z + \frac{d}{2}\log 2\pi T + O(\varepsilon^2).$$

*Proof.*

$$\Pr\left(\left|\frac{\widehat{Z}}{Z} - 1\right| \geq \varepsilon\right) = \mathcal{Q}^u\left(\left|\frac{\mathrm{d}\mathcal{Q}^*}{\mathrm{d}\mathcal{Q}^u} - 1\right| \geq \varepsilon\right) \lesssim \frac{\mathrm{TV}(\mathcal{Q}^u, \mathcal{Q}^*)}{\varepsilon} \lesssim \frac{\sqrt{\mathrm{KL}(\mathcal{Q}^u\|\mathcal{Q}^*)}}{\varepsilon}.$$

Therefore, ensuring $\mathrm{KL}(\mathcal{Q}^u\|\mathcal{Q}^*) \lesssim \varepsilon^2$ up to some sufficiently small constant guarantees that the above probability remains bounded by $\frac{1}{4}$. Furthermore, by the data-processing inequality, $\mathrm{KL}(\mathcal{Q}^u_T\|\pi) \leq \mathrm{KL}(\mathcal{Q}^u\|\mathcal{Q}^*) \lesssim \varepsilon^2$. $\square$

**Controlled Monte Carlo Diffusion (CMCD, Vargas et al. (2024)).** We borrow the notations from Sec. 3 due to its similarity with JE.

Given $(\widetilde{\pi}_t)_{t\in[0,T]}$ and the ALD (2), we know from the proof of Thm. 1 that to make $X_t \sim \widetilde{\pi}_t$ for all $t$, the compensatory drift term $(v_t)_{t\in[0,T]}$ must generate $(\widetilde{\pi}_t)_{t\in[0,T]}$. Now, consider the task of learning such a vector field $(u_t)_{t\in[0,T]}$ by matching the following forward and backward SDEs:

$$\mathcal{P}^\rightarrow:\ \mathrm{d}X_t = (\nabla\log\widetilde{\pi}_t + u_t)(X_t)\mathrm{d}t + \sqrt{2}\mathrm{d}B_t,\ X_0 \sim \widetilde{\pi}_0,$$

$$\mathcal{P}^\leftarrow:\ \mathrm{d}X_t = (-\nabla\log\widetilde{\pi}_t + u_t)(X_t)\mathrm{d}t + \sqrt{2}\mathrm{d}B_t^\leftarrow,\ X_T \sim \widetilde{\pi}_T,$$

where the loss is $\mathrm{KL}(\mathcal{P}^\rightarrow\|\mathcal{P}^\leftarrow)$, discretized in training. Obviously, when trained to optimality, both $\mathcal{P}^\rightarrow$ and $\mathcal{P}^\leftarrow$ share the marginal distribution $\widetilde{\pi}_t$ at every time $t$. By Girsanov's theorem (Lem. 3), one can prove the following identity for a.s. $X \sim \mathcal{P}^\rightarrow$: $\log\frac{\mathrm{d}\mathcal{P}^\rightarrow}{\mathrm{d}\mathcal{P}^\leftarrow}(X) = W(X) + C^u(X) - \Delta F$, where $\Delta F$ and $W(X)$ are defined as in Thm. 1, and

$$C^u(X) := -\int_0^T (\langle u_t(X_t), \nabla\log\widetilde{\pi}_t(X_t)\rangle + \nabla\cdot u_t(X_t))\mathrm{d}t.$$

We refer readers to Vargas et al. (2024, Prop. 3.3) for the detailed derivation. By $\mathbb{E}_{\mathcal{P}^\rightarrow}\frac{\mathrm{d}\mathcal{P}^\leftarrow}{\mathrm{d}\mathcal{P}^\rightarrow} = 1$, we know that $\mathbb{E}_{\mathcal{P}^\rightarrow}\mathrm{e}^{-W(X)-C^u(X)} = \mathrm{e}^{-\Delta F}$. As the paper has not established inference-time performance guarantee given the training loss, we prove the following result characterizing the relationship between the training loss and the accuracy of the sampled distribution as well as the estimated normalizing constant.

**Proposition 5.** *Let $\widehat{Z} = Z_0\mathrm{e}^{-W(X)-C^u(X)}$, $X \sim \mathcal{P}^\rightarrow$ be an unbiased estimator of $Z = Z_0\mathrm{e}^{-\Delta F}$. Then, to achieve both $\mathrm{KL}(\mathcal{P}^\rightarrow_T\|\pi) \lesssim \varepsilon^2$ (where $\mathcal{P}^\rightarrow_T$ is the law of $X_T$ in the sampled trajectory $X \sim \mathcal{P}^\rightarrow$) and $\Pr\left(\left|\frac{\widehat{Z}}{Z} - 1\right| \leq \varepsilon\right) \geq \frac{3}{4}$, it suffices to choose $u$ that satisfies $\mathrm{KL}(\mathcal{P}^\rightarrow\|\mathcal{P}^\leftarrow) \lesssim \varepsilon^2$.*

*Proof.* The proof of this theorem follows the same reasoning as that of Prop. 4. For normalizing constant estimation,

$$\Pr\left(\left|\frac{\widehat{Z}}{Z} - 1\right| \geq \varepsilon\right) = \mathcal{P}^\rightarrow\left(\left|\frac{\mathrm{d}\mathcal{P}^\leftarrow}{\mathrm{d}\mathcal{P}^\rightarrow} - 1\right| \geq \varepsilon\right) \lesssim \frac{\mathrm{TV}(\mathcal{P}^\rightarrow, \mathcal{P}^\leftarrow)}{\varepsilon} \lesssim \frac{\sqrt{\mathrm{KL}(\mathcal{P}^\rightarrow\|\mathcal{P}^\leftarrow)}}{\varepsilon} \lesssim 1.$$

For sampling, the result is an immediate corollary of the data-processing inequality. $\square$

# I DETAILS OF EXPERIMENTAL RESULTS

## I.1 MODIFIED MÜLLER BROWN DISTRIBUTION

The Müller Brown potential energy surface is a canonical example of a potential surface used in molecular dynamics. Here, we consider a modified version of this distribution as defined in He et al. (2024, App. D.5). For $x = (x_1, x_2) \in \mathbb{R}^2$, the target distribution is $\pi(x) = \frac{1}{Z}\exp(-0.1(V_q(x) + V_m(x)))$, where

$$V_q(x) = 35.0136(\overline{x}_1 + 0.033923)^2 + 59.8399(\overline{x}_2 - 0.465694)^2,$$

$$V_m(x) = \sum_{i=1}^4 A_i\exp(a_i(\overline{x}_1 - X_i)^2 + b_i(\overline{x}_1 - X_i)(\overline{x}_2 - Y_i) + c_i(\overline{x}_2 - Y_i)^2).$$

In the above equations, $\overline{x}_1 = 0.2(x_1 - 3.5)$, $\overline{x}_2 = 0.2(x_2 + 6.5)$, $A = (-200, -100, -170, 15)$, $a = (-1, -1, -6.5, 0.7)$, $b = (0, 0, 11, 0.6)$, $c = (-10, -10, -6.5, 0.7)$, $X = (1, 0, -0.5, -1)$, $Y = (0, 0.5, 1.5, 1)$. The ground truth value of the normalizing constant computed by numerical integral (`scipy.integrate.dblquad`) is $Z = 22340.9983$ with estimated absolute error $0.0001$.

We run each method with approximately the same oracle complexity. Aside from the quantitative results in Tab. 1, we also visualize the samples drawn from each method against the level curves of the potential in Fig. 2. It is clear from the table and figure that TI and AIS fail to provide accurate estimates of the normalizing constant or sample from the target distribution due to the deficiency of the exploration of different modes. All four RDS-based methods provide accurate estimates of the normalizing constant, with SNDMC and ZODMC being the two best methods.

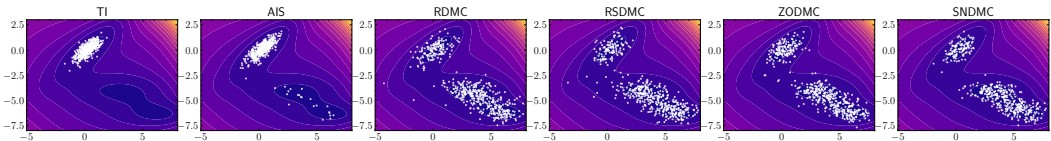

Figure 2: Visualization of the samples from the modified Müller Brown distribution. The generated samples are displayed on top of the level curves of the potential energy surface (darker color corresponds to lower potential energy, i.e., higher probability density).

## I.2 GAUSSIAN MIXTURE DISTRIBUTION

We now consider a Gaussian mixture distribution $\pi$ in $\mathbb{R}^2$ with 4 components, having weights $0.1, 0.2, 0.3, 0.4$, means

$$\begin{pmatrix} 0 \\ 0 \end{pmatrix}, \begin{pmatrix} 0 \\ 11 \end{pmatrix}, \begin{pmatrix} 9 \\ 9 \end{pmatrix}, \begin{pmatrix} 11 \\ 0 \end{pmatrix},$$

and covariances

$$\begin{pmatrix} 1 & 0.5 \\ 0.5 & 1 \end{pmatrix}, \begin{pmatrix} 0.3 & -0.2 \\ -0.2 & 0.3 \end{pmatrix}, \begin{pmatrix} 1 & 0.3 \\ 0.3 & 1 \end{pmatrix}, \begin{pmatrix} 1.2 & -1 \\ -1 & 1.2 \end{pmatrix}.$$

As the p.d.f. is available in closed form, the ground truth value of the normalizing constant is $Z = 1$. Due to the separation of the modes and the imbalance of the weights, this distribution is more challenging to sample from. In the quantitative results shown in Tab. 1, we report the mean and standard deviation of $\frac{\widehat{Z}}{Z}$ as well as two metrics for the quality of the samples: maximum mean discrepancy (MMD) and Wasserstein-2 distance ($W_2$) between the generated samples $\widehat{\pi}_{\text{samp}}$ and ground truth samples from $\pi$. The visualization of the samples is shown in Fig. 3. Again, TI and AIS are confined to mode at zero where the initial samples are located, and fail to provide accurate estimates of the normalizing constant. All RDS-based methods provide accurate estimates of the normalizing constant and high quality samples.

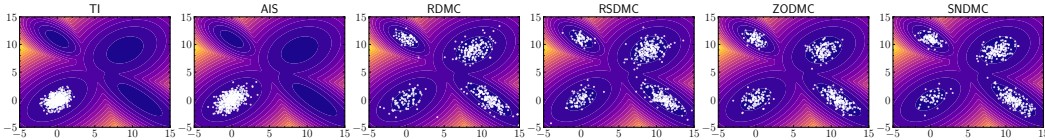

Figure 3: Visualization of the samples from the Gaussian mixture distribution. The generated samples are displayed on top of the level curves of the potential (darker color corresponds to lower potential, i.e., higher probability density).

## I.3 IMPLEMENTATION DETAILS

**General implementation details.** For both experiments, we run each method for $1024$ rounds and output the mean and standard deviation of all $1024$ estimates of $\frac{\widehat{Z}}{Z}$. In each round, we parallelly

run 1024 i.i.d. trajectories, which produces 1024 i.i.d. samples from $\pi$ and 1024 i.i.d. estimates of the normalizing constant, and we treat the average of the estimates as the final estimate of that round. We record the oracle complexity of each algorithm and tune the hyperparameters to make sure that the oracle complexity for producing *each sample* from $\pi$ is between 50000 and 60000 for a fair comparison. For TI, we choose $\lambda_0 = 100$, $\lambda_{i+1} = \frac{1.45}{1+1/\sqrt{d}}\lambda_i$ until $\lambda_i \leq \frac{1}{2\sqrt{d}}$, and $N = 32$ i.i.d. samples. For AIS, we choose $\lambda_0 = 100$, $M = 60000$ steps, and ALMC step size $T_\ell = 0.01$. For all RDS-based methods, we choose the total time duration $T = 5$, early stopping time $\delta = 0.005$, and $N = 50$ uniformly spaced time points $t_n = \frac{n}{N}(T - \delta)$. Specifically, for RDMC, we use 64 samples from $\overline{\pi}_{0|t}(\cdot|x)$ to estimate the score $\nabla \log \overline{\pi}_t(x)$, and run LMC for 16 steps with step size 0.01, initialized by importance sampling from $\overline{\pi}_{0|t}(\cdot|x) \propto \mathrm{e}^{-V(\cdot)} \mathcal{N}\left(\cdot|\mathrm{e}^t x, (\mathrm{e}^{2t} - 1)I\right)$ with proposal $\mathcal{N}\left(\mathrm{e}^t x, (\mathrm{e}^{2t} - 1)I\right)$; for RSDMC, we choose the number of recursive steps as 2, use 16 samples from $\overline{\pi}_{0|t}(\cdot|x)$ to estimate the score $\nabla \log \overline{\pi}_t(x)$, and run LMC for 10 steps with step size 0.01 using the same initialization based on importance sampling; finally, for both ZODMC and SNDMC, we use 1024 samples from $\overline{\pi}_{0|t}(\cdot|x)$ to estimate the score $\nabla \log \overline{\pi}_t(x)$.

**Evaluation metrics for sampling.** In the experiment of Gaussian mixture distribution, in each round, we draw 1024 samples from both the algorithm and the target distribution, and compute the following two metrics to evaluate the quality of the samples. For two sets of samples $\mathcal{X} = \{x_i\}_{i=1}^n$ and $\mathcal{Y} = \{y_j\}_{j=1}^m$, the MMD is defined as

$$\mathrm{MMD}(\mathcal{X}, \mathcal{Y}) := \sqrt{\frac{1}{n^2} \sum_{1 \leq i, i' \leq n} k(x_i, x_{i'}) - \frac{2}{nm} \sum_{1 \leq i \leq m, 1 \leq j \leq n} k(x_i, y_j) + \frac{1}{m^2} \sum_{1 \leq j, j' \leq m} k(y_j, y_{j'})},$$

where $k(x, y) = \frac{1}{K} \sum_{i=1}^K \exp\left(-\frac{\|x-y\|^2}{2\sigma_i^2}\right)$ is a multiscale radial basis function (RBF) kernel. Following the implementation in He et al. (2024), we set $K = 10$ and $\{\sigma_i\}_{i=1}^{10} = \{-4, -2, 0, ..., 12, 14\}$. Second, the $\mathrm{W}_2$ distance is computed by `ot.emd2(a, b, M) ** 0.5` using the Python Optimal Transport (POT) package (Flamary et al., 2021), where $\mathtt{a} = \frac{1}{n}1_n$, $\mathtt{b} = \frac{1}{m}1_m$, and $\mathtt{M} = (\|x_i - y_j\|^2)_{1 \leq i \leq n, 1 \leq j \leq m}$. $1_n$ represents the vector of all ones with length $n$.

