# OpenReview forum: "Complexity Analysis of Normalizing Constant Estimation: from Jarzynski Equality to Annealed Importance Sampling and beyond"
_ICLR.cc/2026/Conference — ICLR 2026 Poster_

### Official Review · Reviewer_8vBo · 2025-10-26

**Soundness:** 4
**Presentation:** 2
**Contribution:** 4
**Rating:** 8
**Confidence:** 4

**Summary:**

This paper provides a full theoretical analysis of the error in Annealed Importance Sampling (AIS), accounting for both the sampling process that generates particles and the estimator of the normalizing constant computed from these samples. The authors derive a general upper bound on the estimation error as a function of the prescribed probability path that guides the sampling process. Two important path choices are examined: the standard geometric interpolation path, which is shown to lead to exponential complexity in the difficulty of the problem, and the reverse diffusion path, which achieves only polynomial complexity—provided oracle access to score functions (otherwise, an additional approximation error appears). The difficulty of the problem is quantified in terms of the between-mode distance of the target distribution.

**Strengths:**

This is a welcome analysis. There is a long literature that analyzes the error of Annealed Importance Sampling, but it usually:
- makes strong assumptions: assuming equilibrium sampling, using the asymptotic regime where the number of samples is big, etc.
- is not very interpretable: the formula of the error does not clearly tell the user how to design the prescribed probability path

It has been a goal in that literature to produce a general formula of the error that is interpretable enough to inform the design choice of the probability path. To my knowledge, this is the first such analysis.

**Weaknesses:**

The writing could be clearer in some parts, but overall the paper is clear.

**Questions:**

Q1. Theorem 4 is particular to the geometric interpolation path? Is it normal that the number of samples $N$, the number of SMC iterations $M$, and the discretization of the Langevin process, do not appear in the error bound (Eq 11)?

---

> ### Author Response · Authors · 2025-11-21
>
> Thank you for your positive feedback and thoughtful comments. We are glad to hear that you find our analysis significant. Please find our responses to your concerns below.
>
> > **Question 1.1**: Is Theorem 4 particular to the geometric interpolation path?
>
> Yes. We would like to first clarify that our analysis for JE (Thm. 2) works for any interpolation curve with sufficient regularity conditions. However, for AIS, we focus on geometric interpolation in Thm. 4 for the sake of simplicity and clarity, and geometric interpolation is also a representative annealing sequence widely used in practice. As long as we know the score $\nabla\log\pi _ \theta$ for all intermediate distributions, we can generalize the analysis to such interpolation curve.
>
> > **Question 1.2**: Hyperparameters not appearing in the error bound
>
> We appreciate your insightful question. In Thm. 4, to obtain an estimate $\widehat Z$ within $\varepsilon$ relative error with high probability, one needs to choose the hyperparameters you mentioned appropriately. We only presented the total oracle complexity for achieving this accuracy in the main text to avoid making it too math-heavy. The specific choices of these hyperparameters given the problem parameters are provided throughout the proof in App. D. In particular, we need to choose $N=\widetilde{\Theta}\left(\frac{\sqrt d}{\varepsilon^2}\right)$ (number of i.i.d. approximate samples for TI), $M=\Theta\left(\frac{m\beta\mathcal{A _ r}^\frac12}{\varepsilon^2}\vee\frac{d\beta^2\mathcal{A _ r}^2}{\varepsilon^4}\right)$ (number of steps for AIS), $T _ \ell=\Theta\left(\frac{\mathcal{A} _ r^\frac{1}{2}}{m\beta}\wedge\frac{\varepsilon^2}{d\beta^2\mathcal{A} _ r}\right)$ (the step size of ALMC at each iteration).
>
> ---
>
> Thank you again for your valuable feedback. We hope these clarifications satisfactorily address your questions. Please feel free to ask further questions or raise additional comments, and we will be happy to respond.

---

> > ### Comment · Reviewer_8vBo · 2025-11-23
> > **Response to authors**
> >
> > Thank you authors for your response which satisfactorily addresses my concerns. Could you include the above clarifications in the main text? It was not clear to me initially that:
> >
> > - theorem 4 was for the geometric path
> > - it presents an oracle complexity which requires specific hyperparameter values (you can cite the relevant equation numbers from App D)
> >
> > Reading the authors' response to other reviewers has also clarified other points for me. I believe the main text would be much clearer by including the clarifications to other reviewers in the camera-ready version.

---

> > > ### Author Response · Authors · 2025-11-24
> > >
> > > Thank you for your continued engagement with our work. We are glad to hear that our responses have satisfactorily addressed your concerns.
> > >
> > > We have uploaded a revised version of the manuscript that incorporates the changes you suggested:
> > >
> > > 1. For Thm. 4, we have mentioned the specific restriction and possible extensions of this result between line 390 and 394.
> > > 2. For required hyperparameter values for AIS, we have added "The required values ... can be found at the end of the proof" on lines 346 and 347, before Thm. 4. For hyperparameter values for TI, we have added a reference to App. H.1 (paragraph (IV)) on line 313.
> > >
> > > We hope these revisions meet your expectations. Please let us know if there are any further questions or comments.

---

### Official Review · Reviewer_s4rX · 2025-10-27

**Soundness:** 2
**Presentation:** 1
**Contribution:** 2
**Rating:** 2
**Confidence:** 4

**Summary:**

The paper uses a novel method to analyze and bound the non-asymptotic estimation error of the normalization constant for an energy functions in terms of smoothness of the potential and the action of annealing curve.
The method relies on conditions that are less restrictive than the commonly used isoperimetric constraints for this type of analysis and applies to non-log-concave distributions. The analysis sheds light on the increased complexity of estimation with geometric interpolation in comparison to OU reversal.

**Strengths:**

I believe the main contribution of the paper, according to itself, is the non-asymptotic analysis (in the number of path samples) for TI and AIS (LMC) which is presented in THM2 and 4. THM2 proves constant probability bound on the error of the normalization factor estimation ratio in the path integral of Thermodynamic integration. THM4 proves an upper bound on the number of oracle calls $M$ in AIS (LMC, Alg. 1).

**Weaknesses:**

1 - Some important recent literature is missing from the work. How do you compare your method to Adjoint sampler or methods approximating the Kantorovich potential of the RDS.

2 - Organization of the paper is very synthetic. The paper is super technical, lacks a properly structured background section and multiple theoretical results are presented without a straight forward relevance to each other. The notations are not clearly defined in the text ($B_t$ and $B_t^\leftarrow$ in Eq 2 and 4, $n$ in line 444) and abbreviations are missing (TI, RN, SDE). Navigating through the paper is very difficult. I don't think the inclusion of Lem1 or THM1 and THM3 and their proofs are necessary for the text - this context can be built in a less technical and more cohesive manner. I recommend the authors restructure the paper and this is the main reason I'm in favor of rejecting this version. In the following I provide a list of modifications that are going to influence my decision.

3 - Use uniform notation across different theorems, sections and appendix as much as possible. For instance, $k$ and $l$ for step indices, forward and backward SDEs in proof of THM2 reuse the same notation.

4 - Simplify the representation of the Theorems

4.1 - The use of scaling $t/T$ in THM2 is not necessary. It appears that the action $\mathcal A = \int_0^1 |\dot \pi_t|^2dt$ is proportional to $T$ which is used for scaling $t$. Therefore, the last line of proof implies a constant upper bound on the path integral for any speed and not just the specified $T$ (see proof of THM2).

4.2 - Prop1 and Prop2 and THM6 seem out of place with respect to the rest of the paper. Is there a corollary missing? It might be the case that sample complexity bound depends on the action, but the relations are only derived for the geometric interpolation (Eq 7) in THM4. It is not enough to suggest that the lower bound on action in geometric and upper bound in RDS translates directly to better upper bound for RDS sample complexity.

4.3 - In THM5 what is the use of $\delta$ in practice? This Theorem is adapted from previous work with the addition of $delta$ for which I'm not able to see the motivation.

4.4 - Appendix E.5 doesn't provide any analysis on the sample complexity of the mentioned algorithms, only references. Are the orders taken from the references? If so should be stated like that in the paper as opposed to analysis (line 445).

5 - Throughout the paper different units of complexity are used to compare different algorithms. It would be helpful to write down the dependence of all complexity bounds in terms of the action, bounding the action, and with the median trick in a table for clarity.

6 - I'm not certain how Alg. 2 is used in "the series of new algorithms based on reverse diffusion samplers". I'm also assuming the experiment section somehow incorporates Alg. 2. But RDMC, RSDMC, ZODMC, SNDMC all are reference algorithms with their own state update steps and importance weight computation. It is not clear how Alg. 2's state and weight update is incorporated in the reference algorithms to get new results.

7 - Experiments in $\mathbb R^2$ tend to be too simple and lead to overlapping results (confidence bounds) from RDS based algorithms. Results in more complex setups could prove more interesting.

**Questions:**

1 - It feels counter intuitive that a nested linear ($\theta$ in Eq. 10) and exponential (line 317 for $\lambda$) discretization of the schedule would reduces the overall estimation error in comparison to purely linear or exponential schedule (line 324). Since the proof of THM4 bounds the error with $r = 1$, I don't see the motivation for the exponential $\lambda$. Can you provide some examples that show how the cost (11) depends on $r$? What are the conditions for it to improve with $r > 1$? Is there a benefit of exponential cascading of $\lambda$?

2 - As the main issue with multimodal distributions arises in high dimensions, is there any explanation on how the action depends on $d$ in PROP1?

3 - Can you give a brief explanation of mass teleportation an mode switching phenomenom (line 394)?

4 - The order of OU sampling error (reversing the noising process) is known to be independent of $d$ and similar to the OU process accuracy [De Bortoli - NeurIPS2021]. Where does the dependence on $d$ come from in your derivation in line 444?

5 - Can the authors detail how they incorporate Alg. 2 in the mentioned algorithms to get new results?

6 - Alg. 2 seems like a typical diffusion sampler weight update, e.g. similar to that of PIS [Zhang & Chen 2022]. Is the difference with PIS in decomposition of the Brownian motion?

Minor comments:
Typo: line 316 ALMC -> LMC

---

> ### Author Response · Authors · 2025-11-21
>
> Thank you for carefully reading our paper and providing detailed comments. We address your concerns as follows.
>
> > **Weakness 1**: Missing related work
>
> Thanks for bringing these works to our attention.
>
> Adjoint sampler is designed for learning a neural sampler and, through some adaptations, can be used for estimating normalizing constants. We have included it in the related work section of the revised manuscript. For the experiments, as our focus is mainly on theoretically studying the property of annealing curve for normalizing constant estimation, learning-based methods are not our main focus, and therefore we do not intend to compare with them empirically.
>
> To the best of our knowledge, we are not aware of works on normalizing constant estimation that approximate the Kantorovich potential of the RDS. If you have specific references in mind, please let us know.
>
> > **Weakness 2**: Organization of the paper
>
> Thanks for your feedback on the organization of the paper. We agree that the paper is technical due to the nature of the topic, and we will try our best to improve its clarity and accessibility.
>
> As discussed in the introduction, we have integrated the required background information into the main text, and provided a detailed exposition available in App. A, which we believe makes the paper more self-contained and accessible to a broader audience.
>
> As per your suggestions, we have included the definition of $B^\gets _ t$ in the proof of Thm. 1. $n$ in the bound of RDMC appears in an assumption, and we included it in a footnote in the main text. The abbreviations such as TI, RN, and SDE have all been included when they appeared for the first time in the main text, if they were missing before.
>
> We included the sketch of proofs of JE (Thm. 1) and AIS (Thm. 3) in the main text to help readers better understand the *main ideas* of our technical contributions through the forward and backward SDEs / Markov processes. Lem. 1 is a key technical tool that is used in the proof of the complexity of JE (Thm. 2). To make the main text less math-heavy, we removed it and only present the full version (Lem. 4 in the updated manuscript, previously Lem. 5) in the appendix.
>
> > **Weakness 3**: Use uniform notation
>
> Thank you for the suggestion. We agree on aiming for uniform notation where possible; however, $k$ and $\ell$ intentionally denote step indices in *different* algorithms (TI and AIS, respectively). Although both are step indices, they carry different meanings: while $\ell$ in AIS is just the time step for discretizing the continuous-time ALD, $k$ in TI are the indices for the discrete sequence of intermediate distributions, and for each $k$, we need to run many steps of LMC to approximately sample from the $k$-th intermediate distribution. Thus, reusing the same symbol would create ambiguity across sections and proofs. In the proof of Thm. 2, as we only focus on distribution-level quantities, $B _ t$ and $B^\gets _ t$ are independent forward and backward Brownian motions.
>
> > **Weakness 4.1**: Use of scaling $t/T$ in Thm. 2 is not necessary
>
> Thank you for pointing this out this issue. We would like to clarify the purpose of scaling as follows.
>
> As you have noticed, the action is the integral of *squared* metric derivative, so it is proportional to $T$ if we do not scale the time. For a given curve $(\pi _ \theta) _ {\theta\in[0,1]}$ standardized with time parameter between $0$ and $1$, by scaling to a larger time duration $T>1$, we are actually running the annealed Langevin dynamics more slowly, which intuitively leads to smaller estimation error. Therefore, to quantify how slowly we should run the dynamics, or equivalently, how large the total time $T$ should be for simulating the dynamics with a *fixed speed* (which means the coefficient in front of $\mathrm{d}B _ t$ is fixed to be $\sqrt{2}$), we need to include the scaling $t/T$ in the definition of the dynamics for *fair comparison* for the curve property.

---

> > ### Author Response · Authors · 2025-11-21
> >
> > > **Weekness 4.2**: Prop. 1, Prop. 2 and Thm. 6 seem out of place
> >
> > We first clarify that the analysis of JE (Thm. 2) works for *any* interpolation curve with sufficient regularity conditions, and we have shown that the time required for simulating the continuous-time ALD for achieving a certain relative error is proportional to the action of the curve. Our result of AIS (Thm. 4) is indeed specialized to geometric interpolation for the sake of simplicity and clarity, but we can generalize it to any reasonable interpolation curve as long as we know the score function of the intermediate distributions along the curve.
> >
> > We certainly agree that the bound on action, which provides a convergence guarantee for continuous-time JE algorithm, does not directly translate to non-asymptotic algorithmic complexity for discrete-time AIS algorithm. But intuitively, when the action of a curve is large, as the continuous-time ALD requires large time $T$ to achieve small estimation error, the discretized AIS will also require a large number of steps to obtain accurate estimation if we use similar order of step sizes. A more rigorous connection between the action and the sample complexity lower bound for AIS is an interesting topic, and we are happy to explore it in future work.
> >
> > > **Weakness 4.3**: The use of $\delta$ in practice
> >
> > Thank you for raising this important question regarding the practical implementation. As mentioned in the statement of Thm. 5, $\delta$ is an early stopping time for the reverse diffusion process, i.e., we only run the reverse diffusion from time $T$ to time $\delta$ rather than $0$. The purpose of introducing an early stopping time $\delta$ is to avoid the numerical instability issue when running the reverse diffusion to time $0$ (i.e., close to the target distribution), where the regularity on the score function may be bad. This is a typical technique used in practice for diffusion models, see Song et al. (2021). In practice, it is typically chosen to be a small constant, e.g., $0.005$ in our experiments (see App. I.3).
> >
> > **Reference**: Song et al. Score-based generative modeling through stochastic differential equations. ICLR 2021.
> >
> > > **Weakness 4.4**: App. E.5
> >
> > Thanks for pointing this out. The orders are indeed taken from the references, but we would like to clarify that they are all bounds on *sampling* complexity rather than *normalizing constant estimation* complexity. Our contribution in this part is threefold:
> >
> > (1) In Thm. 5, we **proposed an algorithm** for estimating normalizing constants based on these RDS sampling algorithms. What's different from prior works is for sampling one only needs to simulate $\mathbb{Q}^\dagger$ (4), where the score $s _ {T-t _ -}(X _ {t _ -})$ is estimated through different techniques introduced in these four papers; however, for normalizing constant estimation, one also needs to compute the functional $W(X)$ along the trajectory. The detailed algorithm is presented in Alg. 2.
> >
> > (2) In Thm. 5 and the discussion after it, we provided sufficient conditions under which the relative error of the normalizing constant estimation is less than $\varepsilon$ with high probability, i.e., **convergence guarantees**. We proved that the score estimation error in $L^2$ (connected with the KL divergence between the path measures) needs to be small enough. To guarantee this, each of these four papers has its own set of assumptions and analysis, which we have summarized in App. E.5.
> >
> > (3) Finally, we provided **experimental results** on these RDS-based algorithms for normalizing constant estimation.
> >
> > In summary, our contribution here is not a simple restatement of the sampling complexity results in the four references, but rather a novel *algorithm* and *analysis*.

---

> > > ### Author Response · Authors · 2025-11-21
> > >
> > > > **Weakness 5**: Clarity on complexity bounds
> > >
> > > Thank you for the advice. The complexity bounds are summarized as follows:
> > >
> > > |  |  |  |
> > > |---|---|---|
> > > | Continuous-time algorithm | JE | ALD run time $O\left(\frac{\mathcal{A}}{\varepsilon^2}\right)$ |
> > > | Discrete-time algorithm | AIS | Oracle complexity $\widetilde{O}\left(\frac{d\beta^2\mathcal{A}^2}{\varepsilon^4}\right)$ for geometric annealing |
> > > |  | RDS | Oracle complexity generally $O(\mathrm{e}^{\mathrm{poly}(d,\beta)})$ without isoperimetry |
> > > | Bound on action $\mathcal{A}$ | Geometric annealing | Without mode switching: $O(\mathrm{poly}(d,\beta,\mathrm{mode ~ separation}))$; With mode switching: $\Omega(\exp(\mathrm{poly}(\mathrm{mode ~ separation})))$ |
> > > |  | OU process | $O(d\beta+m^2)$ |
> > >
> > > > **Weakness 6 & Question 5**: Incorporation of Alg. 2 in the mentioned algorithms
> > >
> > > These four RDS algorithms serve as the **non-parametric score estimator $s _ t(\cdot)\approx\nabla\log\overline\pi _ t(\cdot)$** in Alg. 2. For each of these four algorithms, we can use the proposed method to estimate the score, and use the framework in Thm. 5 to provide estimation of the normalizing constant. As these works are originally intended for sampling rather than normalizing constant estimation, they themselves do not provide a way to compute the importance weight $W(X)$. We have highlighted the role of these four algorithms -- score estimator -- in the input of Alg. 2 in the revised manuscript.
> > >
> > > > **Weakness 7**: Experimental details
> > >
> > > Thank you for your suggestion. Our main focus in the experiments is to verify the theoretical findings regarding the impact of action on the performance of different algorithms. In particular, we aim to demonstrate that RDS-based algorithms outperform AIS, showing the practical benefits of the OU curve in comparison to geometric interpolation, which is clearly illustrated in our 2D experiments.
> > >
> > > > **Question 1**: Clarification on the discretization schedule
> > >
> > > We appreciate your insightful comments. The exponential here refers to the **exponential integrator** scheme for numerically simulating SDEs with partially linear drift terms, instead of the exponential annealing schedule in $\lambda$.
> > >
> > > We review the exponential integrator numerical scheme as follows. Suppose $\mathrm{d}X _ t=(a _ tX _ t+b(X _ t,t))\mathrm{d}t+\sigma\mathrm{d}B _ t$, where $a _ t$ is a time-dependent scalar and $b(x,t)$ is a general non-linear function. The Euler-Maruyama scheme assumes the full drift term $(a _ tX _ t+b(X _ t,t))$ is fixed within time interval $[s,s+h]$, leading to the update
> > > $$X _ {s+h}\approx X _ s+(a _ sX _ s+b(X _ s,s))h+\sigma\sqrt{h}\mathcal{N}(0,I).$$
> > > In contrast, the exponential integrator scheme only assumes $b(X _ t,t)$ is fixed within $[s,s+h]$, and solves the linear ODE part exactly, leading to the update
> > > $$X _ {s+h}\approx\mathrm{e}^{a _ sh}X _ s+\left(\int _ 0^h\mathrm{e}^{a _ s(h-u)}\mathrm{d}u\right)b(X _ s,s)+\sigma\sqrt{\frac{\mathrm{e}^{2a _ sh}-1}{2a _ s}}\mathcal{N}(0,I).$$
> > >
> > > By assuming only part of the drift term is fixed, the exponential integrator scheme can better reduce the discretization error. In our example, since the ALD with geometric interpolation has a partially linear drift term (corresponding to the quadratic part of the Gaussian prior), the exponential integrator is appropriate here. For more details and applications to diffusion model, we refer to Zhang and Chen (2023).
> > >
> > > Finally, our bound in Thm. 4 is uniform for all $1\le r\le O(1)$. The $r$ serves as a way of adjusting the annealing schedule $\lambda(\theta)\propto(1-\theta)^r$, as intuitively we should spend more steps when we are close to the target distribution (where the distribution is more complicated and challenging to sample from), and thus we would like to choose larger $r$ to have finer discretization near $\theta=1$.
> > >
> > > **Reference**: Zhang and Chen. Fast sampling of diffusion models with exponential integrator. ICLR 2023.
> > >
> > > > **Question 2**: How the action depends on dimension
> > >
> > > We acknowledge the importance of addressing the dimension dependence of the proposed method. In some simple cases, $\mathcal{A}=O(d)$: for example, when $\pi _ \theta(x)=\prod _ {i=1}^d\nu _ \theta(x  _  i)$ for some curve of one-dimensional probability distributions $(\nu _ \theta) _ {\theta\in[0,1]}$, we have $W _ 2^2(\pi _ \theta,\pi _ {\theta+\delta})=dW _ 2^2(\nu _ \theta,\nu _ {\theta+\delta})$ by definition of the Wasserstein distance, and hence $|\dot\pi| _ \theta^2=d|\dot\nu| _ \theta^2$, so the action is $O(d)$. In general, the dimensional dependence of the action of the curve $\mathcal{A}$ is problem-dependent and not straightforward. For instance, if the number of modes grows with dimension, then the action may grow polynomially or even exponentially with dimension. This is an interesting future direction to further investigate sufficient conditions for polynomial action in high-dimensional settings, and we appreciate your suggestions on this.

---

> > > > ### Author Response · Authors · 2025-11-21
> > > >
> > > > > **Question 3**: A brief explanation of mass teleportation and mode switching phenomenon
> > > >
> > > > Thank you for your question. This is an important point that we should clarify in the manuscript. Briefly speaking, during the annealing process, the probability mass needs to be transported from one mode of the distribution to another well-separated mode in a short period of time, which leads to torpid mixing for many samplers. This can happen when the relative **weights** of different modes change significantly over the annealing schedule (e.g., in our proof of Prop. 1). An illustrative example can be found at Chemseddine et al. (2025, Figs. 1, 5), which involves geometric annealing. In contrast, the OU interpolation preserves the weights as scaling and Gaussian convolution are linear operators on distributions, so it does not suffer from this problem.
> > > >
> > > > **Reference**: Chemseddine et al. Neural Sampling from Boltzmann Densities: Fisher-Rao Curves in the Wasserstein Geometry. ICLR 2025.
> > > >
> > > > > **Question 4**: OU sampling error dependence on dimension
> > > >
> > > > This is an insightful observation. The result you mentioned holds under the assumption that (i) the log density of target distribution satisfies higher-order regularity, and (ii) the learned score is $L^\infty$-accurate, i.e., $||s _ \theta(x,.)-\nabla\log p _ t(x)||\le\mathtt{M}$ for *all* $x$ and $t$, which is generally impossible especially for Monte Carlo estimation of the score. These are much stronger assumptions that could possibly lead to dimension-free bounds. However, in the four RDS papers, (i) we only have a much weaker assumption (Assump. 2) for the target distribution, and (ii) the score needs to be estimated via Monte Carlo methods. Thus, the complexity of obtaining accurate score estimators typically depends on the dimension.
> > > >
> > > > > **Question 6**: Difference between Alg. 2 and PIS
> > > >
> > > > Thanks for this insightful observation. Both weight updates in PIS (see App. H.2) and RDS (Thm. 5) are used for computing the RN derivative between two path measures, which, by taking expectation under some transformation, provides estimate of the normalizing constant. The key difference is that RDS estimates the drift term (in particular, the score function) through learning-free non-parametric method, while PIS requires training a neural network to estimate the drift term. The decomposition of the Brownian motion in RDS is due to the use of exponential integrator scheme for discretization, a more accurate scheme than the Euler-Maruyama scheme used in PIS due to the linear term in the reverse SDE.
> > > >
> > > > ---
> > > >
> > > > We thank you again for your valuable feedback, which has helped us improve the quality of our work. We hope that our responses satisfactorily address your points and would welcome an updated evaluation. We would be happy to provide further clarifications if needed.

---

### Official Review · Reviewer_TbKt · 2025-10-31

**Soundness:** 4
**Presentation:** 4
**Contribution:** 3
**Rating:** 6
**Confidence:** 4

**Summary:**

The authors show query complexity bounds for normalization constant estimation using annealing-based methods including Jarzynski equality for SDE's and annealed importance sampling. The details for JE and AIS are different (being continuous/discrete), but they share a similar analysis framework.

The bounds are in terms of the "action", or total squared metric derivative through the annealing sequence. The authors give an example where this is exponentially large in a situation with mass teleportation, and use this to motivate applying JE to reverse diffusion samplers through a polynomial bound on the action only under smoothness and 2nd moment.

**Strengths:**

This is a very timely topic given the increasing presence of these algorithmic methods (especially Jarzynski's equality which has been adapted for sampling with flow-based neural network models) and the limited theory for them. Normalizing constant estimation is a fundamental problem in ML and statistics, and the authors do a good job of giving examples of applications. The paper is technically well-written, and I especially appreciate the careful treatment of the forward/backwards SDE's. I expect the theory developed in this paper will be useful for future analyses on this and similar algorithms.

The application to reverse diffusion samplers is an especially nice result. This shows that efficient normalizing constant estimation follows from general conditions given an accurate score estimate.

**Weaknesses:**

The proof is written just for geometric interpolation, though it would be better to have a general bound that works for an arbitrary annealing sequence and added drift, as this flexibility is often important in the literature, especially in approaches that depend on learning the annealing sequence and drift (e.g. Máté and Fleuret, 2023; Albergo and Vanden-Eijnden, 2025). Would a similar proof work, or is the proof specialized to the annealing sequence? What changes are necessary? Relatedly, would this framework apply to give guarantees for sampling?

The main downside is the polynomial dependence in all bounds on the "action." While this is weaker than isoperimetry, it's not clear whether this matches other bounds for annealing/tempering-type methods. In fact, I doubt this covers many cases where we'd want to use annealing-based methods: The authors show a lower bound for the action in the case of a mixture of 2 (1-D) gaussians using power tempering. It seems this analysis can also be adapted to show a lower bound in any case with well-separated modes where their relative weights change, a situation which some other analyses of tempering-based methods are able to tackle; this would mean that in multimodal settings, the bounds in the paper are only effiicient when the relative weights don't change. Can the authors please clarify when they expect the action to be polynomial, and how this assumption relates to other "beyond isoperimetry" assumptions in the literature (e.g. multimodal/mixture settings)? Under what settings do we expect the action to be reasonable? (The reverse diffusion sampler is an example where this is nice, which I appreciate.)

I would be happy to increase my score if these points are satisfactorily addressed.

**Questions:**

In Theorem 4, the authors interpolate from a log-concave distribution, with the partition function for the log-concave distribution estimated separately via TI. However, one could also attempt to use JE through the entire sequence from the Gaussian approximation. How would these compare theoretically or experimentally? Is there a reason to prefer one or the other besides convenience of theoretical analysis?

The statement of Lemma 6 is a bit confusing, since what is meant is that the conditions are to be satisfied for small enough constants, not arbitrary constants. Please clarify this in the lemma statement (right now this is only in Remark 5).

---

> ### Author Response · Authors · 2025-11-21
>
> Thank you for carefully reading our paper and providing thoughtful comments and suggestions. We appreciate your positive feedback, and are glad that you find our work timely and well-written. Please find our responses to your concerns below.
>
> > **Weakness 1**: More general bound that works for an arbitrary annealing sequence and added drift
>
> Thank you for raising this insightful question.
>
> First, we would like to clarify that our continuous-time result, the JE (Thm. 2), holds for *any* interpolation curve $(\pi _ \theta) _ {\theta\in[0,1]}$ under mild regularity conditions. For the case of an additional drift you mentioned (e.g., parameterized through a neural network), this is exactly the idea behind **controlled Monte Carlo diffusion** (**CMCD**, Vargas et al. (2024)), which can be adapted for normalizing constant estimation. Please check App. H.2 for a literature review on CMCD and how we adapt our analysis to it for normalizing constant estimation (Prop. 5) -- in particular, inference-time performance guarantee given the training loss on $u$.
>
> Moreover, we also provide a Thm. 2-like result as follows. Consider the following forward and backward SDEs with an additional drift term $u _ t$:
>
> $$\mathcal{P}^\to:\mathrm{d}X _ t=(\nabla\log\widetilde\pi _ t(X _ t)+u _ t(X _ t))\mathrm{d}t+\sqrt{2}\mathrm{d}B _ t, ~ X _ 0\sim\widetilde\pi _ 0;$$
> and
> $$\mathcal{P}^\gets:\mathrm{d}X _ t=(-\nabla\log\widetilde\pi _ t(X _ t)+u _ t(X _ t))\mathrm{d}t+\sqrt{2}\mathrm{d}B^\gets _ t, ~ X _ T\sim\widetilde\pi _ T,$$
>
> where $u _ t$ is an arbitrary drift term. One can compute $\log\frac{\mathrm{d}\mathcal{P}^\to}{\mathrm{d}\mathcal{P}^\gets}(X)=W(X)+C^u(X)-\Delta F$ through Girsanov's theorem (Lem. 4), and obtain the generalized JE $\mathbb{E} _ {\mathcal{P}^\to}\mathrm{e}^{-W(X)-C^u(X)}=\mathrm{e}^{-\Delta F}$. Here, $W(X)$ is the same functional defined in JE (Thm. 1), and $C^u(X)=-\int _ 0^T(\langle u _ t(X _ t),\nabla\log\widetilde\pi _ t(X _ t)\rangle+\nabla\cdot u _ t(X _ t))\mathrm{d}t$ is the correction term due to the additional drift.
>
> To analyze the estimation error, let $\mathcal{P}$ be the optimal path measure defined in the proof of Thm. 2 (more specifically, (17), with an compensating vector field $v _ t$). Then, by applying the Girsanov's theorem again, we obtain
> $$\operatorname{KL}(\mathcal{P}||\mathcal{P}^\to)=\operatorname{KL}(\mathcal{P}||\mathcal{P}^\gets)=\frac{1}{4}\int _ 0^T||u _ t-v _ t||^2 _ {L^2(\widetilde\pi _ t)}\mathrm{d}t,$$
> and thus one can bound the probability that the relative error is $\ge\varepsilon$ through the same argument in (18):
> $$\Pr\left(\left|\frac{\widehat Z}{Z}-1\right|\ge\varepsilon\right)\lesssim\frac{\sqrt{\operatorname{KL}(\mathcal{P}||\mathcal{P}^\to)}+\sqrt{\operatorname{KL}(\mathcal{P}||\mathcal{P}^\gets)}}{\varepsilon}\lesssim\frac{\sqrt{\int _ 0^T||u _ t-v _ t||^2 _ {L^2(\widetilde\pi _ t)}\mathrm{d}t}}{\varepsilon}.$$
>
> Although the proof can be generalized, in the main text of presenting the JE, we only focus on the case where this additional drift is zero, which rules out the influence of the additional drift in order to highlight the relationship between the estimation error and the *intrinsic property of the interpolation curve*. Finally, these results definitely provide guarantees for *sampling*, as we have obtain the expression on the KL divergence between two path measures $\operatorname{KL}(\mathcal{P}||\mathcal{P}^\to)$ above, which immediately gives KL bounds for final-time marginal distributions through the data processing inequality.
>
> Moreover, our analysis on AIS, a time-discretized version of JE, can also be generalized to (i) arbitrary annealing schemes $(\pi _ \theta) _ {\theta\in[0,1]}$ and (ii) arbitrary additional drift term $u _ t$, through similar approaches. The discrizatization analysis is similar if we know the Lipchitz bounds on the score and drift term. The use of the geometric interpolation is mainly for the convenience of theoretical analysis and presentation. Also, it is a representative annealing sequence most widely used in practice.
>
> **Reference**: Vargas et al. Transport meets variational inference: Controlled Monte Carlo diffusions. ICLR 2024.
>
> > **Weakness 2.1**: Not clear whether the action matches other bounds for annealing
>
> Thanks for raising this important question. As our approach is different compared with existing works on annealing/tempering, it is not easy to compare whether the bounds can be matched directly. For instance,  Woodard et al. (2009) leveraged the analysis of spectral gap (or equivalently, Poincaré constant) of tempering chains, and Chehab et al. (2025) leveraged the analysis of log-Sobolev constant. In contrast, our analysis is based on the Girsanov's theorem and the results in optimal transport, which leads to the action-dependent bounds. Note, however, that as indicated in our discussion after Thm. 2, our analysis matches the existing asymptotic results.

---

> > ### Author Response · Authors · 2025-11-21
> >
> > > **Weakness 2.2**: Lower bound on the action in multimodal settings with changing relative weights; this may cover cases where we want to use annealing-based methods
> >
> > We completely agree with your intuition that if the weights of different modes change, then annealing will face mode-switching issues, which leads to large action. We believe this is a fundamental issue for some annealing-based MCMC samplers such as AIS studied in this paper, where the particles can not *jump between the modes*. This issue may be possibly alleviated through more advanced annealing techniques such as simulated tempering or replica exchange (also known as parallel tempering), where particles can jump between the modes, or by the RDS, which uses the OU process, a curve without mode switching, although the score does not have closed form expression.
> >
> > We appreciate your suggestion to extend the technique for establishing action lower bounds in Prop. 1 to more general multimodal settings, which we agree is an interesting future direction. Below, we present a preliminary result on the effect of weight change on mixture distributions. We hope this would provide some insight into the study of more general case of mixture distributions.
> >
> > **Proposition.** Consider the curve of arithmetic interpolation $\pi _ \theta=(1-\theta)\pi _ 0+\theta\pi _ 1$, where we assume $\pi _ 0$ and $\pi _ 1$ are one-dimensional distributions having sub-exponential tails around their respective modes $m _ 0$ and $m _ 1$, i.e., there exist constants $c\ge2$, $0\le r\le\frac{m _ 1-m _ 0}{2}-1$, such that
> > $$\pi _ 0(x)\le \mathrm{e}^{-c|x-m _ 0|},~\forall|x-m _ 0|\ge r; ~~ \pi _ 1(x)\le \mathrm{e}^{-c|x-m _ 1|}, ~ \forall|x-m _ 1|\ge r,$$
> > and assume further that the mode separation $m _ 1-m _ 0\ge4$. Then, the action of this curve is $\mathrm{e}^{\Omega(m _ 1-m _ 0)}$.
> >
> > **Proof.** Following the same argument as in the proof of Prop. 1, let $F _ \theta$ be the c.d.f. of $\pi _ \theta$, then
> > $$|\dot\pi| _ \theta^2=\int\frac{(\partial _ \theta F _ \theta(x))^2}{\pi _ \theta(x)}\mathrm{d}x=\int\frac{(F _ 0(x)-F _ 1(x))^2}{(1-\theta)\pi _ 0(x)+\theta\pi _ 1(x)}\mathrm{d}x.$$
> >
> > Consider the integral on the interval $\left[\frac{m _ 0+m _ 1}{2}-1,\frac{m _ 0+m _ 1}{2}+1\right]$. We know that both $\pi _ 0$ and $\pi _ 1$ are exponentially small with respect to the mode separation, i.e., $\pi _ 0(x),\pi _ 1(x)\le\mathrm{e}^{-\frac{c}{4}(m _ 1-m _ 0)}$ for all $x$ in this interval. Meanwhile, for any $x$ in this interval, we have
> > $$\sup _ x(1-F _ 0(x))\le\sup _ x\int _ x^\infty\mathrm{e}^{-c|y-m _ 0|}\mathrm{d}y=\int _ {\frac{m _ 0+m _ 1}{2}-1}^\infty\mathrm{e}^{-c|y-m _ 0|}\mathrm{d}y=\int _ {\frac{m _ 1-m _ 0}{2}-1}^\infty\mathrm{e}^{-cy}\mathrm{d}y\le\int _ 1^\infty\mathrm{e}^{-cy}\mathrm{d}y=\frac{\mathrm{e}^{-c}}{c}.$$
> >
> > Likewise, we have $\sup _ x F _ 1(x)\le\frac{\mathrm{e}^{-c}}{c}$. Therefore, for any $x$ in this interval, $F _ 0(x)-F _ 1(x)\ge1-\frac{2}{c\mathrm{e}^c}\ge\frac12$, so $|\dot\pi| _ \theta^2\gtrsim\mathrm{e}^{\Omega(m _ 1-m _ 0)}$, which directly implies exponential lower bound on the action. $\square$
> >
> > > **Weakness 2.3**: When to expect polynomial action
> >
> > We provide two possible directions for finding curves with polynomial action.
> >
> > **(I)** Curves with geometric conditions. Such conditions may be on the weights of different modes (e.g., geometric annealing v.s. Gaussian convolution and the OU process used in RDS), the relative locations of modes (e.g., Guo et al. (2025)'s Gaussian mixture with all centers having the same norm and the same covariance), etc. For instance, suppose $\pi _ \theta$ can be written as $\pi _ \theta(x)=\sum _ {i=1}^k w _ i\nu^{(i)} _ \theta(x)$, where the weights $w _ i$ are independent of $\theta$, and each curve $(\nu^{(i)} _ \theta) _ {\theta\in[0,1]}$ has polynomially large action. Then one can construct a coupling $(X,X')$ of $(\pi _ \theta,\pi _ {\theta+\delta})$ as follows: $\Pr(I=i)=w _ i$, and conditional on $I=i$, $X\sim\nu^{(i)} _ \theta$, $X'\sim\nu^{(i)} _ {\theta+\delta}$ are optimally coupled such that $\mathbb{E}[||X-X'||^2|I=i]=W _ 2^2(\nu^{(i)} _ \theta,\nu^{(i)} _ {\theta+\delta})$. This implies $W _ 2^2(\pi _ \theta,\pi _ {\theta+\delta})\le\sum _ {i=1}^k w _ iW _ 2^2(\nu^{(i)} _ \theta,\nu^{(i)} _ {\theta+\delta})$, and thus $|\dot\pi| _ \theta^2\le\sum _ {i=1}^k w _ i|\dot\nu^{(i)}| _ \theta^2$. This is the key why RDS avoids mode switching.
> >
> > **(II)** Wasserstein geodesic or Wasserstein gradient flow. On a Wasserstein geodesic, the action is exactly the squared Wasserstein distance between the two end distributions; for Wasserstein gradient flow for the functional $\operatorname{KL}(\cdot||\phi)$ starting from $\pi$, we have modified the proof of Prop. 2 to show that the action of this curve is exactly $\operatorname{KL}(\pi||\phi)$, which can be polynomial under weak assumptions. However, for both cases, the interpolation curves are typically not available in closed form.

---

> > > ### Author Response · Authors · 2025-11-21
> > >
> > > It is an interesting future direction to further investigate sufficient conditions for polynomial action in more general settings, and we appreciate your suggestions on this.
> > >
> > > **Reference**:
> > >
> > > - Woodard et al. Sufficient conditions for torpid mixing of parallel and simulated tempering. Electronic Journal of Probability, 2009.
> > > - Chehab et al. Provable convergence and limitations of geometric tempering for Langevin dynamics. ICLR 2025.
> > > - Guo et al. Provable benefit of annealed Langevin Monte Carlo for non-log-concave sampling. ICLR 2025.
> > >
> > > > **Question 1**: Using JE through the entire sequence from the Gaussian approximation
> > >
> > > Thank you for this interesting question. Indeed, in practice one can also use JE / AIS through the entire sequence from an delta-like distribution with $\Theta\left(\frac{d\beta}{\varepsilon}\right)$ condition number (which enables accurate estimaion of normalizing constant in one step), rather than the $\pi _ 0$ we choose in our paper whose potential only has $O(1)$ condition number.
> > >
> > > The reason for doing so in our paper is that, for well-conditioned (i.e., with strongly convex and smooth potential) distributions, TI has a better complexity bound than AIS, which is $O\left(\frac{d^{4/3}}{\varepsilon^2}\right)$ as proved in Ge et al. (2020). Our analysis on AIS, under much *weaker* assumptions than strong log-concavity, results in a worse complexity bound of $O\left(\frac{d\beta^2\mathcal{A}}{\varepsilon^4}\right)$ is this case, which is due to our approach based on Girsanov's theorem.
> > >
> > > Finally, as the *dominant complexity* lies in the sampling for the non-log-concave distributions, by shifting to using AIS through the entire sequence, the final result will still be the order same up to constant factors, depending on the action of the curve on the non-log-concave part. In our experimental results, both the AIS and TI are performed through the entire sequence from a delta-like distribution, and neither work well in tackling multimodality.
> > >
> > > **Reference**: Ge, Lee, Lu. Estimating Normalizing Constants for Log-Concave Distributions: Algorithms and Lower Bounds. STOC 2020.
> > >
> > > > **Question 2**: Clarification on Lem. 6
> > >
> > > Thanks for pointing this out. We have clarified this in the revised manuscript.
> > >
> > > ---
> > >
> > > We would like to thank you again for your valuable feedback, which has greatly helped us improve the quality of our work, and we sincerely appreciate your willingness to raise the score. We hope these changes satisfactorily address your points and would welcome an updated evaluation; if anything remains unclear, we will provide further clarifications as needed.

---

### Official Review · Reviewer_J6xT · 2025-11-01

**Soundness:** 3
**Presentation:** 3
**Contribution:** 3
**Rating:** 6
**Confidence:** 3

**Summary:**

The work proposes a scheme for normalizing constant estimation based on the annealed Langevin framework. It shows, via a relatively novel annealing scheme, that one can achieve multiplicative accuracy with number of queries depending polynomially on the action of the curve, defined as the integrated norm of the metric derivative (in a Wasserstein sense).

**Strengths:**

I think this is an interesting and useful result, providing a principled means of normalizing constant estimation with interpretable guarantees. I concur with the authors that this is an important problem in computational statistics.

The connections between this and annealed sampling algorithms are elegant and nicely parallel the connection in discrete settings. This fact is also noticed by the authors.

The connections to statistical mechanics are always nice to see.

**Weaknesses:**

I find the action of a curve to be a somewhat inscrutable quantity from the perspective of algorithm design. The authors do a good job of trying to make this quantity accessible to unfamiliar readers but it certainly merits further investigation.

**Questions:**

(2) and (3) do not seem to be time reversals of each other? Or is ``backward’’ meant in a more pedestrian sense?

23: Girsanov theorem -> Girsanov’s theorem

111: denoising diffusion model -> denoising diffusion models

136: estimating normalizing constant -> estimating the normalizing constant

287: for the study non-asymptotic -> for the study of non-asymptotic

390: full proof is in -> The full proof is in

463: biased estimate -> biased estimates

---

> ### Author Response · Authors · 2025-11-21
>
> Thank you for your comments and suggestions. We are glad that you acknowledge the significance of the question we address.
>
> > **Weakness**: The action of a curve to be a somewhat inscrutable quantity from the perspective of algorithm design
>
> Thank you for this comment. We acknowledge that the action is an abstract quantity that is non-trivial to bound for general class of interpolation curves, and we appreciate your suggestion on this important future direction.
>
> In this paper, the actions for geometric interpolation and OU process are meant to provide insights on how to design better interpolation curves, and explain why OU-based RDS algorithms and diffusion model work better than geometric interpolation in practice. In particular, as discussed in the paper, the OU process avoids the mode-switching issue during annealing, which leads to smaller action. We believe that this insight can inspire the design of better interpolation curves that avoid mode-switching issues.
>
> > **Question**: Backward SDEs, (2) and (3)
>
> You are completely correct that (2) and (3) are not time reversals of each other. The "backward" here refers to the **backward SDE** with time-reversed Brownian motion $(B^\gets _ t) _ {t\in[0,T]}$, i.e., $B^\gets _ {t-\Delta t}-B^\gets _ t\sim\mathcal{N}(0,\Delta tI)$ and is independent of $(B^\gets _ s) _ {s\in[t,T]}$. See App. A.1 for a detailed introduction.
>
> The backward LD (3), as discussed in the second paragraph after Thm. 2, can be seen as a time-reversed ALD starting from $X _ T\sim\pi _ T$ at time $T$ and going to $0$, which targets $\pi _ t$ at time $t$. One can also perceive it as the time-reversal of the following SDE with standard Brownian motion $(\hat B _ t) _ {t\in[0,T]}$:
>
> $$\mathrm{d}Y _ t=\nabla\log\widetilde\pi _ {T-t}(Y _ t)\mathrm{d}t+\sqrt{2}\mathrm{d}\hat B _ t, ~ t\in[0,T], ~ Y _ 0\sim\widetilde\pi _ T,$$
>
> such that $Y _ t$ and $X _ {T-t}$ have the same distribution for all $t\in[0,T]$, where $X$ follows the backward LD (3). We have included this discussion in the sketch of proof to help the reader better understand the backward LD.
>
> (2) and (3) are not time reversals of each other because their marginal distributions at each time $t$ can't be $\widetilde\pi _ t$ due to the lag of the ALD.
>
> -----
>
> Thanks for pointing out the typos and grammatical errors. We have fixed them in the revised manuscript. We hope that the above explanations address your concerns. We will be happy to respond any further questions or additional comments, and we also welcome an updated evaluation.

---

### Author Response · Authors · 2025-11-21
**General Response**

We would like to express our sincere gratitude to all reviewers for carefully reading our paper and providing insightful comments and valuable suggestions.
We are pleased that the reviewers find our work addresses an important problem (J6xT, TbKt, 8vBo), is well-written (TbKt), and has novel theoretical contributions (J6xT, s4rX).
The main concerns raised the reviewers are:

1. Presentation of the paper due to high technicality (s4rX, 8vBo).
2. Limitations of the results for discrete-time AIS (Thm. 4) on only geometric interpolation (TbKt, s4rX).

For presentation issues, we have revised the manuscript to improve clarity and accessibility, where the major changes are highlighted in $\color{orange}{\rm orange}$. For the second concern on AIS, the use of geometric interpolation is mainly for the sake of simplicity and clarity, which is also the most widely used annealing scheme in practice. The same discretization analysis can be generalized to general curves $(\pi _ \theta) _ {\theta\in[0,1]}$ and/or with additional drift terms $u _ t$ as in CMCD (App. H.2). Also, this example demonstrates the possible drawbacks of geometric interpolation in multimodal settings.

We provide detailed responses to each reviewer's comments below. We hope that they could address the reviewers' concerns, and we also welcome any further questions or comments.

---

### Author Response · Authors · 2025-12-03
**Rebuttal Overview -- Key Contributions, Concerns, Responses, and Updates**

We thank all reviewers again for their valuable feedback and constructive comments, and we also thank PCs, SACs and ACs for their time and effort in managing the review process in face of the OpenReview system issues during the rebuttal period. As the reviewers cannot engage in further discussions anymore at the current stage, we summarize the main points of our rebuttal here for clarity for the ACs in hope that it can help with the final decision.

# Contribution and Novelty of our Work

This paper studies the algorithmic complexity of estimating the normalizing constant of an unnormalized probability distribution $\pi\propto\mathrm{e}^{-V}$ on $\mathbb{R}^d$, a fundamental problem in Bayesian statistics, statistical mechanics, and machine learning, with wide applications including free-energy difference estimation, marginal likelihood computation, etc. We provide analysis for Jarzynski equality (JE), annealed importance sampling (AIS), and a class of reverse diffusion sampler (RDS)-based algorithms. Our main contributions and novelties are as follows:

- **Novel framework**: We develop a novel strategy to establish complexity bounds of normalizing constant estimation through Girsanov's theorem, optimal transport, and forward/backward SDEs.
- **General applicability**: Our analysis for JE (Thm. 2) applies to **general target distributions** and **interpolation curves** without restrictive assumptions such as log-concavity, while the whole proof for analyzing the AIS (Thm. 4) is also adaptable to general settings in essence, although we only present the bound on geometric interpolation (7) for simplicity.
- **New insights**: Our results reveal the **intrinsic connection** between the estimation error and a geometric quantity of the interpolation curve named **action**, which offers new insights into the design of effective annealing scheme. In particular, we point out a potential limitation of the commonly used geometric interpolation, which serves as a quantitative explanation of the **mass teleportation** phenomenon.

Overall, we believe that our work makes significant contributions to the theoretical understanding of normalizing constant estimation, and is to inspire future research in this direction both in theory and practice.

# Main Concerns Raised by Reviewers and our Responses

We thank the reviewers for their careful reading of our paper and insightful comments. We summarize the representative concerns shared by the reviewers and our response as follows.

## (I) Discrete-time results only for geometric interpolation

The reviewers TbKt, s4rX, and 8vBo pointed out that our analysis of AIS (Thm. 4) is only proved for geometric interpolation, and asked whether they can be extended to more general settings.

In our rebuttal, we clarified that the use of geometric interpolation is

- To focus on the **most widely used annealing scheme** in practice;
- To demonstrate the **possible drawbacks** of geometric interpolation in multimodal settings though with an easily-accessible form of the score;
- For the sake of **simplicity and clarity** in presentation.

Furthermore, we emphasize that the same discretization analysis can be generalized to

- Any absolute continuous curve $(\pi _ \theta) _ {\theta\in[0,1]}$ whose score $\nabla\log\pi _ \theta$ is known and has known Lipschitz constant;
- With an additional drift term $u _ t$ as in CMCD (App. H.2), also with known Lipschitz constant.

Further discussion can be found in our response to the weakness 1 from reviewer TbKt and the last paragraph of Sec. 4.

---

> ### Author Response · Authors · 2025-12-03
>
> ## (II) Interpretability of action
>
> The **action**, which is the integral of the squared Wasserstein-2 metric derivative along the curve of probability distributions, is a key property to quantify the efficiency of the annealing curve for normalizing constant estimation. The reviewers raised conceptual questions on this quantity regarding
>
> 1. How action offers insight on *algorithm design* (J6xT);
> 2. Whether the *lower bound* on action (Prop. 1) can be extended to mixture distributions with *changing mode weights* (TbKt);
> 3. When to expect *polynomially large action* (TbKt);
> 4. The *dimensional dependence* on the action (s4rX).
>
> In response,
>
> 1. We pointed out that the actions for geometric interpolation explained why OU-based RDS algorithms and diffusion model work better than geometric annealing in practice by avoiding the mass teleportation issue.
> 2. The reviewer had a correct intuition -- we provided an action lower bound for arithmetic interpolation in the response to the weakness 2.2 of reviewer TbKt.
> 3. We mentioned two possible classes: (i) curves with geometric conditions, such as preservation of weights in mixture; (2) Wasserstein geodesic or Wasserstein gradient flow. See our response to the weakness 2.3 of reviewer TbKt.
> 4. We provided an $O(d)$ bound on the action when each $\pi _ \theta$ is the tensor product of $d$ one-dimensional distribution $\nu _ \theta$. However, we also acknowledge that this bound is typically problem-dependent, e.g., if the number of modes grows with dimension, the action may have super-polynomial dependence on $d$.
>
> ## (III) Improving presentation of the paper
>
> Finally, the reviewers s4rX and 8vBo hoped to see an improvement of the presentation due to the paper's high technicality. We have seriously taken these suggestions into consideration. Please see the next part for a summary of the main changes to the manuscript.
>
> # Improvements of the Paper during Rebuttal
>
> In addition to our response to the reviewers' comments, we have also made significant improvements to the paper during the rebuttal period. Please check our updated manuscript, in which notable changes are highlighted in $\color{orange}{\rm orange}$, including:
>
> - **Reorganization of the content**: We moved the notations and definitions from App. A.1 to Sec. 1 (s4rX); we removed the informal lemma of the connection between the metric derivative and the continuity equation in the main text, and only presented its full version in App. A.2 (s4rX).
> - **Better incorporation of the preliminary knowledge on-the-fly**: We integrated more background knowledge into the main text, such as Langevin diffusion and backward SDE in Sec. 3, to make it more accessible for general audience (s4rX).
> - **Further clarifications**: We discussed the limitations and possible extensions of the analysis of AIS at the final paragraph of Sec. 4 (TbKt, s4rX), included an intuitive explanation of mass teleportation after Prop. 1 (s4rX), rewrote the proof of Prop. 2 to generalize it to the curve of Wasserstein gradient flow of KL (TbKt), highlighted the purpose of the {RDMC, RSDMC, ZODMC, or SNDMC} algorithms after Eq. (13) and in Alg. 2 (s4rX), and explained of the hyperparameters used in the bound of RDS' after Thm. 5 in the footnote (s4rX).
>
> # Concluding Remarks
>
> We thank the reviewers and ACs again for their time and effort in reviewing our paper and managing the review process. We hope that our responses and improvements during the rebuttal period have adequately addressed the concerns raised by the reviewers. We will continue to improve the paper based on the valuable feedback received.

---

### Meta-Review · Area_Chair_P1b5 · 2025-12-27

**Summary:**

This paper considers the problem of estimating the normalizing constant for a probability density distribution using annealed importance sampling. Its main contribution is a new upperbound on the oracle complexity for estimating Z with \epsilon error, based on a quantity \mathcal{A} which is the "action" of a curve of probability measures. This new quantity does not directly require isoperimetric conditions and as the paper shows can apply to a few interesting cases (although as the paper also noted it certainly can be exponential). Overall the reviewers find the new bound to be interesting and timely. There were some concerns on how specific the theorems are, how the "action" term can be interpreted and on the overall presentation of the paper, but most of these seem addressed in the author response.

**Reviewer Concerns:**

Several reviewers were concerned about whether the proof can be generalized beyond geometric interpolation. This seems to be well addressed by the author response.

Another main concern on the interpretability of the action term and how that compares with some existing results. The authors provided some explanations and special cases, although the authors also admitted that it's hard to compare this bound with some other existing bounds.

There are many suggestions on the presentations, which are partly addressed in the new version. The paper probably still needs a more careful pass to make the presentation most clear.

**Reviewer Scores:**

Reviewer J6xT: score unlikely to change
Reviewr TbKt: score might be increased to 7
Reviewer S4rX: this is a careful review with a score that seems lower than what the review itself suggests, while many of the points are addressed to some degree I don't know if they will satisfy the reviewer. I do expect the score to at least be a 4 or 5 just based on what was written and how they were addressed.
Reviewer 8vBo: score unlikely to change

---

### Decision · Program_Chairs · 2026-01-26

Accept (Poster)